# SAMPLE EFFICIENT CORRECTIVE DEEP UNLEARNING

## ABSTRACT

Machine unlearning enables machine learning models to selectively forget a subset of training data, ensuring compliance with privacy laws and allowing for the efficient removal of outdated or harmful data samples. Current machine unlearning algorithms are restricted to specific models or are applicable only to a subset of learning and unlearning settings, while requiring full knowledge of data points to unlearn. In this paper, we propose a sample efficient corrective deep unlearning algorithm that achieves competitive empirical performance across various unlearning settings without degrading model performance. Our experiments demonstrate that our algorithm achieves strong unlearning performance while requiring only a small computation budget and a small unlearning sample size, thus making it a viable solution for scalable and practical machine unlearning.

## 1 INTRODUCTION

Modern Machine Learning (ML) models are trained on large datasets and are increasingly deployed in safety-critical applications such as healthcare and self-driving cars. In many such scenarios with risk, data integrity is critical, yet models may be trained with corrupted, poisoned, or otherwise manipulated data (Paleka & Sanyal, 2023). Retraining large models from scratch to remove such data is computationally expensive. Machine unlearning provides a promising approach to address these challenges by selectively removing the influence of undesired data samples without full retraining (Cao & Yang, 2015).

Early unlearning methods were proposed to support privacy by enabling the removal of data for compliance with data deletion requests (Bourtoule et al., 2021; Ginart et al., 2019). However, recent studies have shown that privacy-driven unlearning can be fragile (Goel et al., 2022; Thudi et al., 2022), and is vulnerable to relearning attacks (Hu et al., 2024). In the present paper, we consider applying unlearning to the goal of correcting for the effects of corrupted training data. Goel et al. (2024) introduced the notion of *corrective unlearning*, which focuses on mitigating the impact of corrupted data samples, the so-called forget set, while knowledge of only a subset of it. Most existing unlearning methods were designed for privacy goals and are tailored to either classification or regression tasks, and adapting them to other problems proves non-trivial (Tarun et al., 2023). These limitations highlight the need for efficient, robust, and broadly compatible corrective unlearning algorithms.

We propose *SPARC (Selective Parameter Adjustment and ReCalibration)*, a novel unlearning algorithm that identifies and updates weights associated with the forget set by leveraging the idea that similar inputs follow similar *activation paths* through the network. Previous works (Goel et al., 2024; Pawelczyk et al., 2024) have demonstrated that existing state-of-the-art unlearning algorithms fail at corrective unlearning and are more focused on achieving privacy guarantees. SPARC is designed specifically for correcting unlearning demonstrates strong performance in removing the influence of forget samples with limited knowledge across both classification and regression tasks, while retaining utility in the original prediction task.

Our contributions in this paper are as follows:

- We propose SPARC, an efficient algorithm for corrective deep unlearning applicable across architectures and learning tasks.
- We demonstrate the strong performance of our algorithm under limited knowledge of corrupted data and on a range of datasets and models in various corrective unlearning settings.

## 1.1 RELATED WORK

**Definitions of Unlearning.** Machine unlearning was first framed as exact alignment with a retraining-from-scratch without the forget set (Cao & Yang, 2015). To mitigate the cost of retraining, approximate definitions emerged, drawing on differential privacy (Ginart et al., 2019; Guo et al., 2020) and evaluated via metrics such as KL divergence (Golatkar et al., 2020) or membership inference attacks (Ma et al., 2023). These views have been criticized as restrictive and fragile (Goel et al., 2022; Thudi et al., 2022). More recently, researchers have argued for task- or application-specific objectives (Kurmanji et al., 2023). Of particular relevance, Goel et al. (2024) introduced the *corrective unlearning* paradigm, which explicitly addresses adversarial manipulations such as poisons or backdoors.

**Algorithmic Approaches.** Exact approaches such as SISA (Bourtoule et al., 2021) guarantee strict removal via segmented training and rollback, but incur heavy data-management overhead. Most recent work pursues approximate unlearning: SCRUB (Kurmanji et al., 2023) uses teacher–student min–max training, SSD (Foster et al., 2024) relies on Fisher information for parameter damping, and Blindspot and Gaussian-Amnesiac adapt targets for regression settings (Tarun et al., 2023). Other related methods mention the context of corrective unlearning, but do not directly address it. Delta-Influence (Li et al., 2024) leverages influence functions to detect and flag poisoned samples, serving primarily as a data filtering method rather than an unlearning algorithm. Similarly, SAP (Kodge et al., 2025) applies projection techniques to mitigate label noise, but its focus is robustness to mislabeled data rather than removing the effects of adversarial manipulations or backdoors. We therefore view these approaches as complementary but distinct from corrective unlearning. A parallel thread adapts multi-task gradient surgery (e.g., PCGrad) for conflict-aware updates, though typically in continual or multi-task learning rather than unlearning.

**Our positioning.** Overall, prior methods either enforce costly retraining guarantees or rely on influence/detection mechanisms with different objectives. Projection-based corrections highlight the utility of gradient geometry, while Fisher-based or min–max schemes emphasize parameter importance or distributional alignment. Building on these insights, our two-stage method for corrective unlearning combines selective parameter adjustment with lightweight gradient-based recalibration.

## 2 PROBLEM FORMULATION

We formalize the machine unlearning problem through the lens of *corrective unlearning* (Goel et al., 2024). Let $D_{\text{train}}$ denote a training dataset containing $n$ samples of the form $(x, y)$, where $x \in \mathcal{X}$ denotes the features and $y \in \mathcal{Y}$ denotes the labels. Let $M_o(\theta)$ be an ML model parameterized by weights $\theta$ trained on this dataset using $A$, a learning algorithm. We refer to $M_o$ as the original model. Let $D_u \subset D_{\text{train}}$ be the set of samples to be unlearned. Note that depending on the application setting, we may also have a set $D_{\text{test},u} \subset D_{\text{test}}$ that are test samples from the same domain as $D_u$. An unlearning algorithm $U$ updates the original model $M_o$ to $M_u$, with the twin goals of ensuring that $M_u$ does not retain knowledge of the unlearning set $D_u$ and that $M_u$, still generalizes well, that is, performs well on $D_{\text{test}}$, the test dataset.

The simplest, naive method for unlearning is to retrain: run $A$ again on $D_{\text{train}} \setminus D_u$ to obtain an updated model. This is not always feasible in practice due to the computational complexity of retraining a new model. Further, depending on the unlearning application, we may not have knowledge of all samples in $D_u$. We thus consider the setting where the goal is to remove the influence of samples in $D_u$ by updating the model parameters $\theta$ in a more efficient manner and using only a subset $D_f \subset D_u$, while preserving the performance on a retain dataset $D_r = D_{\text{train}} \setminus D_u$ and the test dataset $D_{\text{test}}$.

The unlearning algorithm $U$ takes as input $M_o$, $D_{\text{train}}$, and $D_f$ and gives an unlearned model $M_u$. The performance of $U$ is evaluated by two corrective unlearning evaluation axes:

1. *Unlearning Success* measures how well the unlearning algorithm corrects for the influence of samples in the forget set, and indeed all unlearn samples $D_u$ and $D_{\text{test},u}$.

2. *Utility* measures how well $M_u$ preserves performance on the retained samples $D_r$ and on the test samples $D_{\text{test}}$.

Table 1: Unlearning Success and Utility metrics for various unlearning settings

| Unlearning Setting | Unlearning Success | Utility |
|---|---|---|
| Label-and-Feature Manipulation | $\mathbb{E}\left[\mathbb{I}(M_u(x_m) = y)\mid(x_m, y) \in D_{trigger}\right]$ | $\mathbb{E}\left[\mathbb{I}(M_u(x) = y)\mid(x, y) \in D_{test}\right]$ |
| Label-only Manipulation | $\mathbb{E}\left[\mathbb{I}(M_u(x) = y)\mid(x, y) \in D_{\text{test},u}\right]$ | $\mathbb{E}\left[\mathbb{I}(M_u(x) = y)\mid(x, y) \in D_{test}\right]$ |
| Feature-only Manipulation | $\mathbb{E}\left[\mathbb{I}(M_u(x) = y)\mid(x, y) \in D_{\text{target}}\right]$ | $\mathbb{E}\left[\mathbb{I}(M_u(x) = y)\mid(x, y) \in D_{test}\right]$ |
| Classification | $\mathbb{E}\left[\mathbb{I}(M_u(x) \neq y)\mid(x, y) \in D_{test,u}\right]$ | $\mathbb{E}\left[\mathbb{I}(M_u(x) = y)\mid(x, y) \in D_{test,r}\right]$ |
| Regression | $\mathbb{E}\left[M_u(x) \notin [r_1, r_2]\mid(x, y) \in D_{\text{test},u}\right]$ | $\mathbb{E}\left[\|M_u(x) - y\|\mid(x, y) \in D_{\text{test},r}\right]$ |

These objectives are inherently in tension: stronger corrections can reduce general utility. We show empirically in Section 5 that SPARC performs well on both these measures as compared to previous methods.

## 2.1 UNLEARNING SETTINGS

We instantiate the corrective unlearning objective across multiple settings, covering both adversarial manipulations and broader unlearning tasks such as class or regression removal.

**Manipulation Unlearning**  We consider three types of data manipulations that can occur during training: *label-and-feature* manipulations, *label-only* manipulations, and *feature-only* manipulations. These manipulations commonly appear in the form of data poisoning or backdoor attacks, where an adversary injects or modifies a fraction of training samples to induce unwanted behavior in the trained model. The manipulated samples in the training set are denoted $D_u$ and the forget set, the set of samples identified as being manipulated, is a subset $D_f \subseteq D_u$. The retain set is $D_r = D_{\text{train}} \setminus D_u$.

Since the goal of unlearning here is to correct the model from manipulations, we measure unlearning success as the fraction of manipulated samples that are predicted correctly, as they would be without the corruption. This is formulated differently for each manipulation setting, as described below. For all manipulation settings, we measure utility by calculating the accuracy on the test set (Table 1).

**Label-and-feature manipulations: Backdoor Attack**  Manipulations that include both feature perturbations and label alterations using backdoor attacks in the training data were first proposed by Sommer et al. (2022).Following Goel et al. (2024), we use the BadNet backdoor attack (Gu et al., 2019) to install malicious backdoors in a trained neural network. In our experiments we insert a trigger pattern of a $3 \times 3$ white patch at the bottom-right part of the manipulated images. The labels of all these manipulated images are changed to 0, and this manipulated dataset is used to train the model.

Let $b$ be the backdoor target label, and $\phi$ be the modifier function that adds the backdoor to the input features. $D_{\text{trigger}} = \{(x_m = \phi(x), y) \mid \forall(x, y) \in D_{test}\}$ denotes the set of test samples with the backdoor trigger and original true label. A successful unlearning algorithm should correctly classify these triggered inputs as their true labels, rather than the target label. This forms our metric for success in this setting (Table 1).

**Label-only manipulations: Interclass Confusion Test**  The Interclass Confusion Test (ICT)(Goel et al., 2022) introduces manipulations by swapping the labels between two randomly chosen classes, denoted $c_1$ and $c_2$, for a fraction of samples from these classes during training, causing the model to confuse these classes. A successful unlearning algorithm should be able to remove the induced confusion.

The manipulated dataset is denoted $D\prime_{\text{train}}$. The unlearning test set is $D_{\text{test},u} = \{(x, y) \mid y \in \{c_1, c_2\}, \forall(x, y) \in D_{\text{test}}\}$. We measure unlearning success as the fraction of unlearn test samples that are correctly classified (Table 1).

**Feature-only manipulations: Poisoning Attack**  This is a targeted poisoning where the adversary's goal is to cause the model to misclassify a set of specific data points $D_{\text{target}} = \{(x_{\text{t}}, y_{\text{t}})\}$ from the test set $D_{\text{test}}$, to a pre-selected adversarial label $y_{\text{adv}}$. We use the gradient matching poisoning attack of Geiping et al. (2020) to generate feature-only manipulations, which selects a set of samples

$P$ with labels $y = y_{\text{adv}}$ from $D_{\text{train}}$ to poison, and adds perturbations to the features of these poison samples so as to align their gradients with that of the target sample. As proposed by Pawelczyk et al. (2024), we can evaluate the effectiveness of unlearning algorithms by assessing their ability to eliminate the influence of these feature-based manipulations and predicting the original label $y$ rather than $y_{\text{adv}}$ (Table 1).

**Classification Unlearning**   In the classification unlearning scenario the goal is to remove all samples belonging to a particular class, given a fraction of samples belonging to that class. The unlearned model should not predict the removed class. With $c_f$ denoting the class we want to forget, the unlearning set is $D_u = \{(x,y)|y = c_f, \forall (x,y) \in D_{\text{train}}\}$. The unlearning test set is $D_{\text{test},u} = \{(x,y)|y = c_f, \forall (x,y) \in D_{\text{test}}\}$ and the retain test set is $D_{\text{test},r} = D_{\text{test}} \setminus D_{\text{test},u}$. We measure unlearning success as the fraction of test samples belonging to class $c_f$ that are misclassified by the unlearned model $M_u$ and for utility we measure the accuracy on the retain test set (Table 1).

**Regression Unlearning**   In the regression unlearning setting as described in Tarun et al. (2023), the unlearn set consists of all samples where the true target is in a certain range $[r_1, r_2]$. Like classification unlearning, the unlearned model should not make any predictions in this range. The unlearning set is $D_u = \{(x,y)|y \in [r_1, r_2], \forall (x,y) \in D_{\text{train}}\}$, the unlearning test set is $D_{\text{test},u} = \{(x,y)|y \in [r_1, r_2], \forall (x,y) \in D_{\text{test}}\}$ and the retain test set is $D_{\text{test},r} = D_{\text{test}} \setminus D_{\text{test},u}$. We measure unlearning success as the fraction of test samples with their true target in the forget range $[r_1, r_2]$ whose predictions do not lie in the forget range and we use the mean absolute error on the retain test set to evaluate the utility of the unlearned model (Table 1).

# 3 SELECTIVE PARAMETER ADJUSTMENT AND RECALIBRATION (SPARC)

We now present our unlearning algorithm, SPARC, which performs selective parameter modification followed by efficient fine-tuning. While the unlearning applications vary across classification, regression, and other tasks, SPARC is broadly applicable due to its architecture-agnostic design. Existing machine unlearning algorithms are predominantly tailored to specific tasks, such as classification or regression, limiting their generalizability. In contrast, SPARC relies solely on a parameter's relative influence over the forget and retain sets, allowing it to operate on any neural network architecture, independent of the learning objective.

SPARC performs unlearning through two efficient and decoupled steps after computing parameter importance for the forget set relative to the retain set: (i) a targeted parameter adjustment step to selectively forget undesired data, and (ii) a recalibration phase that updates low-importance parameters to recover overall model utility. The parameter importance estimation is lightweight, requiring only forward passes through the network, making it significantly more efficient than gradient-based fine-tuning. Unlike existing baselines that entangle forgetting and utility restoration, often requiring multiple unlearning epochs, SPARC's explicit separation allows it to achieve strong performance with far fewer steps in practice (typically two epochs compared to ten in baselines) and with fewer forget samples. An estimated FLOPs comparison of SPARC and baseline algorithms is given in Section C.7.

## 3.1 INFLUENCE MEASURE

The importance of each model parameter is calculated with an influence measure. We use the intuition that the relative influence of a parameter with respect to certain data points should depend on the activations of units in the same path as that parameter. This intuition is further motivated by recent works that suggest activation pathways carry semantically meaningful traces of specific data or behaviors (Templeton et al., 2024; Lieberum et al., 2024). Reducing the values of parameters with a high influence measures on samples in $D_f$ should then reduce the influence of the samples in $D_u$, with minimal impact on samples in $D_r$.

Let $\theta_k^l$ be a parameter at layer $l$ of a neural network. We set $\mathcal{I}\left(\theta_k^l\right)$ to be the set of indices of activations units of layer $l-1$ that are multiplied by $\theta_k^l$, this is the node upstream to that parameter in feed forward neural network, or in the case of convolutional neural networks, the corresponding elements of the input activation map in convolutional layers. Similarly, $\mathcal{O}\left(\theta_k^l\right)$ is the set of indices

of activation units of layer $l$ that $\theta_k^l$ flows into. The model input will be denoted as layer 0. Let $a_i^l(x)$ denote the activation of the $i^{\text{th}}$ node in layer $l$ for the model input $x$ and $\phi : \mathbb{R} \to \mathbb{R}$ be a nonlinear function, such as the absolute value function $\phi(z) = |z|$, or the positive part function $\phi(z) = \max(0, z)$. We define the *mean activation difference* between the forget and retain sets as follows:

$$\bar{a}_i^l = \phi \left( \frac{\sum_{x \in D_f} a_i^l(x)}{|D_f|} - \frac{\sum_{x \in D_r} a_i^l(x)}{|D_r|} \right) \tag{1}$$

The *influence measure* of parameter $\theta_k^l$ then is defined as:

$$\mu_k^l = |\theta_k^l| \times \sum_{j \in \mathcal{I}(\theta_k^l)} \bar{a}_j^{l-1} \times \sum_{i \in \mathcal{O}(\theta_k^l)} \bar{a}_i^l \tag{2}$$

We normalize this score using a function $\mathcal{N}(\cdot)$, which can implement various normalization schemes, such as global normalization across all layers, layer-wise normalization, or a hybrid approach (e.g., normalizing within each layer and then scaling globally). The resulting *normalized influence measure* is defined as

$$\hat{\mu}_k^l = \mathcal{N}(\mu_k^l) \tag{3}$$

## 3.2 SELECTIVE PARAMETER ADJUSTMENT

The algorithm selectively reduces the parameter values based on their normalized influence scores and a threshold $\tau$. If $\hat{\mu}_k^l > \tau$ for parameter $\theta_k^l$, we update as follows,

$$\theta_k^l \leftarrow \max \left( 1 - \gamma . \hat{\mu}_k^l, 0 \right) \theta_k^l, \tag{4}$$

Here, the influence threshold $\tau$ is a hyperparameter that controls the threshold for identifying high-influence parameters. A lower $\tau$ means more parameters are treated as forget-relevant, resulting in more aggressive forgetting. Conversely, a higher $\tau$ makes the algorithm conservative, only adjusting parameters with extreme influence. The second hyperparameter, the forgetting factor $\gamma$, modulates how aggressively the parameter is reduced. It determines the scale of reduction for each parameter based on its influence. Larger $\gamma$ values lead to stronger forgetting but risk hurting overall performance. While the role of $\tau$ is to capture the set of parameters that are forget-relevant, the role of $\gamma$ is in determining the severity of forgetting using these filtered parameters.

## 3.3 RECALIBRATION: ORTHOGONAL GRADIENT DESCENT

There remains a risk that the forgetting procedure conflicts with the objective of preserving the knowledge of the retain set $D_r$. To address this issue, SPARC recalibrates low-influence parameters using orthogonal gradient descent, inspired by PCGrad (Yu et al., 2020), which is used for a very different and unrelated application. If the gradients on the forget and retain sets have the same direction, indicating some similarity, we project the retain set gradient (blue) orthogonal to the forget set gradient (red) as shown in Figure 1. This ensures that the model does not relearn knowledge from the forget set while preserving the original utility.

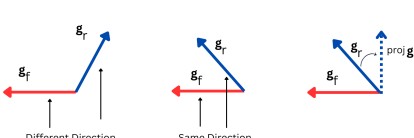

Figure 1: The orthogonal projection of the retain set gradients is used to update model parameters if the forget and retain set gradients have the same direction

Let $\mathbf{g}_f$ be the gradient for the samples in the forget set $D_f$ and $\mathbf{g}_r$ be the gradient for the samples in the retain set $D_r$ for all parameter $\theta_k^l$ with $\hat{\mu}_k^l \le \tau$. First, we determine if the direction of updates for the two gradients are in the same direction by computing the cosine similarity between $\mathbf{g}_f$ and $\mathbf{g}_r$. If the cosine similarity is negative, it means that the gradients are not in the same direction, and we update the model parameters using $\mathbf{g}_r$. Otherwise, we modify $\mathbf{g}_r$ by removing the component that lies in the direction of $\mathbf{g}_f$, effectively projecting it onto the subspace orthogonal to $\mathbf{g}_f$. This is computed as:

$$\mathbf{g}_r = \mathbf{g}_r - \frac{\mathbf{g}_r \cdot \mathbf{g}_f}{\|\mathbf{g}_f\|^2} \mathbf{g}_f \tag{5}$$

The orthogonal gradients are then used to update the model parameters that were not updated in the forget step, i.e., with $\hat{\mu}_k^l \leq \tau$. We update the parameters using Adam optimization (Kingma & Ba, 2014), though these orthogonal gradients can be used with any gradient-based optimization algorithm.

# 4 EXPERIMENTS: SETTING

We evaluate SPARC on widely used datasets: CIFAR-10, CIFAR-100 (Krizhevsky et al., 2009), Lacuna-10 (Golatkar et al., 2020), and AgeDB (Moschoglou et al., 2017), covering coarse- and fine-grained classification as well as regression tasks. To ensure architectural diversity, we include All-CNN (Springenberg et al., 2014) a fully convolutional model without dense layers; ResNet-18 (He et al., 2016) which incorporates residual connections and fully connected classification layers; and Vision Transformer (ViT) (Kolesnikov et al., 2021), a modern attention-based architecture. This combination spans convolution-only, residual CNN, and transformer paradigms, allowing us to evaluate unlearning across both classical and modern model families. All results are averaged over five random seeds, with error bars in figures denoting the standard error.

## 4.1 BASELINES

We compare our approach against the state-of-the-art approximate unlearning algorithms and widely used baselines: **Naive Retraining (NR)** trains the model from scratch on the retain set; though impractical, it provides a reference for ideal unlearning. **Exact Unlearning-k (EUk)** and **Catastrophic Forgetting-k (CFk)** (Goel et al., 2022) freeze all but the last $k$ layers, either reinitializing (EUk) or directly fine-tuning (CFk) them. **SCRUB** (Kurmanji et al., 2023) employs a teacher-student min-max objective. **Selective Synaptic Dampening (SSD)** (Foster et al., 2024) updates model parameters selectively based on Fisher information. **Gaussian-Amnesiac learning (GAm)** (Tarun et al., 2023) modifies forget-set targets by Gaussian sampling and fine-tunes the model. **Blindspot unlearning (BS)** (Tarun et al., 2023) fine-tunes using guidance from a separately trained blindspot teacher model. A more detailed description of the baselines is given in the appendix.

## 4.2 HYPERPARAMETER TUNING

Hyperparameters were tuned using Optuna (Akiba et al., 2019) with cross-validation to ensure robustness across datasets. Each configuration was run multiple times, and the best settings were chosen by jointly balancing utility and unlearning success. Runs that collapsed to trivial performance (random guessing for classification or mean-regressor baselines for regression) were excluded. This procedure reflects how hyperparameters would be selected in practice, favoring configurations that yield reliable models rather than unstable or degenerate ones. The full search ranges and tuned values are reported in the appendix.

# 5 EXPERIMENTS: RESULTS

We now present and analyze the results from experiments. The focus is on assessing the algorithm's performance in terms of unlearning success for different forget set sizes compared to the baselines discussed in Section 4.1. We analyze our proposed algorithm both with and without recalibration, where SPA denotes just the forgetting step and SPARC includes recalibration with orthogonal gradient descent. For all settings, except label-only manipulations, the orthogonal gradient descent is run for one epoch (2% of naively retrained budget), whereas for label-only manipulations, it runs for nine epochs (10% of naive retraining budget). SSD does not involve training and only utilizes 1% of the naively retrained budget. All other baseline algorithms are run for the fixed unlearning budget of 10% of the naively retrained budget. This highlights SPARC's efficiency as compared to the baseline algorithms. The original model and the naively retrained model's results are also presented for comparison. The performance of the naively retrained model at 100% forget set size is the gold standard and gives a loose upper bound on the performance of unlearning algorithms. We only report the results for ResNet-18 model and CIFAR-10 dataset in the main paper, and present the complete results for all settings, models, and datasets in the appendix.

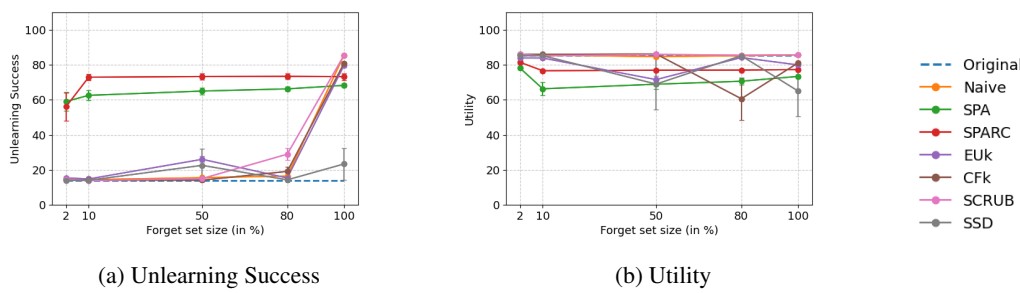

(a) Unlearning Success

(b) Utility

Figure 2: Label-and-Feature Manipulations: Unlearning success and Utility for different forget set sizes with ResNet-18 model and CIFAR-10 dataset

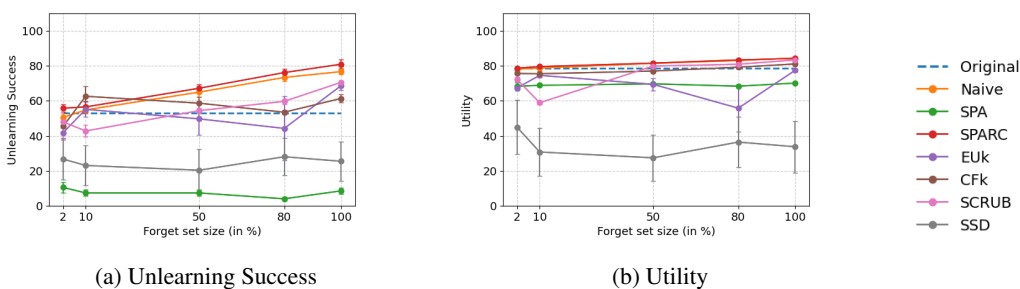

(a) Unlearning Success

(b) Utility

Figure 3: Label-Only Manipulations: Unlearning success and Utility for different forget set sizes with ResNet-18 model and CIFAR-10 dataset

**Label-and-Feature Manipulations**    Figure 2 shows that SPA and SPARC achieve the best overall unlearning performance and demonstrate robustness over even small forget set sizes. SSD failed to unlearn the effect of backdoors even with a $100\%$ forget set. Most baseline algorithms perform similarly to the Naively retrained model for the CIFAR-10 and CIFAR-100 datasets but do not perform as well for Lacuna-10 and have low utility. This is likely because the number of samples in Lacuna-10 is much less than the other datasets. SPARC is the most effective method across all datasets, achieving the best balance between high unlearning success and utility. In particular, the algorithm outperforms other when the forget set sizes are small, even is as low at $2\%$.

**Label-Only Manipulations**    Interclass confusion is the most challenging unlearning setting, as evidenced by the results where the baseline algorithm failed to improve on the original model's unlearning success, even at a 100% forget size, as shown in Figure 3. This difficulty arises because in interclass confusion, mere forgetting is insufficient to enhance unlearning success. Instead, it requires addressing the confusion between classes by effectively "relearning" or adjusting predictions for the confused classes.

In interclass confusion, unlearning success and utility are closely intertwined. As demonstrated by Goel et al. (2024), no small subset of parameters is disproportionately more important for the forget set compared to the retain set. This is because the two subsets are highly interconnected, making it harder to isolate and unlearn the target data. This contrasts with classification and regression unlearning, where the datasets are related, but the objective is distinct: to degrade performance on the forget subset while retaining high performance on the retain subset. In interclass confusion, the goal is to achieve good performance across both subsets, making the task fundamentally more complex.

SPARC is the only unlearning algorithm with unlearning performance similar to that of the Naively retrained model. This is because unlike the baseline algorithms, SPARC has a separate forgetting step which allows it to focus on forgetting the confusion before restoring utility.The low unlearning success of SPA demonstrates the need for the recalibration mechanism in this setting. Figure 3b shows SPARC performs the best even for utility.

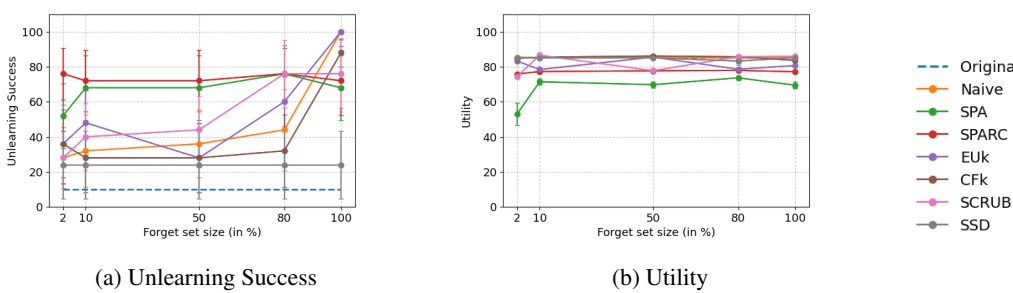

(a) Unlearning Success                    (b) Utility

Figure 4: Feature-Only Manipulations: Unlearning success and Utility for different forget set sizes with ResNet-18 model and CIFAR-10 dataset

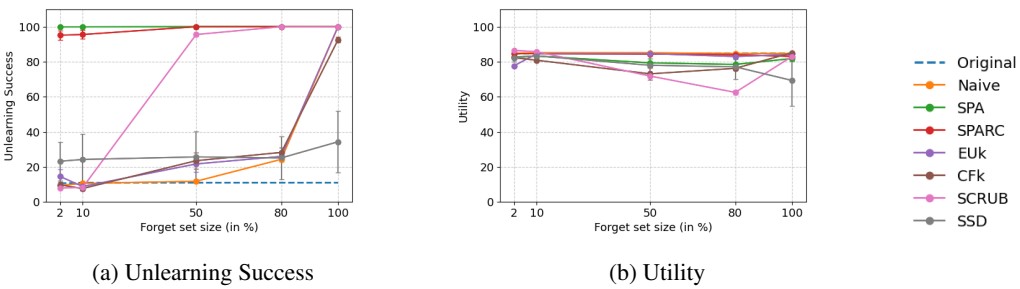

(a) Unlearning Success                    (b) Utility

Figure 5: Classification Unlearning: Unlearning success and Utility for different forget set sizes with ResNet-18 model and CIFAR-10 dataset

**Feature-Only Manipulations**    Figure 4 shows that SPARC is the most effective method with the highest average unlearning success and consistent performance across forget set sizes. Only EUk completely removes the influence of poison samples for 100% forget set, which is likely because it involves reinitializing layers in the neural network. SCRUB has similar performance to SPARC for forget set size $\geq 80\%$. SSD does not show any change in unlearning success over the original model.

**Classification Unlearning**    Figure 5 shows that both SPA and SPARC show superior unlearning success while maintaining competitive utility across varying forget set sizes. SSD has the poorest utility-unlearning tradeoff and suffers a drastic utility drop even with moderate unlearning success. EUk and CFk exhibit lower unlearning success for small forget set sizes but achieve effective unlearning at 100% forget set size. SCRUB struggles with datasets where the number of samples in a class is less but performs well for the CIFAR-10 dataset with a forget set size $\geq 50\%$. Overall, SPARC achieves the most effective unlearning while maintaining the original model's utility.

**Regression Unlearning**    Figure 6 shows that SPA achieves the strongest overall performance, consistently combining high unlearning success with preserved utility, acros all forget set sizes. In contrast, SPARC underperforms SPA in this regression setting: the recalibration step based on orthogonal gradient descent, effective in classification and manipulation tasks, proves less suited for continuous targets and can be overly conservative where prediction errors are distributed across a range rather than discrete classes. Blindspot achieves moderate unlearning but has the best utility. EUk and CFk perform reasonably well for forget range $\leq 30$ but indicate less effective forgetting in the other range. GaussianAmnesiac has the least favorable unlearning performance among all the algorithms.

### 5.1 SENSITIVITY ANALYSIS

We conducted a sensitivity analysis to explore the trade-off between unlearning success and utility for our proposed algorithm, focusing on the two critical hyperparameters of SPARC: the forgetting factor ($\gamma$) and the importance threshold ($\tau$). Figure 7 illustrates this relationship, where unlearning

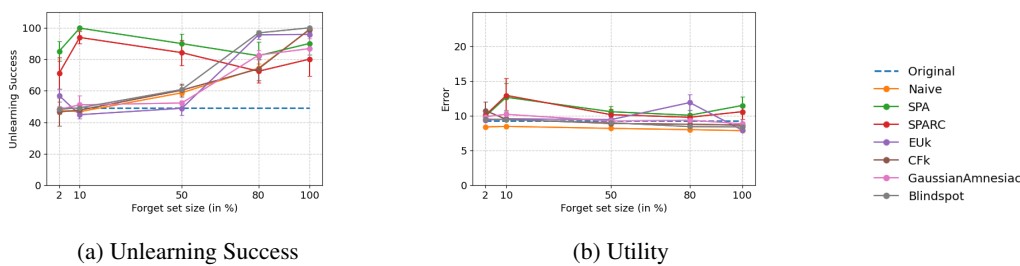

(a) Unlearning Success

(b) Utility

Figure 6: Regression Unlearning: Unlearning success and Utility for different forget set sizes with ResNet-18 model and AgeDB dataset for Forget Range: 60-101

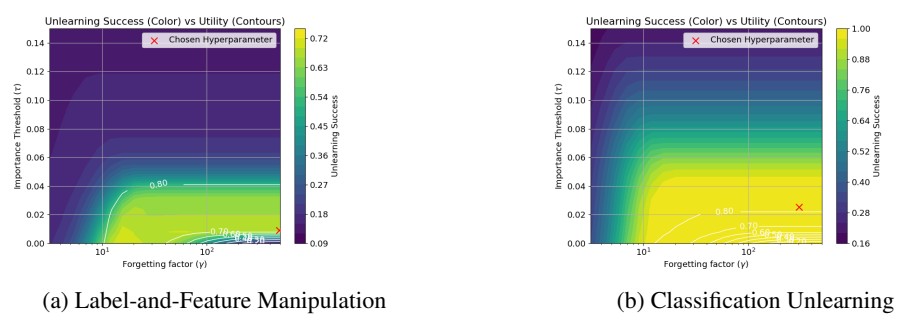

(a) Label-and-Feature Manipulation

(b) Classification Unlearning

Figure 7: Sensitivity Analysis on ResNet-18 model and CIFAR-10 dataset with 100% forget set size

success is represented by color intensity, and utility levels are indicated by contour lines. Additional results of sensitivity analysis are in the appendix.

The analysis reveals fairly broad regions of desirable outcomes, significantly easing a practitioner's task of selecting effective hyperparameters. Notably, our algorithm's design inherently supports adjusting the trade-off between unlearning success and utility by tuning these hyperparameters independently. These insights facilitate informed hyperparameter selection, allowing practitioners substantial flexibility to tailor the algorithm according to specific needs for unlearning efficacy and model performance.

## 6 DISCUSSION

Despite the promising results and improvements presented by SPARC, it has some limitations, and several avenues for future research remain open. First, exploring additional influence measures could further refine parameter identification, potentially enhancing SPARC's forgetting precision in diverse neural architectures and task settings. Second, all the experiments use vision datasets. How well the influence-based importance estimation and orthogonal recalibration translate to different modalities of data remains to be seen, however we anticipate similar good results. Lastly, the theoretical analysis of corrective unlearning remains an open area. Developing formal theoretical frameworks for machine unlearning could provide stronger assurances regarding data erasure compliance, security implications, and robustness against adversarial manipulations. Addressing these open challenges promises to drive further breakthroughs and deepen our understanding of machine unlearning and unlearning algorithms.

**Conclusion** This paper introduces Selective Parameter Adjustment and ReCalibration (SPARC), an efficient machine unlearning algorithm designed to selectively eliminate the influence of unwanted data samples from deep neural networks. Our extensive experiments highlight the effectiveness of our proposed two-phase approach, in particular in realistic settings where knowledge of the unlearn set of samples is limited. We have demonstrated the adaptability of SPARC across diverse tasks and manipulation settings, revealing its promise as a general, resource-efficient corrective unlearning framework.

## ETHICS STATEMENT

This work complies with the ICLR Code of Ethics. All experiments rely on publicly available datasets (CIFAR-10, CIFAR-100, Lacuna-10, AgeDB) that contain no personal or sensitive information. Our method is intended to improve safety, robustness, and compliance with data protection laws, however, a malevolent actor could misuse unlearning to remove the influence of data points for reasons unrelated to poisoning or privacy (e.g., to obscure evidence or censor information). We emphasize that responsible deployment requires safeguards such as transparent logging and auditing of unlearning events.

## REPRODUCIBILITY STATEMENT

We have taken several steps to ensure reproducibility of our results. All datasets used in this work (CIFAR-10, CIFAR-100, Lacuna-10, AgeDB) are publicly available. We provide detailed implementation of SPARC, SPA, and all baseline methods, together with training and evaluation scripts, in our code repository (to be released upon publication). Hyperparameter search procedures, ranges, and tuned values are reported in the appendix. All experiments were run with five random seeds and results are reported as mean $\pm$ standard error. We include exact instructions for dataset preprocessing, model initialization, and evaluation metrics, and specify software and hardware configurations to facilitate replication. Together, these details allow independent researchers to reproduce our experiments without requiring access to proprietary resources.

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

## A  Appendix: Baseline Details

- **Naive Retraining (NR):** Naive Retraining involves training the model from scratch on $D_r$ for the same number of epochs as the original model. It is considered the golden standard of unlearning. We use the same learning rate as initial model training and start training from the same initial model state.

- **Exact Unlearning the last $k$-layers (EUk)** Goel et al. (2022): EUk is a deep unlearning algorithm that reinitializes the last $k$ layers of the neural network $M_o$ and retrains while freezing all other layers to get $M_u$. We reinitialize the last $k$ layers of the neural network using the same initial values as the original training.

- **Catastrophically Forgetting the last $k$-layers (CFk)** Goel et al. (2022): CFk is similar to EUk, in which we train only the last $k$ layers but without reinitialization.

- **SCalable Remembering and Unlearning unBound (SCRUB)** Kurmanji et al. (2023): SCRUB is a teacher-student framework-based unlearning algorithm that involves optimizing the original model $M_o$ using a three-way min-max objective to get $M_u$. For each unlearning step, we first maximize the objective,

$$\frac{\beta}{|D_f|} \cdot \sum_{(x_f, y_f) \in D_f} D_{\mathrm{KL}}\left(M_o(x_f) \| M_u(x_f)\right)$$

and then minimize the objective,

$$\frac{\alpha}{|D_r|} \cdot \sum_{(x_r, y_r) \in D_r} D_{\mathrm{KL}}\left(M_o(x_r) \| M_u(x_r)\right) + \frac{\gamma}{|D_r|} \cdot \sum_{(x_r, y_r) \in D_r} l(M_u(x_r), y_r)$$

Here, $D_{\mathrm{KL}}(\cdot)$ is the Kullback–Leibler (KL) divergence, $l$ is the loss function for training, and $\alpha$, $\beta$ and $\gamma$ are hyperparameters.

- **Selective Synaptic Dampening (SSD)** Foster et al. (2024): In SSD, we selectively update the model parameters based on a decision threshold that depends on the diagonal of the Fisher information matrix. We first calculate the importance of each parameter for a dataset $D$ using the approximated value of the diagonal of the Fisher Information Matrix (FIM) as,

$$\mathcal{I}_D = \mathbb{E}_{(x,y) \in D}\left[\left(\frac{\partial ln(p(f(x;\theta)))}{\partial \theta}\right)\left(\frac{\partial ln(p(f(x;\theta)))}{\partial \theta}\right)^T |\theta\right]$$

Here, $\theta$ are the model parameters. Then, $\forall \theta_i \in \theta$ where $\mathcal{I}_{D_f,i} > \alpha.\mathcal{I}_{D,i}$, we update the model parameters using the equation,

$$\theta_i \leftarrow \min\left(\frac{\lambda.\mathcal{I}_{D,i}}{\mathcal{I}_{D_f,i}}, 1\right)\theta_i$$

- **Gaussian-Amnesiac learning (GAm)** Tarun et al. (2023): In GAm, we update the targets of $D_f$ by sampling from a Gaussian distribution and training the original model for a few epochs. Let $\mu$ and $\sigma^2$, be the mean and variance of the original targets. Then,

$$D'_f \leftarrow \{(x_f, y'_f)|y'_f \sim \mathcal{N}(\mu, \sigma^2) \forall (x_f, y_f) \in D_f\}$$
$$D' \leftarrow D_r \cup D'_f$$

The modified dataset $D'$ is used to fine-tune the original model.

- **Blindspot unlearning (BS)** Tarun et al. (2023): Blindspot involves training a blindspot model $M_b$ with parameters $\theta_b$ from scratch on $D_r$ for two epochs. This model is used in the objective function for fine-tuning. For samples $(x_r, y_r) \in D_r$, we update the model parameters using the objective,

$$l(f(x_r;\theta), y_r)$$

and for samples $(x_f, y_f) \in D_f$, we use the objective,

$$l(f(x_f;\theta), f(x_f;\theta_b)) + \lambda. \sum_j ||a_j^{\theta} - a_j^{\theta_b}||$$

Here, $a_j$ are the model activations, and $\lambda$ is a hyperparameter.

**Note:** SCRUB and SSD are classification unlearning algorithms and Gaussian-Amnesiac and Blindspot unlearning are regression unlearning algorithms.

---

**Algorithm 1** Selective Parameter Adjustment (SPA)

---

1: **Input:** model parameters $\theta$, forget set $D_f$, retain set $D_r$
2: **Parameters:** forget threshold $\tau$, forgetting factor $\gamma$

3: Compute influence measure $\mu$ using equation equation 2
4: Normalize influence measure to get $\hat{\mu}$ using equation equation 3
5: **for** each layer $l = 1, \ldots, L$ **do**
6:     **for** each parameter index $k = 1, \ldots, K$ **do**
7:         **if** $\hat{\mu}_k^l > \tau$ **then**
8:             Update $\theta_k^l \leftarrow \max\left(1 - \gamma . \hat{\mu}_k^l, 0\right) \theta_k^l$
9:         **end if**
10:     **end for**
11: **end for**

12: **return** $\theta$

---

---

**Algorithm 2** Selective Parameter Adjustment and ReCalibration (SPARC)

---

1: **Input:** model parameters $\theta$, forget set $D_f$, retain set $D_r$
2: **Parameters:** forget threshold $\tau$, forgetting factor $\gamma$, learning rate $\eta$, number of epochs $n$

3: ▷ *Call Algorithm 1: SPA*
4: $\theta \leftarrow \text{SPA}(\theta, D_f, D_r, \tau, \gamma)$

5: ▷ *Recalibration*
6: **for** each epoch $i = 1, \ldots, n$ **do**
7:     Compute gradient $\mathbf{g}_f$ for $D_f$ w.r.t parameters $\theta_k^l$ s.t. $\hat{\mu}_k^l \leq \tau$
8:     Compute gradient $\mathbf{g}_r$ for $D_r$ w.r.t parameters $\theta_k^l$ s.t. $\hat{\mu}_k^l \leq \tau$
9:     **if** $\mathbf{g}_r . \mathbf{g}_f > 0$ **then**
10:         Update $\mathbf{g}_r$ using equation 5
11:     **end if**
12:     Update $\theta$ with Adam update and learning rate $\eta$
13: **end for**

14: **return** $\theta$

---

# B APPENDIX: SPARC IMPLEMENTATION DETAILS

Due to space limitations in the main paper, we present the full algorithms for SPA (Algorithm 1) and SPARC (Algorithm 2) here.

## B.1 INFLUENCE MEASURE CALCULATION AND NORMALIZATION

The efficacy of both SPA and SPARC heavily relies on the accurate computation of the influence measure defined in equation equation 2. This measure quantifies the impact of individual training samples from the forget set on specific model parameters, relative to their impact from the retain set. As detailed in the main paper, we use the mean activation difference (equation equation 1) between the forget and retain sets to derive the influence measure. This involves a function $\phi : \mathbb{R} \to \mathbb{R}$, which for our experiments is implemented as the positive part function, i.e., $\phi(z) = \max(0, z)$. This choice helps avoid giving a high influence measure value to highly influential parameters for the retain set and, thus, has a negative mean activation difference value for both incoming and outgoing activation units. We can also have two different functions for incoming activation units $\mathcal{I}\left(\theta_k^l\right)$ and outgoing activation units $\mathcal{O}\left(\theta_k^l\right)$.

To ensure numerical stability and consistent comparisons across different layers and models, the calculated raw influence scores ($\mu_k^l$) are subsequently normalized. The normalization scheme em-

ployed, $\mathcal{N}(\cdot)$, is a global max scaling across all layers and parameters of the network such that

$$\hat{\mu}_k^l = \frac{\mu_k^l}{\max_{i,j}\left(\mu_i^j\right)}$$

This projects all influence score values into the range [0,1], allowing for a unified interpretation of the influence threshold $\tau$.

## C  APPENDIX: EXPERIMENT DETAILS

### C.1  DATASETS

In this study, we employ the following datasets to evaluate unlearning algorithms.

1. **CIFAR-10** (Krizhevsky et al., 2009)**:** Used for classification and manipulation unlearning settings, CIFAR-10 consists of 60,000 images (50,000 training and 10,000 testing) (32×32 pixels) evenly distributed across 10 classes. Its balanced class distribution and manageable complexity make it an ideal benchmark for evaluating various unlearning approaches.

2. **CIFAR-100** (Krizhevsky et al., 2009)**:** A more complex dataset than CIFAR-10, CIFAR-100 also contains 60,000 images (50,000 training and 10,000 testing) (32×32 pixels) but is divided into 100 classes, with only 500 samples per class in the training set. This dataset presents unique challenges for unlearning algorithms due to its higher class diversity and limited per-class data.

3. **Lacuna-10** (Golatkar et al., 2020)**:** Lacuna-10 consists of faces of 10 different celebrities from VGGFaces2 dataset Cao et al. (2018), with randomly sampled 500 images (32×32 pixels) each. The data is split into 400 samples for the training set and 100 samples for the test set for each class. The dataset has a very few number of samples compared to CIFAR-10 and CIFAR-100 and was especially created to evaluate unlearning algorithms.

4. **AgeDB** (Moschoglou et al., 2017)**:** Used for regression unlearning, AgeDB is a benchmark dataset for age estimation tasks. It consists of facial images (32x32 pixels) with corresponding age labels in the range 1-101, making it suitable for testing the performance of unlearning methods in regression settings. The dataset is split into 11,528 training samples and 4,960 test samples.

### C.2  MODELS

We employ the following deep learning models to evaluate unlearning algorithms.

1. **ResNet-18** (He et al., 2016)**:** A variant of the Residual Network architecture, ResNet-18 is a widely used convolutional neural network (CNN) known for its effectiveness in image classification tasks. Its key innovation lies in the use of residual connections, which help mitigate the vanishing gradient problem in deep networks, allowing for the training of significantly deeper models. We utilize ResNet-18 as a robust baseline due to its proven performance and relatively compact size.

2. **AllCNN** (Springenberg et al., 2014)**:** AllCNN or "All Convolutional Network" is a simpler CNN architecture that replaces pooling layers with convolutional layers with increased stride. This design choice aims to retain more spatial information throughout the network, which can be beneficial for certain image recognition tasks. AllCNN serves as a lightweight yet capable model to assess unlearning performance in scenarios where computational efficiency might be a factor.

3. **ViT-Tiny** (Kolesnikov et al., 2021)**:** Vision Transformer Tiny (ViT-Tiny) is a smaller version of the Vision Transformer model, which adapts the transformer architecture, originally developed for natural language processing, to image recognition tasks. Unlike CNNs that process images through convolutions, ViT divides images into patches and processes them as sequences, leveraging self-attention mechanisms to capture global dependencies. ViT-Tiny provides an opportunity to evaluate unlearning algorithms on a more recent and

conceptually different architecture, offering insights into its behavior and efficacy with transformer-based models.

### C.3 TRAINING DETAILS

The models are trained for 100 epochs using the Adam optimizer (Kingma & Ba, 2014). For ResNet-18 and AllCNN models, the initial learning rate was set to 0.001 for classification datasets and 0.01 for AgeDB, with a decay factor of 0.1 on the learning rate plateau. For ViT-Tiny, an initial learning rate of 0.001 with a cosine annealing schedule. ResNet-18 and AllCNN models are trained from scratch, and for ViT-Tiny, we start with a model pretrained on the ImageNet dataset.

### C.4 UNLEARNING EXPERIMENTS DETAILS

For label-and-feature manipulation experiments, a poisoning budget of 750 samples was used for CIFAR-10, and 75 samples each for CIFAR-100 and Lacuna-10. To measure the unlearning success, we insert the trigger pattern in all test set images and measure the fraction of manipulated test images that were classified correctly, i.e., the predicted label is the true unmanipulated label.

For label-only manipulation experiments, Interclass Confusion Test (ICT) was applied by manipulating $1/3$ of the samples from two randomly chosen classes in each of the CIFAR-10, CIFAR-100, and Lacuna-10 datasets. The manipulated classes were randomized in each experimental run to ensure diverse evaluation conditions. The goal of this setup is to systematically test the ability of unlearning algorithms to disentangle the representations of confused classes, thereby providing a robust evaluation of their efficacy.

In label-and-feature manipulations, a poisoning budget of 750 samples was used for CIFAR-10. This experiment was not conducted on the Lacuna-10 and CIFAR-100 datasets, nor with the ViT model, due to consistently observed low poisoning success in the trained models. This limitation is likely due to the small sample size of 500 per class in the other datasets. When 50% of these samples are used for the attack, the number of poison samples becomes insufficient to effectively implement gradient matching.

In classification unlearning experiments, forget class is the one with target label 9 for all the datasets, to ensure consistency in evaluation. In regression unlearning experiments, two distinct forget ranges were evaluated: ages less than 30 (comprising 2,434 samples) and ages greater than 60 (comprising 2,650 samples). The forget ranges represent distinct age groups with sufficient sample sizes, enabling a robust evaluation of unlearning performance across diverse demographic segments, and are consistent with the existing literature.

To maintain consistency with existing literature and ensure fair comparisons, all baseline unlearning algorithms, with the exception of SSD, were executed for ten unlearning epochs. This duration represents 10% of the full naive retraining and original training epochs. Due to its design, the SSD algorithm, which primarily involves a single forgetting step, was run for only one unlearning epoch.

Our proposed algorithm operates in two distinct phases: forgetting and recalibration. SPA, which encompasses solely the forgetting step, requires only one unlearning epoch. This epoch involves the computation of an influence measure, which primarily necessitates a forward pass through the model, followed by an update of the model parameters based on this measure. SPARC, which combines forgetting with a subsequent recalibration phase utilizing orthogonal gradient descent, consists of one forgetting epoch and $n$ recalibration epochs. For all experiments, except those pertaining to label-only manipulation unlearning and Vision Transformers, SPARC was run for a total of two unlearning epochs (one for forgetting and one for recalibration). In the case of label-only manipulation and Vision Transformers, SPARC was extended to ten unlearning epochs (one for forgetting and nine for recalibration). A key efficiency advantage of SPARC lies in its orthogonal gradient descent update, which selectively modifies model parameters whose influence measure falls below a hyperparameterized threshold. This targeted update mechanism contributes to SPARC's high efficiency, requiring fewer unlearning epochs and making each epoch computationally less demanding compared to other baselines.

For sensitivity analysis of SPA, we run $20 \times 20$ combinations of the hyperparameters forgetting factor ($\gamma$) and the importance threshold ($\tau$). We plot the results in a contour plot where contour

lines represent the utility performance and contour colors represent the unlearning success. The contour plot helps to analyze the sensitivity of hyperparameters and the tradeoff between utility and unlearning success.

## C.5 HYPERPARAMETER TUNING

In this study, hyperparameter tuning was conducted to optimize the performance of unlearning algorithms across various datasets and experimental settings. We employed Optuna (Akiba et al., 2019), an efficient and flexible hyperparameter optimization framework, to achieve this. The tuning process utilized 5-fold cross-validation to ensure the robustness and generalization of the selected hyperparameters. Each algorithm was evaluated over 10 independent runs, and the optimal hyperparameters were selected based on a weighted average that balanced utility and unlearning performance. For the experiments involving ViT-Tiny we only run five hyperparameter trials because of the high computation requirements.

To ensure the relevance of our results, runs exhibiting utility performance comparable or inferior to random predictions (for classification tasks) or to a mean regressor baseline (for regression tasks) were excluded from consideration. To equally weigh utility and unlearning performance in regression tasks, we normalized the utility metric, mean absolute error (MAE), to a $[0, 1]$ range by assigning a scaled value of 1 to an MAE of 0 (perfect prediction) and a scaled value of 0 to an MAE equal to that of a mean regressor (baseline prediction).

### C.5.1 HYPERPARAMETER RANGES

The following ranges of hyperparameters were explored for each unlearning algorithm during the tuning process:

1. SPARC: $\tau : [0, 1], \gamma : [10^{-3}, 500],$ learning rate $: [10^{-5}, 1]$

2. CFk: $k : [1, \text{total layers in network}],$ learning rate $: [10^{-5}, 1]$

3. EUk: $k : [1, \text{total layers in network}],$ learning rate $: [10^{-5}, 1]$

4. SCRUB: $\alpha : [10^{-3}, 10], \beta : [10^{-3}, 10], \gamma : [10^{-3}, 10],$ learning rate $: [10^{-5}, 1]$

5. SSD: $\alpha : [0.1, 100], \lambda : [0.01, 5]$

6. GaussianAmnesiac: learning rate $: [10^{-7}, 1]$

7. Blindspot: $\lambda : [1, 100],$ learning rate $: [10^{-7}, 1]$

## C.6 COMPUTATIONAL RESOURCES

All experiments were conducted on a dedicated server to ensure consistency and reproducibility of results.. The server is configured with the following specifications:

- **CPU:** Intel(R) Xeon(R) Gold 6326 CPU @ 2.90GHz
- **GPU:** 4 x Nvidia A10 (24 GB)
- **RAM:** 256 GB

We isolate each experiment to a single GPU device, enabling the concurrent execution of four independent experiments on the server. The typical execution time for one unlearning epoch varied depending on the dataset, model, and unlearning setting. For experiments involving ResNet-18 and CIFAR-10 dataset an unlearning epoch takes around 2 minutes, while for ViT-Tiny on CIFAR-10 it takes around 10 minutes.

## C.7 COMPUTATIONAL COST ANALYSIS

We analyze the computational cost of unlearning algorithms by estimating the floating-point operations (FLOPs) required for each method. Our analysis is based on the following assumptions and measurement methodologies:

- **Forward Pass**: Directly measured by instrumenting the model architecture with hooks that count operations in each layer (Conv2d, Linear, BatchNorm2d, MultiheadAttention).

- **Backward Pass**: Estimated as $2\times$ forward pass FLOPs, following standard automatic differentiation complexity **?**.

- **Optimizer Operations**: For Adam optimizer, we estimate 5 operations per parameter.

- **SPARC-Specific Operations**:

  - Activation processing: 3 operations per parameter (sum, division, subtraction)
  - Utility calculation: Estimated as 5 operations per parameter based on layer-specific computations
  - Weight multiplication and normalization: 4 operations per parameter
  - Gradient projection (recalibration): 5 operations per parameter (dot product, norm, division, subtraction, multiplication)

Table 2 presents the computational cost of the unlearning algorithms measured in tera floating-point operations (TFLOPs) for AllCNN model and CIFAR-10 dataset.

Table 2: Computational cost comparison of unlearning algorithms on CIFAR-10 with AllCNN architecture. All methods except SPARC run for 10 epochs; SPARC runs for 1-2 epochs.

| Method | Total FLOPs (TFLOPs) |
|---|---|
| SPA | 34.79 |
| SPARC | 138.89 |
| CFk (k=1) | 1,019.68 |
| EUk (k=1) | 1,019.68 |
| SCRUB | 1,717.21 |
| SSD | 104.10 |
| Gaussian Amnesiac | 1,040.97 |
| Blindspot | 1,626.60 |
| Naive Retraining | 10,196.76 |

## D   APPENDIX: ADDITIONAL RESULTS FOR LABEL-AND-FEATURE MANIPULATIONS UNLEARNING

In this section, we provide additional results for the AllCNN model and the Vision Transformer model. For the Vision Transformer, we only evaluate it for a subset of baseline algorithms because of its high computational requirements.

The performance of unlearning algorithms on label-and-feature manipulations is detailed in Figures 8, 9, 10, 11, and 12. SPA and SPARC consistently show robust unlearning success across all datasets, with SPARC striking an optimal balance between high unlearning success and utility. EUk performs well for the Lacuna-10 dataset but shows limited capability in the other datasets. CFk, SCRUB, and SSD do not show signs of unlearning for smaller forget set sizes. Also, we can note from Figure 13 that SPARC shows competitive performance for Vision Transformer network. SSD also has a high unlearning success for ViT model.

## E   ADDITIONAL RESULTS FOR LABEL-ONLY MANIPULATIONS UNLEARNING

We present the additional results for label-only manipulations unlearning in this section. Results given in Figures 14, 15, 16, 17, and 18 show that SPARC performs similar to the naively retrained model. EUk and SCRUB perform well for the CIFAR-10 dataset but not for other datasets. SSD performs the worst across all datasets and forget set sizes.

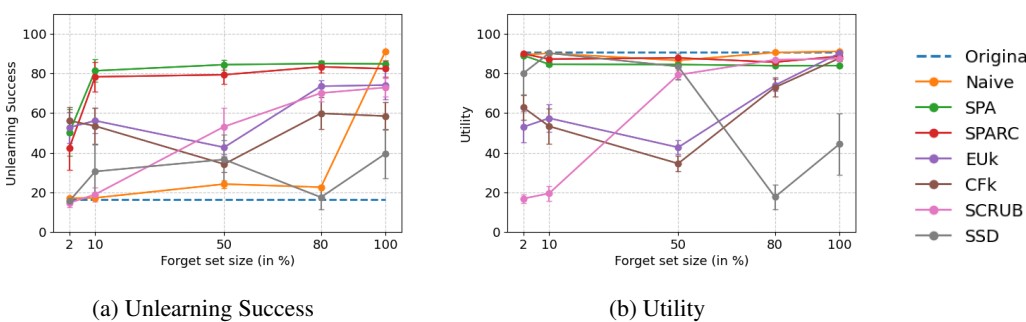

(a) Unlearning Success        (b) Utility

Figure 8: Label-and-Feature Manipulations: Unlearning success and Utility for different forget set sizes with ResNet-18 model and Lacuna-10 dataset

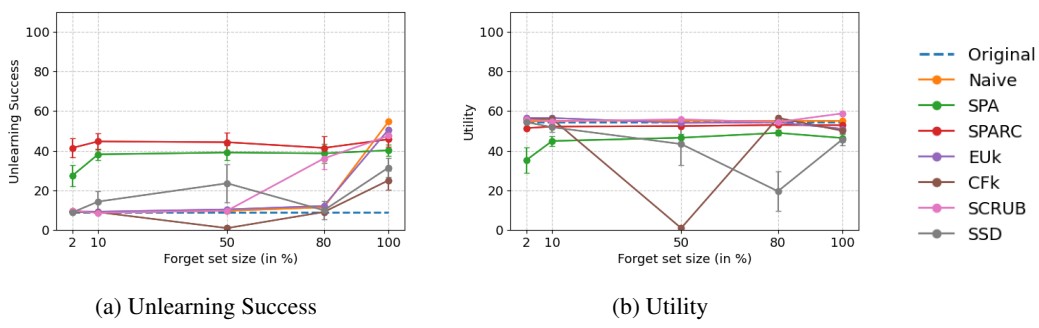

(a) Unlearning Success        (b) Utility

Figure 9: Label-and-Feature Manipulations: Unlearning success and Utility for different forget set sizes with ResNet-18 model and CIFAR-100 dataset

## F ADDITIONAL RESULTS FOR FEATURE-ONLY MANIPULATIONS UNLEARNING

The additional results for feature-only manipulations are presented in this section. We conducted the gradient matching attack only on the CIFAR-10 dataset and ResNet-18 and AllCNN models, as attempts to optimize the hyperparameters for other settings resulted in only marginal poisoning success.

Figures 4 and 19 show that SPA and SPARC outperform other algorithms significantly, with SPA achieving the highest unlearning success and SPARC providing better utility. Among the baselines, EUk achieves the best unlearning success, whereas SSD performs worse than the original model. CFk and SCRUB achieve limited improvements over the original model.

## G ADDITIONAL RESULTS FOR CLASSIFICATION UNLEARNING

We show the full results for classification unlearning in this section. Figures 20, 21, 22, 23, and 24 illustrate that SPA and SPARC demonstrate excellent unlearning success across datasets and forget set sizes, with SPARC showing superior utility retention compared to SPA. SSD achieves high unlearning on CIFAR-10 but suffers from reduced utility compared to all other algorithms. EUk, CFk, and SCRUB exhibit relatively moderate unlearning success and generally maintain good utility. The results for Vision Transformer in Figure 25 are similar to the label-and-feature manipulation unlearning as SPARC and SSD both show high unlearning success, although there is a significant decrease in utility for all unlearning algorithms.

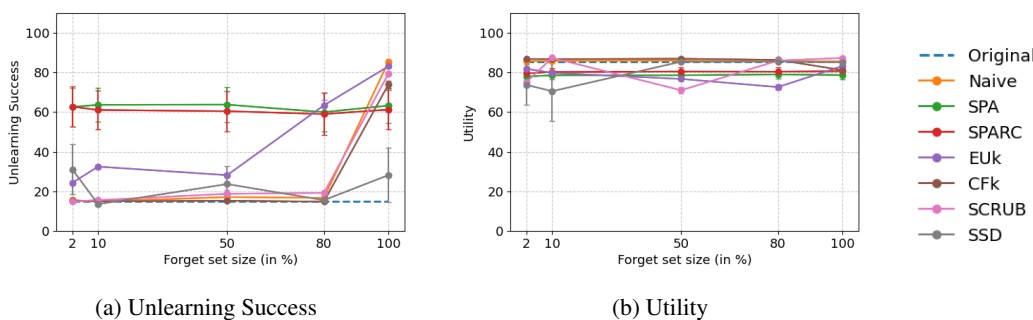

(a) Unlearning Success          (b) Utility

Figure 10: Label-and-Feature Manipulations: Unlearning success and Utility for different forget set sizes with AllCNN model and CIFAR-10 dataset

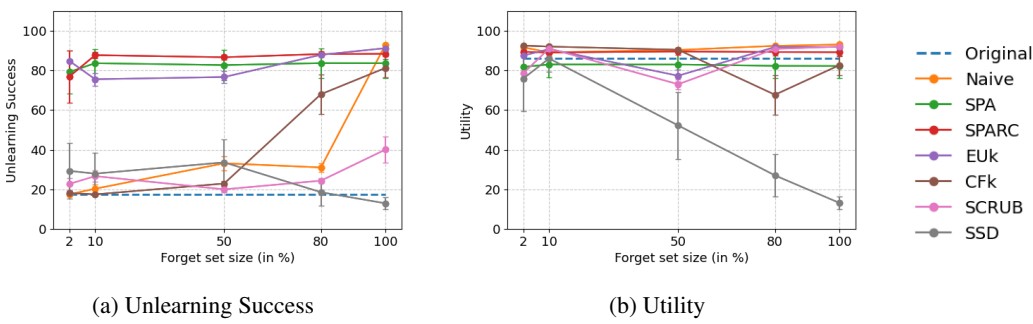

(a) Unlearning Success          (b) Utility

Figure 11: Label-and-Feature Manipulations: Unlearning success and Utility for different forget set sizes with AllCNN model and Lacuna-10 dataset

## H   ADDITIONAL RESULTS FOR REGRESSION UNLEARNING

This section presents the additional results for the regression unlearning setting. Regression unlearning performance is given in Figures 26, 27, and 28. Both SPA and SPARC algorithms achieve superior unlearning success, particularly for the forget range of 60-101, while demonstrating competitive utility. GaussianAmnesiac achieves high unlearning success in the 0-30 range but less in the older range. Whereas, Blindspot does not exhibit any unlearning in the range 0-30 but outperforms other baselines in the range 60-101.

## I   ADDITIONAL SENSITIVITY ANALYSIS RESULTS

We present the full results for the sensitivity analysis of label-and-feature manipulation unlearning, classification unlearning, and regression unlearning in Figures 29, 30, and 31. The red cross marks the hyperparameter chosen for the final results presented in the paper. From all the figures, we can conclude that regions of high utility and unlearning success are broad, which makes it easier to select the effective hyperparameters. The figures also show the tradeoff boundaries, allowing model owners to flexibly modify the model to achieve desired utility and unlearning success.

## LLM USAGE

Large Language Models (LLMs) such as Grammarly and ChatGPT were used only for improving grammar, writing clarity, and style. They were not used for generating ideas, designing algorithms, conducting experiments, analyzing results, or writing technical content. All technical contributions, experiments, and analyses are entirely the work of the authors.

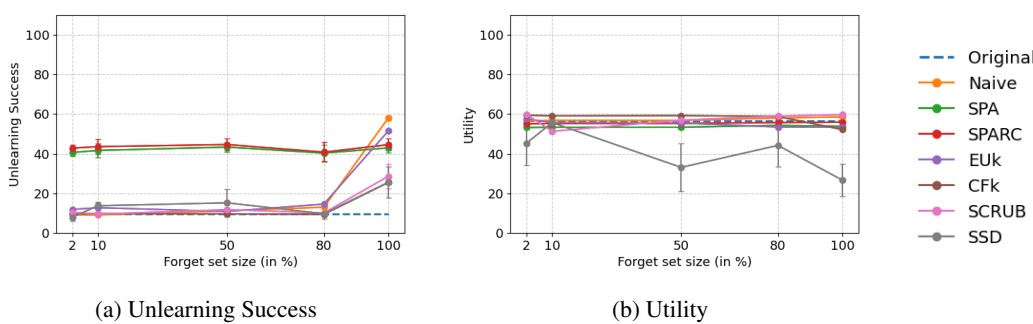

(a) Unlearning Success

(b) Utility

Figure 12: Label-and-Feature Manipulations: Unlearning success and Utility for different forget set sizes with AllCNN model and CIFAR-100 dataset

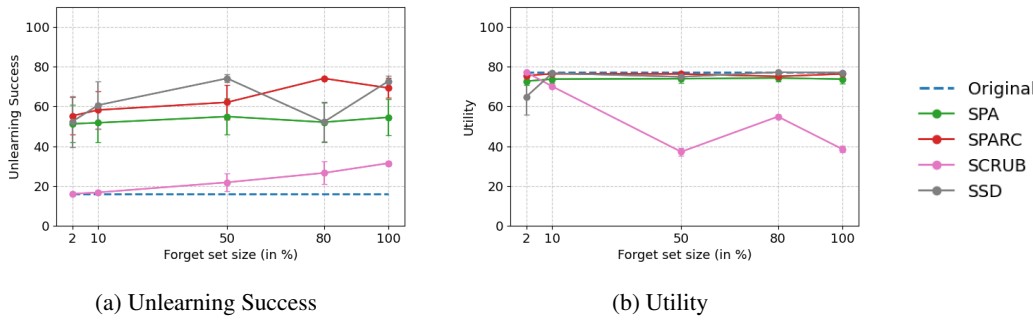

(a) Unlearning Success

(b) Utility

Figure 13: Label-and-Feature Manipulations: Unlearning success and Utility for different forget set sizes with ViT model and CIFAR-10 dataset

This appendix presents comprehensive experimental results for various machine unlearning approaches across different scenarios and datasets.

## I.1 LABEL-AND-FEATURE MANIPULATIONS (BADNET)

This section presents results for BadNet attacks, where both labels and features are manipulated through backdoor poisoning.

### I.1.1 RESNET-18 ON LACUNA-10

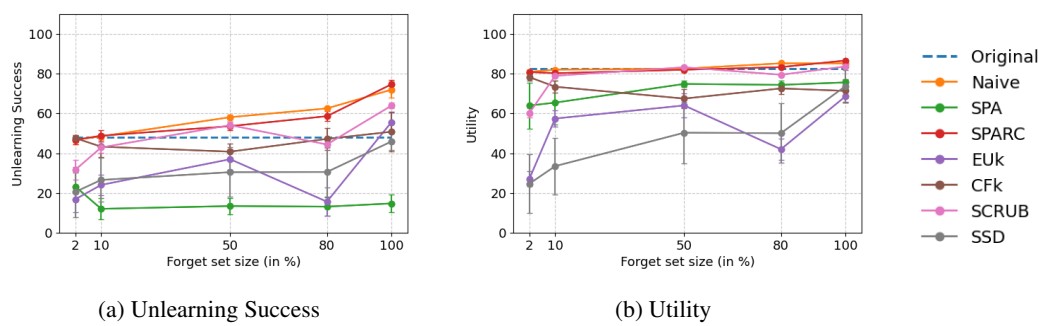

(a) Unlearning Success

(b) Utility

Figure 14: Label-Only Manipulations: Unlearning success and Utility for different forget set sizes with ResNet-18 model and Lacuna-10 dataset

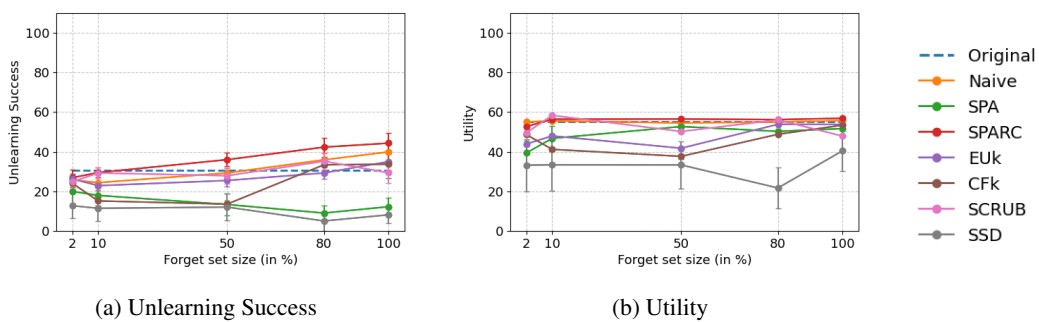

(a) Unlearning Success

(b) Utility

Figure 15: Label-Only Manipulations: Unlearning success and Utility for different forget set sizes with ResNet-18 model and CIFAR-100 dataset

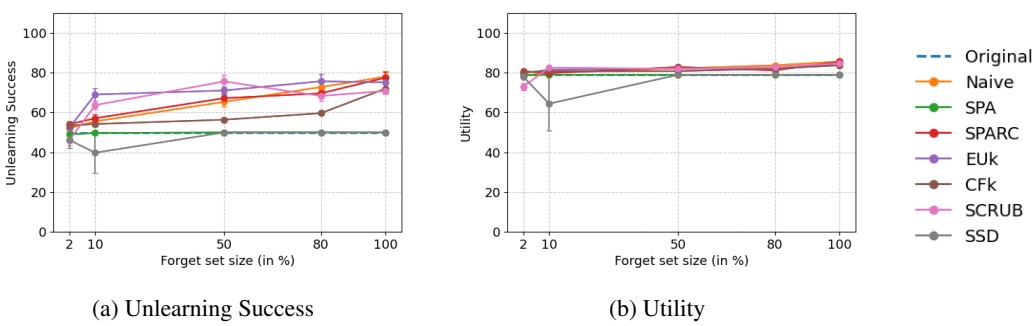

(a) Unlearning Success

(b) Utility

Figure 16: Label-Only Manipulations: Unlearning success and Utility for different forget set sizes with AllCNN model and CIFAR-10 dataset

| Unlearner | Utility | Unlearning Success |
|-----------|---------|--------------------|
| Original | $90.680 \pm 0.240$ | $16.460 \pm 0.503$ |
| Naive | $89.375 \pm 0.144$ | $17.100 \pm 0.745$ |
| SPA | $89.020 \pm 0.479$ | $50.200 \pm 11.813$ |
| SPARC | $89.840 \pm 0.229$ | $42.360 \pm 11.049$ |
| EUk | $52.960 \pm 7.929$ | $52.740 \pm 7.854$ |
| CFk | $62.800 \pm 6.236$ | $56.160 \pm 6.918$ |
| SCRUB | $16.860 \pm 2.213$ | $14.680 \pm 2.211$ |
| SSD | $80.000 \pm 10.752$ | $15.600 \pm 0.794$ |

Table 3: BadNet results for ResNet-18 on Lacuna-10 with 2.0% forget size

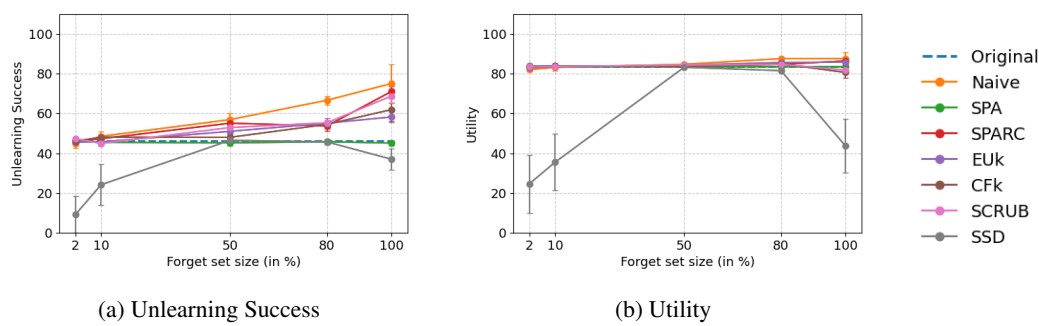

Figure 17: Label-Only Manipulations: Unlearning success and Utility for different forget set sizes with AllCNN model and Lacuna-10 dataset

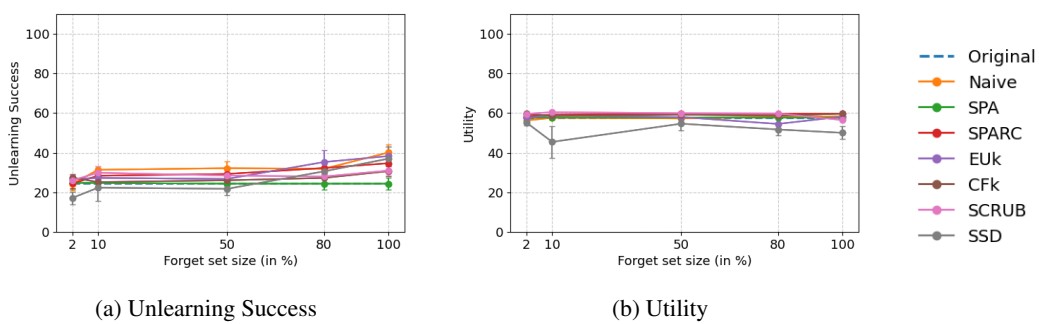

Figure 18: Label-Only Manipulations: Unlearning success and Utility for different forget set sizes with AllCNN model and CIFAR-100 dataset

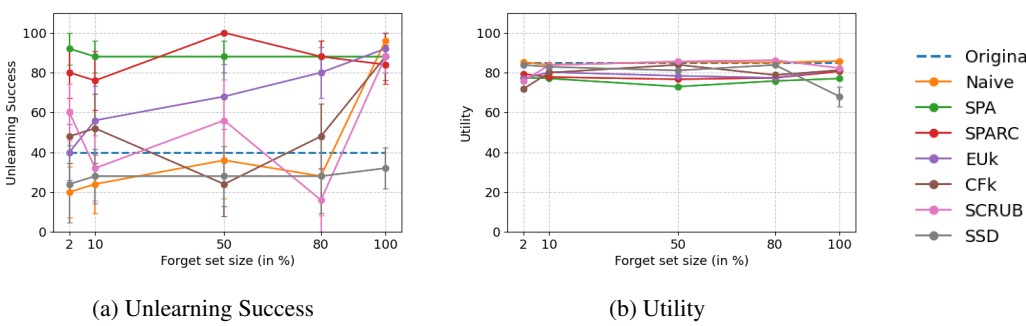

Figure 19: Feature-Only Manipulations: Unlearning success and Utility for different forget set sizes with AllCNN model and CIFAR-10 dataset

| Unlearner | Utility | Unlearning Success |
|-----------|---------|--------------------|
| Original | $90.680 \pm 0.240$ | $16.460 \pm 0.503$ |
| Naive | $90.075 \pm 0.431$ | $17.250 \pm 0.794$ |
| SPA | $84.560 \pm 0.947$ | $81.280 \pm 5.684$ |
| SPARC | $87.160 \pm 0.679$ | $78.240 \pm 7.453$ |
| EUk | $57.400 \pm 6.788$ | $56.140 \pm 6.362$ |
| CFk | $53.320 \pm 9.002$ | $53.440 \pm 9.203$ |
| SCRUB | $19.520 \pm 3.734$ | $18.980 \pm 3.579$ |
| SSD | $90.160 \pm 0.588$ | $30.540 \pm 13.900$ |

Table 4: BadNet results for ResNet-18 on Lacuna-10 with 10.0% forget size

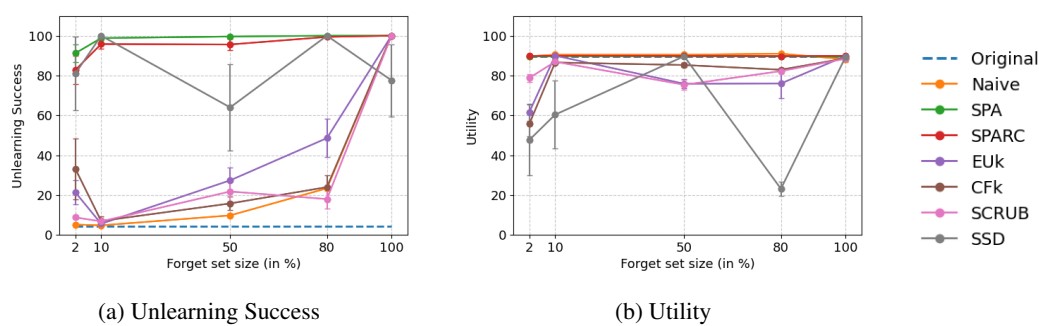

(a) Unlearning Success

(b) Utility

Figure 20: Classification Unlearning: Unlearning success and Utility for different forget set sizes with ResNet-18 model and Lacuna-10 dataset

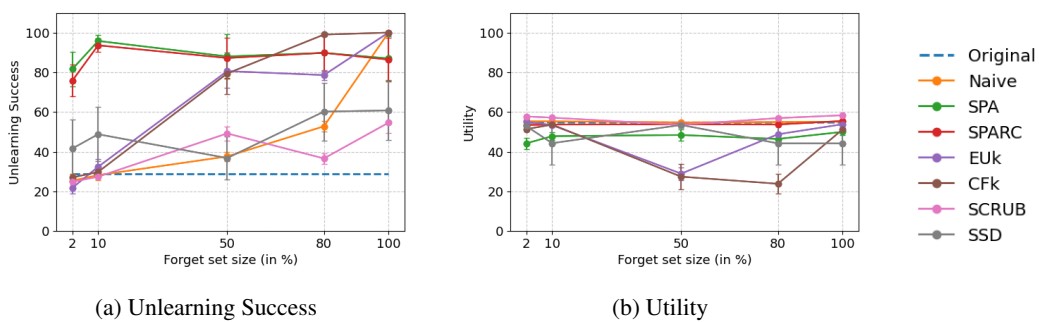

(a) Unlearning Success

(b) Utility

Figure 21: Classification Unlearning: Unlearning success and Utility for different forget set sizes with ResNet-18 model and CIFAR-100 dataset

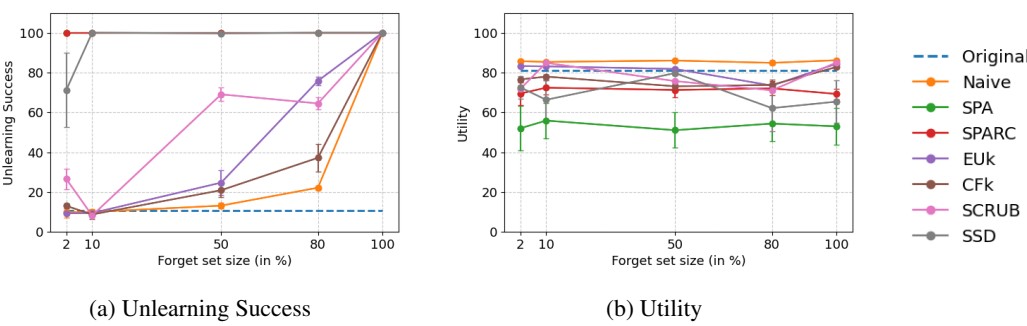

(a) Unlearning Success

(b) Utility

Figure 22: Classification Unlearning: Unlearning success and Utility for different forget set sizes with AllCNN model and CIFAR-10 dataset

| Unlearner | Utility | Unlearning Success |
|---|---|---|
| Original | $90.680 \pm 0.240$ | $16.460 \pm 0.503$ |
| Naive | $86.475 \pm 0.913$ | $24.200 \pm 1.978$ |
| SPA | $84.420 \pm 0.733$ | $84.360 \pm 2.424$ |
| SPARC | $87.780 \pm 0.599$ | $79.280 \pm 4.661$ |
| EUk | $42.680 \pm 3.601$ | $42.700 \pm 3.650$ |
| CFk | $34.540 \pm 3.982$ | $34.240 \pm 3.982$ |
| SCRUB | $79.200 \pm 2.211$ | $53.140 \pm 9.561$ |
| SSD | $83.440 \pm 6.615$ | $36.600 \pm 12.454$ |

Table 5: BadNet results for ResNet-18 on Lacuna-10 with 50.0% forget size

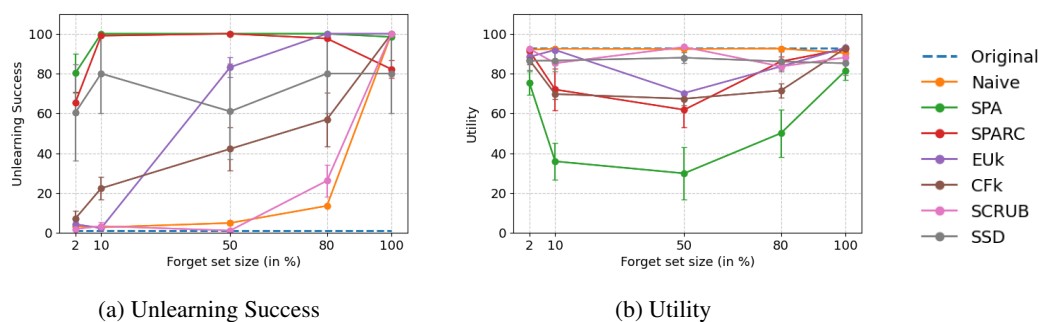

(a) Unlearning Success        (b) Utility

Figure 23: Classification Unlearning: Unlearning success and Utility for different forget set sizes with AllCNN model and Lacuna-10 dataset

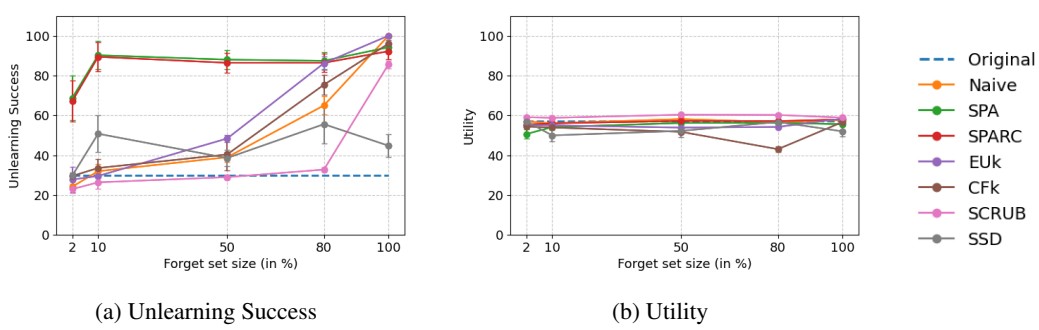

(a) Unlearning Success        (b) Utility

Figure 24: Classification Unlearning: Unlearning success and Utility for different forget set sizes with AllCNN model and CIFAR-100 dataset

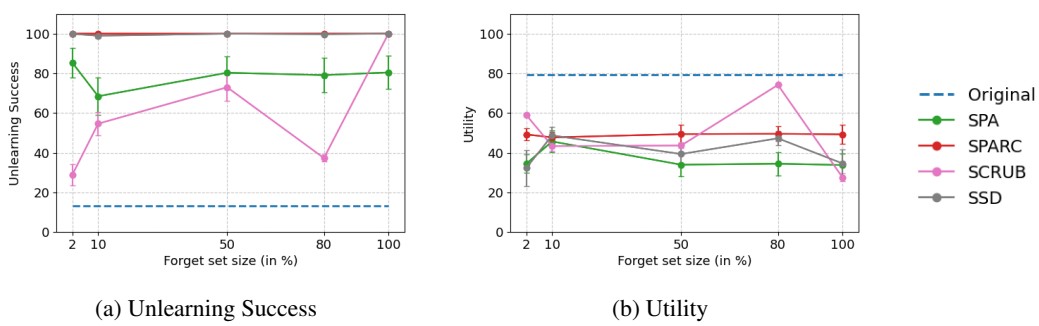

(a) Unlearning Success        (b) Utility

Figure 25: Classification Unlearning: Unlearning success and Utility for different forget set sizes with ViT model and CIFAR-10 dataset

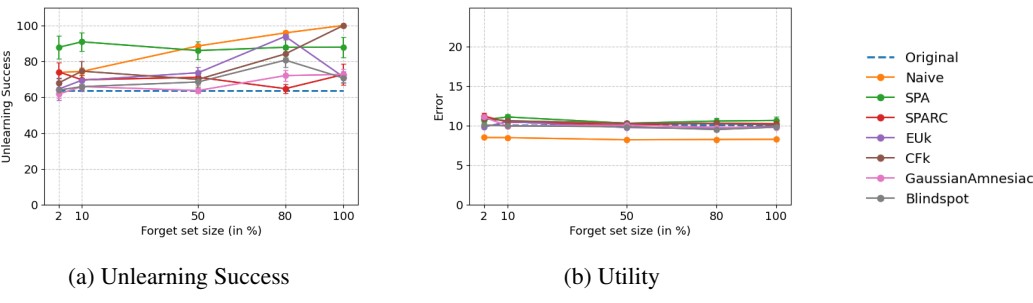

(a) Unlearning Success        (b) Utility

Figure 26: Regression Unlearning: Unlearning success and Utility for different forget set sizes with ResNet-18 model and AgeDB dataset for Forget Range: 0-30

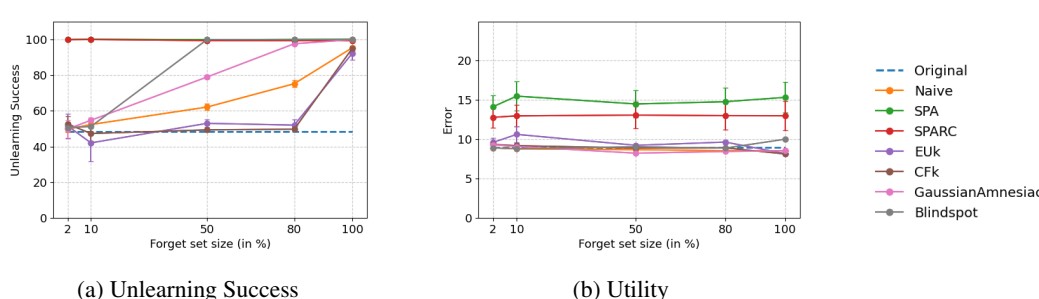

(a) Unlearning Success                    (b) Utility

Figure 27: Regression Unlearning: Unlearning success and Utility for different forget set sizes with AllCNN model and AgeDB dataset for Forget Range: 60-101

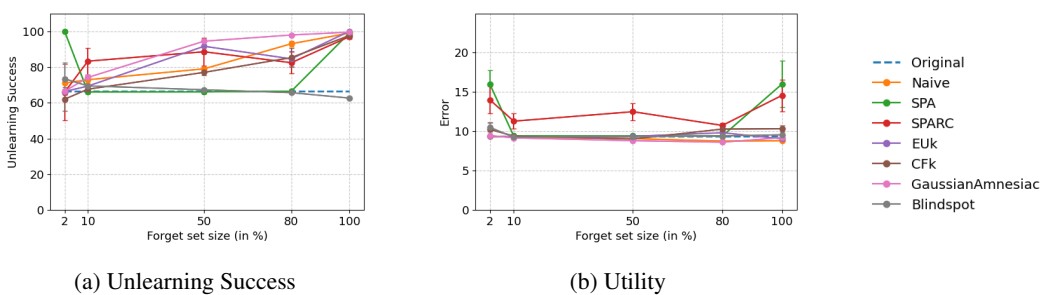

(a) Unlearning Success                    (b) Utility

Figure 28: Regression Unlearning: Unlearning success and Utility for different forget set sizes with AllCNN model and AgeDB dataset for Forget Range: 0-30

| Unlearner | Utility | Unlearning Success |
|---|---|---|
| Original | $90.680 \pm 0.240$ | $16.460 \pm 0.503$ |
| Naive | $90.500 \pm 0.640$ | $22.625 \pm 0.978$ |
| SPA | $83.820 \pm 0.640$ | $84.920 \pm 1.129$ |
| SPARC | $85.600 \pm 0.682$ | $83.280 \pm 2.987$ |
| EUk | $73.960 \pm 3.018$ | $73.500 \pm 2.889$ |
| CFk | $72.920 \pm 4.793$ | $59.820 \pm 7.928$ |
| SCRUB | $86.820 \pm 0.545$ | $70.060 \pm 4.148$ |
| SSD | $17.680 \pm 6.205$ | $17.540 \pm 6.047$ |

Table 6: BadNet results for ResNet-18 on Lacuna-10 with 80.0% forget size

| Unlearner | Utility | Unlearning Success |
|---|---|---|
| Original | $90.680 \pm 0.240$ | $16.460 \pm 0.503$ |
| Naive | $91.050 \pm 0.126$ | $91.050 \pm 0.250$ |
| SPA | $83.880 \pm 0.634$ | $84.760 \pm 1.455$ |
| SPARC | $88.620 \pm 0.380$ | $82.300 \pm 3.947$ |
| EUk | $90.120 \pm 0.233$ | $74.080 \pm 7.358$ |
| CFk | $87.600 \pm 1.399$ | $58.460 \pm 7.019$ |
| SCRUB | $87.420 \pm 0.585$ | $72.860 \pm 4.662$ |
| SSD | $44.360 \pm 15.449$ | $39.520 \pm 12.443$ |

Table 7: BadNet results for ResNet-18 on Lacuna-10 with 100.0% forget size

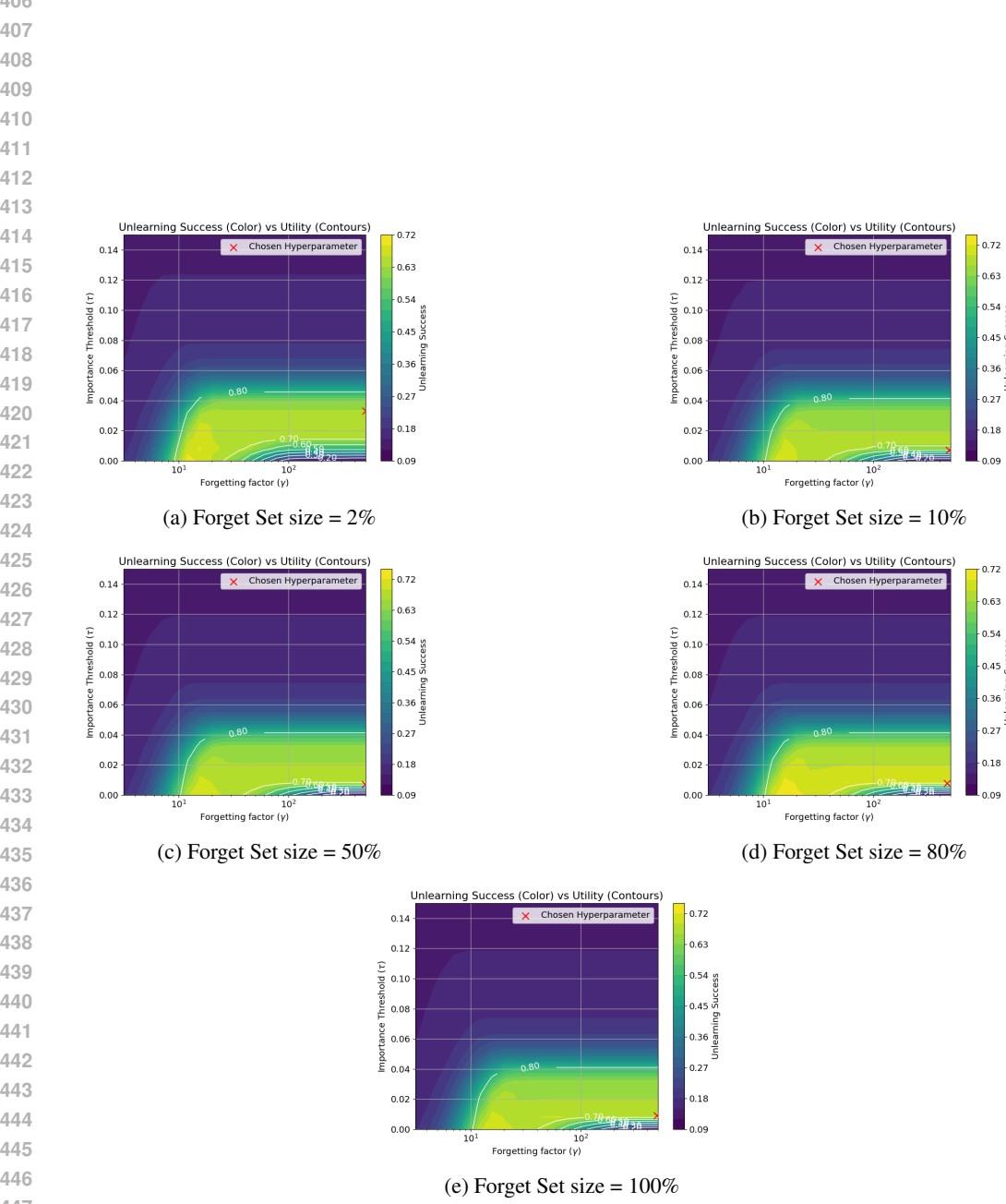

(a) Forget Set size = 2%

(b) Forget Set size = 10%

(c) Forget Set size = 50%

(d) Forget Set size = 80%

(e) Forget Set size = 100%

Figure 29: Sensitivity Analysis of SPA on ResNet-18 model and CIFAR-10 dataset for Label-and-Feature Manipulation Unlearning

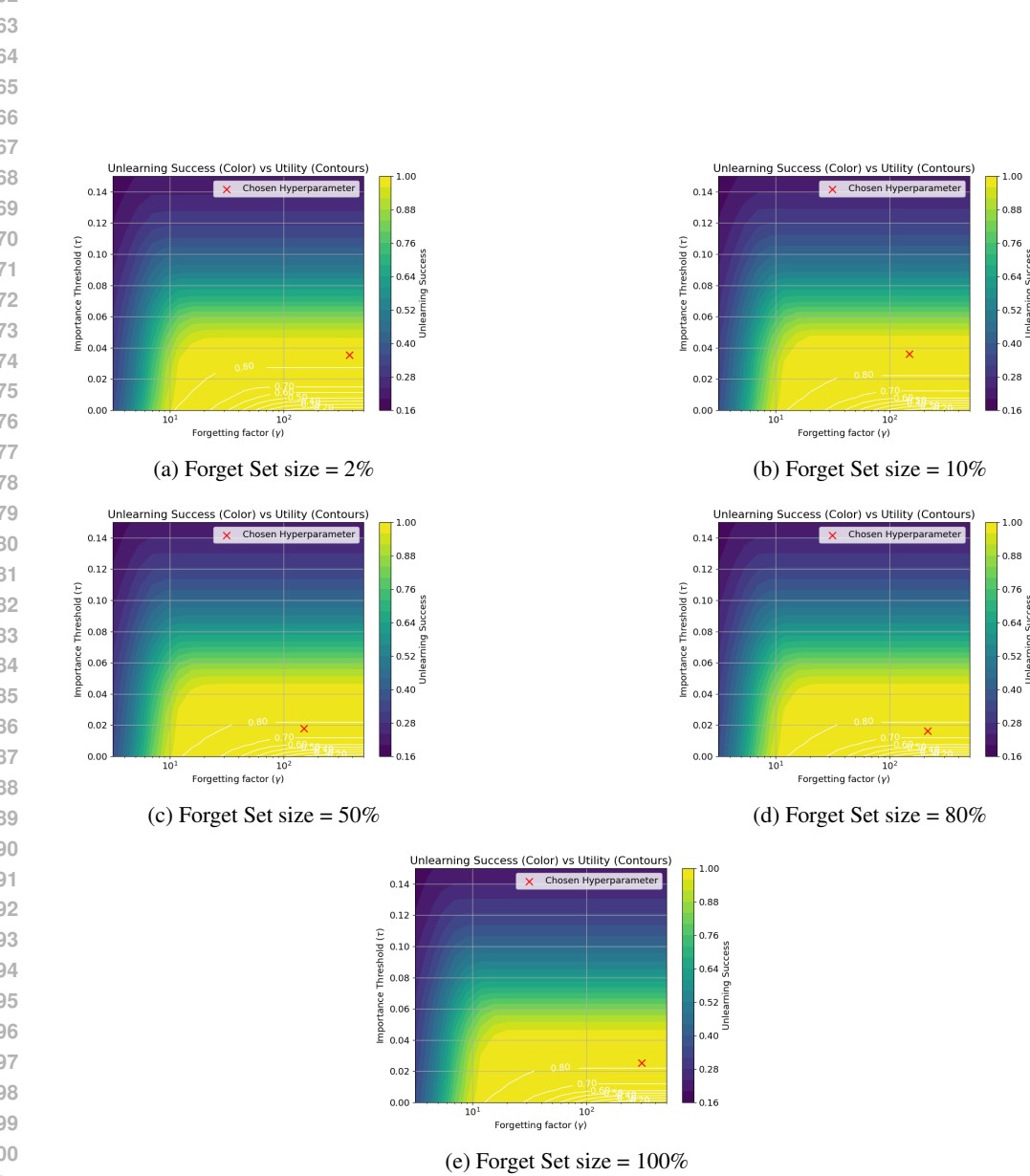

Figure 30: Sensitivity Analysis of SPA on ResNet-18 model and CIFAR-10 dataset for Classification Unlearning

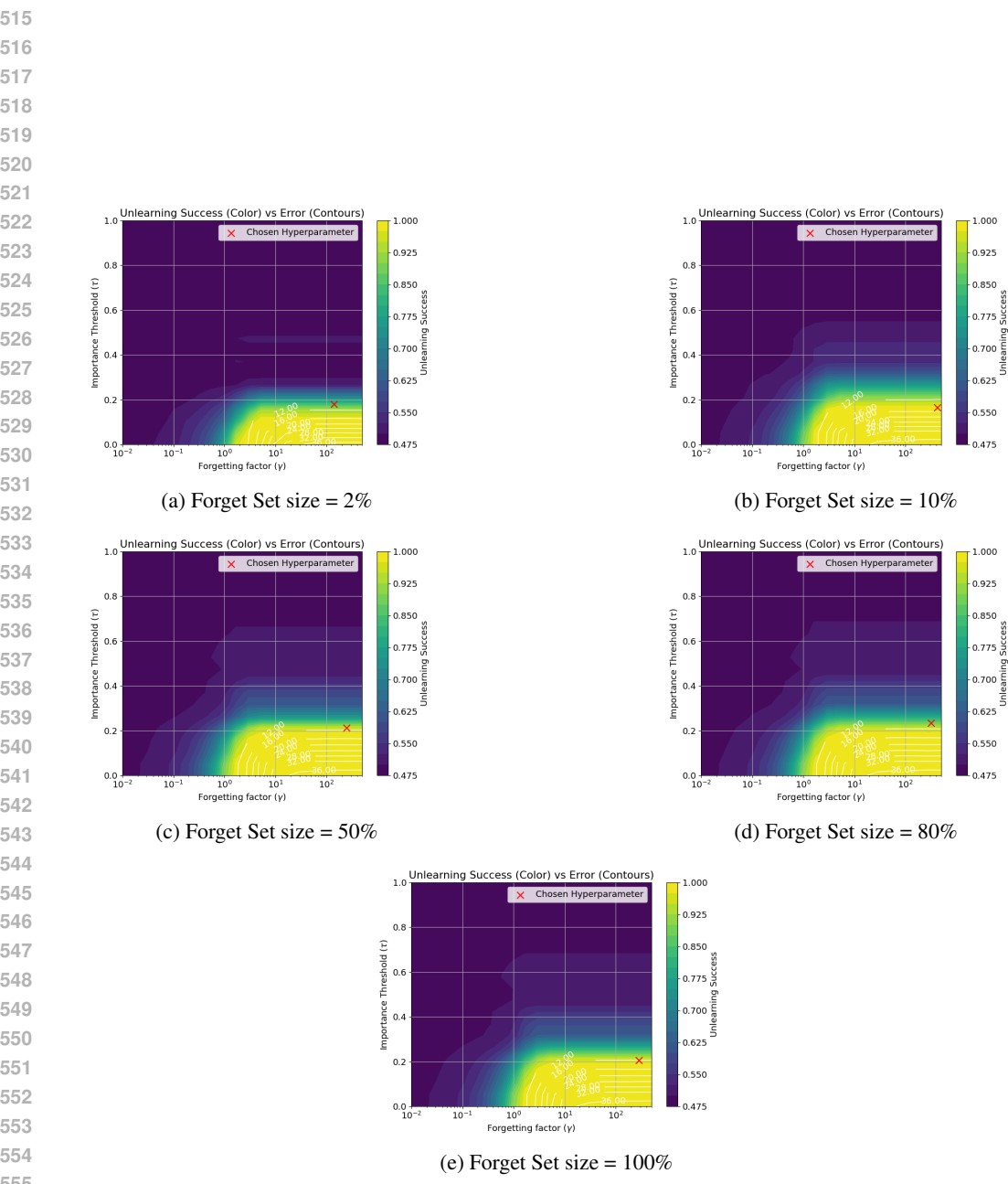

Figure 31: Sensitivity Analysis of SPA on ResNet-18 model and AgeDB dataset with Forget Range of 60-101 for Regression Unlearning

| Unlearner | Utility | Unlearning Success |
|---|---|---|
| Original | $54.398 \pm 0.346$ | $8.886 \pm 0.281$ |
| Naive | $54.705 \pm 0.251$ | $9.553 \pm 0.411$ |
| SPA | $35.260 \pm 6.413$ | $27.410 \pm 5.387$ |
| SPARC | $51.392 \pm 1.004$ | $41.412 \pm 4.716$ |
| EUk | $56.502 \pm 0.107$ | $9.442 \pm 0.307$ |
| CFk | $55.968 \pm 0.134$ | $9.164 \pm 0.311$ |
| SCRUB | $55.716 \pm 0.134$ | $9.654 \pm 0.361$ |
| SSD | $54.402 \pm 0.339$ | $8.898 \pm 0.284$ |

Table 8: BadNet results for ResNet-18 on CIFAR-100 with 2.0% forget size

| Unlearner | Utility | Unlearning Success |
|---|---|---|
| Original | $54.398 \pm 0.346$ | $8.886 \pm 0.281$ |
| Naive | $55.110 \pm 0.347$ | $8.860 \pm 0.373$ |
| SPA | $44.882 \pm 2.547$ | $38.162 \pm 2.872$ |
| SPARC | $52.138 \pm 0.545$ | $44.678 \pm 4.186$ |
| EUk | $56.466 \pm 0.125$ | $9.292 \pm 0.349$ |
| CFk | $56.036 \pm 0.114$ | $9.046 \pm 0.272$ |
| SCRUB | $54.838 \pm 0.288$ | $8.718 \pm 0.138$ |
| SSD | $51.840 \pm 2.559$ | $14.360 \pm 5.317$ |

Table 9: BadNet results for ResNet-18 on CIFAR-100 with 10.0% forget size

### I.1.2 RESNET-18 ON CIFAR-100

| Unlearner | Utility | Unlearning Success |
|---|---|---|
| Original | $54.398 \pm 0.346$ | $8.886 \pm 0.281$ |
| Naive | $55.053 \pm 0.501$ | $9.717 \pm 0.604$ |
| SPA | $46.544 \pm 1.994$ | $39.084 \pm 3.776$ |
| SPARC | $52.416 \pm 0.418$ | $44.280 \pm 4.863$ |
| EUk | $54.006 \pm 0.239$ | $10.436 \pm 0.187$ |
| CFk | $1.004 \pm 0.004$ | $1.004 \pm 0.004$ |
| SCRUB | $55.768 \pm 0.209$ | $9.796 \pm 0.461$ |
| SSD | $43.358 \pm 10.596$ | $23.580 \pm 9.566$ |

Table 10: BadNet results for ResNet-18 on CIFAR-100 with 50.0% forget size

| Unlearner | Utility | Unlearning Success |
|---|---|---|
| Original | $54.398 \pm 0.346$ | $8.886 \pm 0.281$ |
| Naive | $55.023 \pm 0.559$ | $11.463 \pm 0.424$ |
| SPA | $48.970 \pm 1.444$ | $38.656 \pm 4.864$ |
| SPARC | $52.964 \pm 0.416$ | $41.394 \pm 5.796$ |
| EUk | $54.270 \pm 0.151$ | $12.186 \pm 0.255$ |
| CFk | $56.546 \pm 0.138$ | $9.120 \pm 0.324$ |
| SCRUB | $54.446 \pm 0.066$ | $36.104 \pm 5.486$ |
| SSD | $19.578 \pm 9.959$ | $9.980 \pm 4.735$ |

Table 11: BadNet results for ResNet-18 on CIFAR-100 with 80.0% forget size

| Unlearner | Utility | Unlearning Success |
|---|---|---|
| Original | $54.398 \pm 0.346$ | $8.886 \pm 0.281$ |
| Naive | $55.025 \pm 0.344$ | $54.620 \pm 0.363$ |
| SPA | $46.400 \pm 1.992$ | $40.196 \pm 2.853$ |
| SPARC | $52.852 \pm 0.400$ | $45.752 \pm 3.941$ |
| EUk | $51.014 \pm 0.620$ | $50.614 \pm 0.694$ |
| CFk | $50.042 \pm 0.388$ | $25.048 \pm 4.788$ |
| SCRUB | $58.796 \pm 0.146$ | $47.500 \pm 2.235$ |
| SSD | $45.706 \pm 3.130$ | $31.488 \pm 4.698$ |

Table 12: BadNet results for ResNet-18 on CIFAR-100 with 100.0% forget size

| Unlearner | Utility | Unlearning Success |
|---|---|---|
| Original | $85.398 \pm 0.543$ | $15.132 \pm 0.699$ |
| Naive | $85.672 \pm 0.328$ | $15.566 \pm 0.564$ |
| SPA | $77.866 \pm 1.915$ | $62.580 \pm 9.772$ |
| SPARC | $79.472 \pm 1.801$ | $62.694 \pm 10.073$ |
| EUk | $81.772 \pm 0.203$ | $24.380 \pm 0.767$ |
| CFk | $86.722 \pm 0.223$ | $15.488 \pm 0.214$ |
| SCRUB | $75.042 \pm 1.667$ | $14.994 \pm 0.711$ |
| SSD | $73.660 \pm 10.084$ | $31.064 \pm 12.525$ |

Table 13: BadNet results for AllCNN on CIFAR-10 with 2.0% forget size

| Unlearner | Utility | Unlearning Success |
|---|---|---|
| Original | $85.398 \pm 0.543$ | $15.132 \pm 0.699$ |
| Naive | $85.802 \pm 0.300$ | $14.962 \pm 0.333$ |
| SPA | $78.504 \pm 2.165$ | $63.602 \pm 8.515$ |
| SPARC | $80.142 \pm 2.046$ | $61.012 \pm 9.649$ |
| EUk | $79.728 \pm 0.219$ | $32.478 \pm 1.135$ |
| CFk | $86.662 \pm 0.170$ | $15.010 \pm 0.290$ |
| SCRUB | $87.284 \pm 0.136$ | $15.606 \pm 0.267$ |
| SSD | $70.470 \pm 15.126$ | $13.582 \pm 0.928$ |

Table 14: BadNet results for AllCNN on CIFAR-10 with 10.0% forget size

### I.1.3 ALLCNN ON CIFAR-10

| Unlearner | Utility | Unlearning Success |
|---|---|---|
| Original | $85.398 \pm 0.543$ | $15.132 \pm 0.699$ |
| Naive | $85.920 \pm 0.291$ | $17.134 \pm 0.325$ |
| SPA | $78.520 \pm 2.166$ | $63.720 \pm 8.784$ |
| SPARC | $80.400 \pm 2.064$ | $60.410 \pm 10.088$ |
| EUk | $76.654 \pm 0.715$ | $28.184 \pm 1.138$ |
| CFk | $86.862 \pm 0.221$ | $15.330 \pm 0.377$ |
| SCRUB | $70.954 \pm 1.686$ | $18.802 \pm 1.898$ |
| SSD | $85.372 \pm 0.553$ | $23.700 \pm 9.226$ |

Table 15: BadNet results for AllCNN on CIFAR-10 with 50.0% forget size

| Unlearner | Utility | Unlearning Success |
|---|---|---|
| Original | $85.398 \pm 0.543$ | $15.132 \pm 0.699$ |
| Naive | $85.508 \pm 0.389$ | $16.722 \pm 0.551$ |
| SPA | $78.930 \pm 2.296$ | $59.914 \pm 9.765$ |
| SPARC | $80.324 \pm 2.028$ | $58.950 \pm 10.626$ |
| EUk | $72.554 \pm 1.356$ | $63.362 \pm 2.840$ |
| CFk | $86.254 \pm 0.298$ | $14.854 \pm 0.404$ |
| SCRUB | $85.878 \pm 0.079$ | $19.244 \pm 0.729$ |
| SSD | $85.390 \pm 0.546$ | $15.598 \pm 1.147$ |

Table 16: BadNet results for AllCNN on CIFAR-10 with 80.0% forget size

| Unlearner | Utility | Unlearning Success |
|---|---|---|
| Original | $85.398 \pm 0.543$ | $15.132 \pm 0.699$ |
| Naive | $85.510 \pm 0.244$ | $85.298 \pm 0.228$ |
| SPA | $78.620 \pm 2.171$ | $63.244 \pm 8.993$ |
| SPARC | $80.590 \pm 2.159$ | $61.206 \pm 9.933$ |
| EUk | $83.144 \pm 0.294$ | $83.024 \pm 0.308$ |
| CFk | $81.098 \pm 0.499$ | $73.894 \pm 1.920$ |
| SCRUB | $87.242 \pm 0.110$ | $79.292 \pm 0.615$ |
| SSD | $85.076 \pm 0.721$ | $28.198 \pm 13.723$ |

Table 17: BadNet results for AllCNN on CIFAR-10 with 100.0% forget size

| Unlearner | Utility | Unlearning Success |
|---|---|---|
| Original | $85.980 \pm 6.623$ | $17.440 \pm 0.364$ |
| Naive | $91.580 \pm 0.924$ | $17.500 \pm 0.750$ |
| SPA | $81.700 \pm 6.321$ | $79.140 \pm 10.883$ |
| SPARC | $89.320 \pm 1.086$ | $76.840 \pm 13.089$ |
| EUk | $87.460 \pm 0.712$ | $84.740 \pm 0.602$ |
| CFk | $92.460 \pm 0.380$ | $18.220 \pm 0.899$ |
| SCRUB | $78.520 \pm 3.879$ | $22.760 \pm 3.045$ |
| SSD | $75.880 \pm 16.421$ | $29.340 \pm 14.055$ |

Table 18: BadNet results for AllCNN on Lacuna-10 with 2.0% forget size

| Unlearner | Utility | Unlearning Success |
|---|---|---|
| Original | $85.980 \pm 6.623$ | $17.440 \pm 0.364$ |
| Naive | $89.020 \pm 2.127$ | $20.340 \pm 2.116$ |
| SPA | $82.960 \pm 6.382$ | $83.580 \pm 7.052$ |
| SPARC | $88.980 \pm 2.123$ | $87.640 \pm 1.533$ |
| EUk | $90.660 \pm 0.883$ | $75.500 \pm 3.212$ |
| CFk | $91.960 \pm 0.776$ | $17.500 \pm 0.563$ |
| SCRUB | $91.020 \pm 1.098$ | $26.680 \pm 2.756$ |
| SSD | $85.960 \pm 6.617$ | $27.940 \pm 10.566$ |

Table 19: BadNet results for AllCNN on Lacuna-10 with 10.0% forget size

### I.1.4 ALLCNN ON LACUNA-10

| Unlearner | Utility | Unlearning Success |
|---|---|---|
| Original | $85.980 \pm 6.623$ | $17.440 \pm 0.364$ |
| Naive | $90.240 \pm 0.505$ | $33.160 \pm 3.440$ |
| SPA | $82.920 \pm 6.512$ | $82.640 \pm 7.520$ |
| SPARC | $89.480 \pm 1.796$ | $86.580 \pm 1.299$ |
| EUk | $77.260 \pm 2.924$ | $76.620 \pm 2.910$ |
| CFk | $90.360 \pm 0.698$ | $22.960 \pm 1.425$ |
| SCRUB | $72.980 \pm 2.536$ | $20.040 \pm 1.610$ |
| SSD | $52.220 \pm 16.835$ | $33.620 \pm 11.650$ |

Table 20: BadNet results for AllCNN on Lacuna-10 with 50.0% forget size

| Unlearner | Utility | Unlearning Success |
|---|---|---|
| Original | $85.980 \pm 6.623$ | $17.440 \pm 0.364$ |
| Naive | $92.300 \pm 0.202$ | $31.060 \pm 2.071$ |
| SPA | $82.240 \pm 6.286$ | $83.620 \pm 7.437$ |
| SPARC | $89.220 \pm 1.900$ | $88.180 \pm 1.399$ |
| EUk | $91.720 \pm 0.318$ | $87.800 \pm 0.597$ |
| CFk | $67.680 \pm 9.909$ | $67.960 \pm 9.868$ |
| SCRUB | $90.800 \pm 1.770$ | $24.400 \pm 1.273$ |
| SSD | $27.060 \pm 10.669$ | $18.600 \pm 6.860$ |

Table 21: BadNet results for AllCNN on Lacuna-10 with 80.0% forget size

| Unlearner | Utility | Unlearning Success |
|---|---|---|
| Original | $85.980 \pm 6.623$ | $17.440 \pm 0.364$ |
| Naive | $93.000 \pm 0.336$ | $92.900 \pm 0.321$ |
| SPA | $82.180 \pm 6.268$ | $83.580 \pm 7.404$ |
| SPARC | $89.120 \pm 1.935$ | $88.300 \pm 1.312$ |
| EUk | $91.740 \pm 0.492$ | $91.160 \pm 0.336$ |
| CFk | $82.600 \pm 4.944$ | $81.180 \pm 4.643$ |
| SCRUB | $91.900 \pm 0.266$ | $40.060 \pm 6.575$ |
| SSD | $13.200 \pm 3.200$ | $12.980 \pm 2.980$ |

Table 22: BadNet results for AllCNN on Lacuna-10 with 100.0% forget size

| Unlearner | Utility | Unlearning Success |
|-----------|---------|--------------------|
| Original | $56.438 \pm 0.418$ | $9.796 \pm 0.231$ |
| Naive | $56.380 \pm 0.436$ | $9.212 \pm 0.232$ |
| SPA | $53.198 \pm 0.671$ | $40.610 \pm 1.661$ |
| SPARC | $55.042 \pm 0.610$ | $42.810 \pm 1.621$ |
| EUk | $57.606 \pm 0.208$ | $11.938 \pm 0.125$ |
| CFk | $59.388 \pm 0.168$ | $9.752 \pm 0.128$ |
| SCRUB | $59.872 \pm 0.204$ | $10.480 \pm 0.127$ |
| SSD | $45.216 \pm 11.061$ | $8.008 \pm 1.767$ |

Table 23: BadNet results for AllCNN on CIFAR-100 with 2.0% forget size

| Unlearner | Utility | Unlearning Success |
|-----------|---------|--------------------|
| Original | $56.438 \pm 0.418$ | $9.796 \pm 0.231$ |
| Naive | $56.844 \pm 0.350$ | $9.152 \pm 0.249$ |
| SPA | $53.336 \pm 0.624$ | $41.658 \pm 3.443$ |
| SPARC | $55.172 \pm 0.542$ | $43.562 \pm 3.838$ |
| EUk | $55.616 \pm 0.138$ | $12.802 \pm 0.178$ |
| CFk | $59.086 \pm 0.180$ | $9.806 \pm 0.137$ |
| SCRUB | $51.278 \pm 0.446$ | $9.608 \pm 0.243$ |
| SSD | $55.672 \pm 0.591$ | $13.796 \pm 1.912$ |

Table 24: BadNet results for AllCNN on CIFAR-100 with 10.0% forget size

### I.1.5   ALLCNN ON CIFAR-100

| Unlearner | Utility | Unlearning Success |
|---|---|---|
| Original | $56.438 \pm 0.418$ | $9.796 \pm 0.231$ |
| Naive | $56.744 \pm 1.037$ | $11.178 \pm 0.811$ |
| SPA | $53.344 \pm 0.439$ | $43.342 \pm 2.337$ |
| SPARC | $55.210 \pm 0.450$ | $44.618 \pm 3.053$ |
| EUk | $55.462 \pm 0.675$ | $10.902 \pm 0.663$ |
| CFk | $59.206 \pm 0.137$ | $9.674 \pm 0.152$ |
| SCRUB | $56.918 \pm 0.182$ | $11.790 \pm 0.942$ |
| SSD | $33.106 \pm 12.088$ | $15.250 \pm 6.988$ |

Table 25: BadNet results for AllCNN on CIFAR-100 with 50.0% forget size

| Unlearner | Utility | Unlearning Success |
|---|---|---|
| Original | $56.438 \pm 0.418$ | $9.796 \pm 0.231$ |
| Naive | $57.988 \pm 0.820$ | $13.060 \pm 0.650$ |
| SPA | $54.260 \pm 0.370$ | $40.308 \pm 4.073$ |
| SPARC | $55.690 \pm 0.374$ | $40.774 \pm 4.974$ |
| EUk | $53.432 \pm 0.705$ | $14.636 \pm 1.000$ |
| CFk | $58.764 \pm 0.190$ | $9.468 \pm 0.196$ |
| SCRUB | $58.924 \pm 0.082$ | $10.170 \pm 0.250$ |
| SSD | $44.190 \pm 10.816$ | $9.772 \pm 2.504$ |

Table 26: BadNet results for AllCNN on CIFAR-100 with 80.0% forget size

| Unlearner | Utility | Unlearning Success |
|---|---|---|
| Original | $56.438 \pm 0.418$ | $9.796 \pm 0.231$ |
| Naive | $58.530 \pm 1.048$ | $58.114 \pm 1.092$ |
| SPA | $53.776 \pm 0.435$ | $42.926 \pm 2.412$ |
| SPARC | $55.876 \pm 0.355$ | $44.556 \pm 3.007$ |
| EUk | $53.388 \pm 0.183$ | $51.536 \pm 0.242$ |
| CFk | $52.120 \pm 0.401$ | $25.524 \pm 0.968$ |
| SCRUB | $59.682 \pm 0.234$ | $28.622 \pm 6.291$ |
| SSD | $26.668 \pm 8.127$ | $25.624 \pm 7.762$ |

Table 27: BadNet results for AllCNN on CIFAR-100 with 100.0% forget size

| Unlearner | Utility | Unlearning Success |
|-----------|---------|--------------------|
| Original | $77.282 \pm 0.447$ | $15.978 \pm 0.446$ |
| SPA | $72.672 \pm 2.014$ | $51.268 \pm 9.422$ |
| SPARC | $73.450 \pm 1.790$ | $52.152 \pm 9.520$ |
| SCRUB | $77.150 \pm 0.436$ | $16.120 \pm 0.715$ |
| SSD | $64.882 \pm 8.942$ | $52.400 \pm 12.797$ |

Table 28: BadNet results for ViT on CIFAR-10 with 2.0% forget size

| Unlearner | Utility | Unlearning Success |
|-----------|---------|--------------------|
| Original | $77.282 \pm 0.447$ | $15.978 \pm 0.446$ |
| SPA | $73.712 \pm 1.925$ | $51.794 \pm 9.770$ |
| SPARC | $74.712 \pm 1.614$ | $54.000 \pm 9.945$ |
| SCRUB | $70.050 \pm 0.604$ | $16.820 \pm 0.871$ |
| SSD | $76.664 \pm 0.546$ | $60.632 \pm 11.806$ |

Table 29: BadNet results for ViT on CIFAR-10 with 10.0% forget size

## I.1.6    VIT ON CIFAR-10

| Unlearner | Utility | Unlearning Success |
|---|---|---|
| Original | $77.282 \pm 0.447$ | $15.978 \pm 0.446$ |
| SPA | $73.908 \pm 1.938$ | $54.912 \pm 9.118$ |
| SPARC | $74.934 \pm 1.586$ | $57.034 \pm 9.181$ |
| SCRUB | $37.224 \pm 1.995$ | $21.874 \pm 4.512$ |
| SSD | $74.842 \pm 1.808$ | $74.078 \pm 1.899$ |

Table 30: BadNet results for ViT on CIFAR-10 with 50.0% forget size

| Unlearner | Utility | Unlearning Success |
|---|---|---|
| Original | $77.282 \pm 0.447$ | $15.978 \pm 0.446$ |
| SPA | $74.254 \pm 1.680$ | $52.084 \pm 9.787$ |
| SPARC | $67.674 \pm 1.416$ | $54.242 \pm 8.275$ |
| SCRUB | $54.890 \pm 0.891$ | $26.598 \pm 5.626$ |
| SSD | $77.188 \pm 0.378$ | $52.198 \pm 10.198$ |

Table 31: BadNet results for ViT on CIFAR-10 with 80.0% forget size

| Unlearner | Utility | Unlearning Success |
|---|---|---|
| Original | $77.282 \pm 0.447$ | $15.978 \pm 0.446$ |
| SPA | $73.704 \pm 2.122$ | $54.548 \pm 9.085$ |
| SPARC | $74.690 \pm 1.143$ | $59.346 \pm 9.018$ |
| SCRUB | $38.536 \pm 1.572$ | $31.502 \pm 1.066$ |
| SSD | $76.964 \pm 0.398$ | $72.694 \pm 2.799$ |

Table 32: BadNet results for ViT on CIFAR-10 with 100.0% forget size

| Unlearner | Utility | Unlearning Success |
|-----------|---------|-------------------|
| Original | $82.360 \pm 0.502$ | $47.900 \pm 2.532$ |
| Naive | $80.860 \pm 1.166$ | $46.700 \pm 2.311$ |
| SPA | $63.940 \pm 11.553$ | $23.300 \pm 7.044$ |
| SPARC | $80.800 \pm 0.459$ | $46.700 \pm 2.239$ |
| EUk | $27.100 \pm 3.737$ | $16.900 \pm 6.558$ |
| CFk | $78.200 \pm 1.747$ | $47.600 \pm 1.134$ |
| SCRUB | $60.160 \pm 2.354$ | $31.700 \pm 4.989$ |
| SSD | $24.700 \pm 14.700$ | $20.600 \pm 12.624$ |

Table 33: Interclass Confusion results for ResNet-18 on Lacuna-10 with 2.0% forget size

| Unlearner | Utility | Unlearning Success |
|-----------|---------|-------------------|
| Original | $82.360 \pm 0.502$ | $47.900 \pm 2.532$ |
| Naive | $81.940 \pm 0.571$ | $48.500 \pm 1.432$ |
| SPA | $65.400 \pm 10.871$ | $12.100 \pm 5.250$ |
| SPARC | $80.180 \pm 0.939$ | $48.800 \pm 2.611$ |
| EUk | $57.420 \pm 4.093$ | $24.100 \pm 5.168$ |
| CFk | $73.400 \pm 2.973$ | $43.300 \pm 5.241$ |
| SCRUB | $78.880 \pm 0.860$ | $42.900 \pm 2.713$ |
| SSD | $33.520 \pm 14.119$ | $26.600 \pm 10.980$ |

Table 34: Interclass Confusion results for ResNet-18 on Lacuna-10 with 10.0% forget size

## I.2 LABEL-ONLY MANIPULATIONS (INTERCLASS CONFUSION)

This section presents results for interclass confusion attacks, where only labels are manipulated without modifying features.

### I.2.1 RESNET-18 ON LACUNA-10

| Unlearner | Utility | Unlearning Success |
|---|---|---|
| Original | $82.360 \pm 0.502$ | $47.900 \pm 2.532$ |
| Naive | $82.480 \pm 0.924$ | $58.100 \pm 1.317$ |
| SPA | $74.800 \pm 1.636$ | $13.500 \pm 4.117$ |
| SPARC | $81.920 \pm 0.748$ | $53.600 \pm 2.187$ |
| EUk | $64.000 \pm 5.932$ | $37.000 \pm 5.143$ |
| CFk | $67.440 \pm 4.769$ | $40.800 \pm 4.027$ |
| SCRUB | $83.160 \pm 0.426$ | $54.100 \pm 1.826$ |
| SSD | $50.300 \pm 15.471$ | $30.500 \pm 12.482$ |

Table 35: Interclass Confusion results for ResNet-18 on Lacuna-10 with 50.0% forget size

| Unlearner | Utility | Unlearning Success |
|---|---|---|
| Original | $82.360 \pm 0.502$ | $47.900 \pm 2.532$ |
| Naive | $85.140 \pm 0.621$ | $62.500 \pm 0.418$ |
| SPA | $74.320 \pm 2.294$ | $13.200 \pm 4.529$ |
| SPARC | $83.240 \pm 0.868$ | $58.600 \pm 2.426$ |
| EUk | $42.020 \pm 5.433$ | $15.700 \pm 7.067$ |
| CFk | $72.540 \pm 2.825$ | $47.200 \pm 5.551$ |
| SCRUB | $79.400 \pm 0.550$ | $44.400 \pm 1.713$ |
| SSD | $50.080 \pm 14.898$ | $30.600 \pm 12.331$ |

Table 36: Interclass Confusion results for ResNet-18 on Lacuna-10 with 80.0% forget size

| Unlearner | Utility | Unlearning Success |
|---|---|---|
| Original | $82.360 \pm 0.502$ | $47.900 \pm 2.532$ |
| Naive | $85.120 \pm 1.795$ | $71.900 \pm 4.139$ |
| SPA | $75.560 \pm 1.257$ | $14.800 \pm 4.326$ |
| SPARC | $86.540 \pm 0.236$ | $74.600 \pm 2.205$ |
| EUk | $68.560 \pm 3.279$ | $55.600 \pm 4.956$ |
| CFk | $71.300 \pm 5.736$ | $50.800 \pm 10.039$ |
| SCRUB | $83.560 \pm 0.488$ | $63.900 \pm 1.478$ |
| SSD | $73.660 \pm 8.017$ | $45.900 \pm 4.357$ |

Table 37: Interclass Confusion results for ResNet-18 on Lacuna-10 with 100.0% forget size

| Unlearner | Utility | Unlearning Success |
|---|---|---|
| Original | $55.118 \pm 0.289$ | $30.500 \pm 3.150$ |
| Naive | $55.046 \pm 0.311$ | $25.800 \pm 2.918$ |
| SPA | $39.374 \pm 6.330$ | $20.000 \pm 6.223$ |
| SPARC | $52.678 \pm 0.179$ | $26.900 \pm 3.910$ |
| EUk | $43.864 \pm 0.382$ | $26.300 \pm 2.422$ |
| CFk | $48.636 \pm 0.366$ | $24.100 \pm 1.860$ |
| SCRUB | $49.516 \pm 0.958$ | $24.800 \pm 3.491$ |
| SSD | $33.234 \pm 13.177$ | $12.800 \pm 6.294$ |

Table 38: Interclass Confusion results for ResNet-18 on CIFAR-100 with 2.0% forget size

| Unlearner | Utility | Unlearning Success |
|---|---|---|
| Original | $55.118 \pm 0.289$ | $30.500 \pm 3.150$ |
| Naive | $55.492 \pm 0.404$ | $24.400 \pm 4.394$ |
| SPA | $46.706 \pm 6.325$ | $18.000 \pm 5.990$ |
| SPARC | $56.370 \pm 0.273$ | $29.500 \pm 2.465$ |
| EUk | $47.962 \pm 0.903$ | $22.900 \pm 2.130$ |
| CFk | $41.172 \pm 0.600$ | $15.200 \pm 2.239$ |
| SCRUB | $58.304 \pm 0.138$ | $29.300 \pm 2.818$ |
| SSD | $33.388 \pm 13.059$ | $11.500 \pm 6.277$ |

Table 39: Interclass Confusion results for ResNet-18 on CIFAR-100 with 10.0% forget size

## I.2.2 RESNET-18 ON CIFAR-100

| Unlearner | Utility | Unlearning Success |
|---|---|---|
| Original | $55.118 \pm 0.289$ | $30.500 \pm 3.150$ |
| Naive | $54.476 \pm 0.317$ | $29.300 \pm 2.478$ |
| SPA | $52.674 \pm 1.188$ | $13.400 \pm 5.662$ |
| SPARC | $56.452 \pm 0.370$ | $36.000 \pm 3.399$ |
| EUk | $41.754 \pm 0.447$ | $25.500 \pm 3.029$ |
| CFk | $37.660 \pm 1.186$ | $13.600 \pm 1.699$ |
| SCRUB | $50.190 \pm 0.782$ | $27.900 \pm 4.035$ |
| SSD | $33.348 \pm 11.884$ | $12.100 \pm 6.634$ |

Table 40: Interclass Confusion results for ResNet-18 on CIFAR-100 with 50.0% forget size

| Unlearner | Utility | Unlearning Success |
|---|---|---|
| Original | $55.118 \pm 0.289$ | $30.500 \pm 3.150$ |
| Naive | $54.970 \pm 0.146$ | $35.900 \pm 3.088$ |
| SPA | $50.258 \pm 1.465$ | $9.100 \pm 3.938$ |
| SPARC | $56.180 \pm 0.364$ | $42.300 \pm 4.784$ |
| EUk | $53.794 \pm 0.169$ | $29.300 \pm 2.909$ |
| CFk | $48.770 \pm 0.329$ | $33.400 \pm 3.215$ |
| SCRUB | $56.050 \pm 0.182$ | $35.200 \pm 4.389$ |
| SSD | $21.722 \pm 10.148$ | $5.100 \pm 4.851$ |

Table 41: Interclass Confusion results for ResNet-18 on CIFAR-100 with 80.0% forget size

| Unlearner | Utility | Unlearning Success |
|---|---|---|
| Original | $55.118 \pm 0.289$ | $30.500 \pm 3.150$ |
| Naive | $55.672 \pm 0.433$ | $39.900 \pm 5.088$ |
| SPA | $51.636 \pm 0.586$ | $12.300 \pm 4.343$ |
| SPARC | $56.834 \pm 0.141$ | $44.300 \pm 5.229$ |
| EUk | $53.726 \pm 0.160$ | $35.000 \pm 4.307$ |
| CFk | $53.376 \pm 0.267$ | $33.700 \pm 6.904$ |
| SCRUB | $47.876 \pm 0.430$ | $29.400 \pm 5.247$ |
| SSD | $40.428 \pm 10.080$ | $8.200 \pm 4.303$ |

Table 42: Interclass Confusion results for ResNet-18 on CIFAR-100 with 100.0% forget size

| Unlearner | Utility | Unlearning Success |
|---|---|---|
| Original | $78.872 \pm 0.395$ | $49.970 \pm 0.979$ |
| Naive | $78.430 \pm 0.170$ | $52.040 \pm 1.528$ |
| SPA | $78.646 \pm 0.401$ | $48.920 \pm 0.811$ |
| SPARC | $80.702 \pm 0.301$ | $54.040 \pm 1.502$ |
| EUk | $79.780 \pm 0.244$ | $52.280 \pm 0.806$ |
| CFk | $80.142 \pm 0.242$ | $53.400 \pm 1.147$ |
| SCRUB | $73.008 \pm 1.823$ | $46.010 \pm 2.446$ |
| SSD | $77.894 \pm 1.044$ | $46.160 \pm 4.205$ |

Table 43: Interclass Confusion results for AllCNN on CIFAR-10 with 2.0% forget size

| Unlearner | Utility | Unlearning Success |
|---|---|---|
| Original | $78.872 \pm 0.395$ | $49.970 \pm 0.979$ |
| Naive | $79.898 \pm 0.751$ | $55.500 \pm 2.071$ |
| SPA | $78.928 \pm 0.347$ | $49.660 \pm 0.984$ |
| SPARC | $79.696 \pm 0.292$ | $57.070 \pm 1.951$ |
| EUk | $81.414 \pm 0.678$ | $68.990 \pm 3.136$ |
| CFk | $80.594 \pm 0.189$ | $54.180 \pm 1.115$ |
| SCRUB | $82.444 \pm 0.230$ | $63.640 \pm 2.174$ |
| SSD | $64.378 \pm 13.596$ | $39.740 \pm 10.187$ |

Table 44: Interclass Confusion results for AllCNN on CIFAR-10 with 10.0% forget size

### I.2.3 ALLCNN ON CIFAR-10

| Unlearner | Utility | Unlearning Success |
|---|---|---|
| Original | $78.872 \pm 0.395$ | $49.970 \pm 0.979$ |
| Naive | $82.088 \pm 0.480$ | $65.280 \pm 2.485$ |
| SPA | $78.870 \pm 0.396$ | $49.970 \pm 0.979$ |
| SPARC | $82.778 \pm 0.260$ | $67.130 \pm 2.285$ |
| EUk | $82.140 \pm 0.480$ | $71.000 \pm 1.894$ |
| CFk | $80.726 \pm 0.437$ | $56.330 \pm 0.941$ |
| SCRUB | $81.910 \pm 0.492$ | $75.550 \pm 3.405$ |
| SSD | $78.886 \pm 0.397$ | $49.960 \pm 0.991$ |

Table 45: Interclass Confusion results for AllCNN on CIFAR-10 with 50.0% forget size

| Unlearner | Utility | Unlearning Success |
|---|---|---|
| Original | $78.872 \pm 0.395$ | $49.970 \pm 0.979$ |
| Naive | $83.648 \pm 0.300$ | $72.700 \pm 2.571$ |
| SPA | $78.874 \pm 0.398$ | $49.970 \pm 0.979$ |
| SPARC | $81.104 \pm 0.429$ | $69.580 \pm 1.535$ |
| EUk | $81.776 \pm 0.659$ | $75.640 \pm 3.568$ |
| CFk | $81.962 \pm 0.105$ | $59.610 \pm 1.049$ |
| SCRUB | $82.744 \pm 0.184$ | $68.250 \pm 2.450$ |
| SSD | $78.872 \pm 0.395$ | $49.970 \pm 0.979$ |

Table 46: Interclass Confusion results for AllCNN on CIFAR-10 with 80.0% forget size

| Unlearner | Utility | Unlearning Success |
|---|---|---|
| Original | $78.872 \pm 0.395$ | $49.970 \pm 0.979$ |
| Naive | $85.530 \pm 0.442$ | $77.940 \pm 2.616$ |
| SPA | $78.872 \pm 0.395$ | $49.970 \pm 0.979$ |
| SPARC | $85.504 \pm 0.362$ | $77.430 \pm 2.949$ |
| EUk | $84.744 \pm 0.350$ | $75.050 \pm 2.873$ |
| CFk | $83.650 \pm 0.339$ | $71.920 \pm 2.531$ |
| SCRUB | $84.626 \pm 0.065$ | $70.780 \pm 1.282$ |
| SSD | $78.868 \pm 0.392$ | $49.850 \pm 1.006$ |

Table 47: Interclass Confusion results for AllCNN on CIFAR-10 with 100.0% forget size

| Unlearner | Utility | Unlearning Success |
|---|---|---|
| Original | $83.460 \pm 0.655$ | $46.300 \pm 1.347$ |
| Naive | $81.980 \pm 0.942$ | $44.900 \pm 2.040$ |
| SPA | $83.960 \pm 0.734$ | $46.700 \pm 1.420$ |
| SPARC | $83.600 \pm 0.494$ | $46.400 \pm 1.005$ |
| EUk | $82.940 \pm 0.610$ | $45.600 \pm 1.355$ |
| CFk | $83.400 \pm 0.552$ | $46.000 \pm 1.387$ |
| SCRUB | $83.640 \pm 0.589$ | $47.300 \pm 0.718$ |
| SSD | $24.660 \pm 14.660$ | $9.200 \pm 9.200$ |

Table 48: Interclass Confusion results for AllCNN on Lacuna-10 with 2.0% forget size

| Unlearner | Utility | Unlearning Success |
|---|---|---|
| Original | $83.460 \pm 0.655$ | $46.300 \pm 1.347$ |
| Naive | $83.080 \pm 1.626$ | $48.500 \pm 2.345$ |
| SPA | $83.420 \pm 0.753$ | $45.500 \pm 1.837$ |
| SPARC | $83.800 \pm 0.735$ | $47.300 \pm 1.300$ |
| EUk | $83.400 \pm 0.551$ | $45.700 \pm 1.428$ |
| CFk | $83.860 \pm 0.574$ | $48.100 \pm 1.308$ |
| SCRUB | $83.580 \pm 0.515$ | $44.900 \pm 0.914$ |
| SSD | $35.520 \pm 14.246$ | $24.200 \pm 10.223$ |

Table 49: Interclass Confusion results for AllCNN on Lacuna-10 with 10.0% forget size

### I.2.4 ALLCNN ON LACUNA-10

| **Unlearner** | Utility | Unlearning Success |
|---|---|---|
| Original | $83.460 \pm 0.655$ | $46.300 \pm 1.347$ |
| Naive | $84.660 \pm 1.170$ | $56.900 \pm 3.184$ |
| SPA | $83.380 \pm 0.614$ | $45.200 \pm 1.402$ |
| SPARC | $84.000 \pm 0.823$ | $55.100 \pm 0.600$ |
| EUk | $84.240 \pm 0.408$ | $51.000 \pm 1.423$ |
| CFk | $83.480 \pm 0.781$ | $48.000 \pm 1.423$ |
| SCRUB | $84.280 \pm 0.489$ | $52.900 \pm 1.017$ |
| SSD | $83.160 \pm 0.726$ | $46.500 \pm 1.275$ |

Table 50: Interclass Confusion results for AllCNN on Lacuna-10 with 50.0% forget size

| **Unlearner** | Utility | Unlearning Success |
|---|---|---|
| Original | $83.460 \pm 0.655$ | $46.300 \pm 1.347$ |
| Naive | $87.500 \pm 0.308$ | $66.600 \pm 2.106$ |
| SPA | $83.420 \pm 0.667$ | $45.800 \pm 1.319$ |
| SPARC | $84.620 \pm 0.728$ | $53.700 \pm 2.591$ |
| EUk | $85.520 \pm 0.581$ | $55.100 \pm 2.353$ |
| CFk | $84.920 \pm 0.665$ | $54.500 \pm 2.043$ |
| SCRUB | $84.640 \pm 0.375$ | $55.300 \pm 0.800$ |
| SSD | $81.460 \pm 0.919$ | $45.700 \pm 1.147$ |

Table 51: Interclass Confusion results for AllCNN on Lacuna-10 with 80.0% forget size

| **Unlearner** | Utility | Unlearning Success |
|---|---|---|
| Original | $83.460 \pm 0.655$ | $46.300 \pm 1.347$ |
| Naive | $87.500 \pm 3.105$ | $75.000 \pm 9.449$ |
| SPA | $83.420 \pm 0.718$ | $45.100 \pm 1.259$ |
| SPARC | $86.340 \pm 0.980$ | $71.000 \pm 2.641$ |
| EUk | $85.820 \pm 0.601$ | $58.200 \pm 2.853$ |
| CFk | $80.560 \pm 2.739$ | $61.900 \pm 5.787$ |
| SCRUB | $81.760 \pm 1.218$ | $68.700 \pm 3.777$ |
| SSD | $43.700 \pm 13.382$ | $37.000 \pm 5.191$ |

Table 52: Interclass Confusion results for AllCNN on Lacuna-10 with 100.0% forget size

| **Unlearner** | Utility | Unlearning Success |
|---|---|---|
| Original | $57.682 \pm 0.722$ | $24.600 \pm 2.843$ |
| Naive | $56.220 \pm 0.721$ | $24.300 \pm 3.250$ |
| SPA | $57.678 \pm 0.711$ | $24.800 \pm 3.036$ |
| SPARC | $58.338 \pm 0.575$ | $24.500 \pm 2.434$ |
| EUk | $57.682 \pm 0.412$ | $26.800 \pm 1.617$ |
| CFk | $59.720 \pm 0.232$ | $27.500 \pm 1.851$ |
| SCRUB | $59.558 \pm 0.355$ | $26.100 \pm 2.727$ |
| SSD | $55.238 \pm 1.707$ | $17.200 \pm 3.165$ |

Table 53: Interclass Confusion results for AllCNN on CIFAR-100 with 2.0% forget size

| **Unlearner** | Utility | Unlearning Success |
|---|---|---|
| Original | $57.682 \pm 0.722$ | $24.600 \pm 2.843$ |
| Naive | $57.568 \pm 0.652$ | $31.400 \pm 1.536$ |
| SPA | $57.730 \pm 0.721$ | $24.800 \pm 3.036$ |
| SPARC | $59.152 \pm 0.108$ | $28.400 \pm 2.795$ |
| EUk | $58.776 \pm 0.365$ | $27.300 \pm 2.442$ |
| CFk | $58.538 \pm 0.538$ | $25.200 \pm 1.881$ |
| SCRUB | $60.414 \pm 0.277$ | $29.900 \pm 3.367$ |
| SSD | $45.438 \pm 8.085$ | $22.400 \pm 6.660$ |

Table 54: Interclass Confusion results for AllCNN on CIFAR-100 with 10.0% forget size

## I.2.5 ALLCNN ON CIFAR-100

| Unlearner | Utility | Unlearning Success |
|---|---|---|
| Original | $57.682 \pm 0.722$ | $24.600 \pm 2.843$ |
| Naive | $57.230 \pm 0.442$ | $32.200 \pm 3.408$ |
| SPA | $57.688 \pm 0.725$ | $24.400 \pm 3.043$ |
| SPARC | $59.634 \pm 0.262$ | $29.300 \pm 2.918$ |
| EUk | $57.920 \pm 0.492$ | $26.800 \pm 1.729$ |
| CFk | $59.152 \pm 0.427$ | $26.100 \pm 1.946$ |
| SCRUB | $59.888 \pm 0.255$ | $28.500 \pm 2.475$ |
| SSD | $54.614 \pm 3.399$ | $21.800 \pm 3.262$ |

Table 55: Interclass Confusion results for AllCNN on CIFAR-100 with 50.0% forget size

| Unlearner | Utility | Unlearning Success |
|---|---|---|
| Original | $57.682 \pm 0.722$ | $24.600 \pm 2.843$ |
| Naive | $57.962 \pm 0.880$ | $31.900 \pm 3.303$ |
| SPA | $57.686 \pm 0.722$ | $24.400 \pm 3.043$ |
| SPARC | $59.542 \pm 0.152$ | $32.300 \pm 2.750$ |
| EUk | $54.500 \pm 0.462$ | $35.300 \pm 6.076$ |
| CFk | $58.552 \pm 0.410$ | $27.300 \pm 2.256$ |
| SCRUB | $59.780 \pm 0.271$ | $28.000 \pm 2.881$ |
| SSD | $51.698 \pm 3.040$ | $30.600 \pm 3.970$ |

Table 56: Interclass Confusion results for AllCNN on CIFAR-100 with 80.0% forget size

| Unlearner | Utility | Unlearning Success |
|---|---|---|
| Original | $57.682 \pm 0.722$ | $24.600 \pm 2.843$ |
| Naive | $58.396 \pm 0.783$ | $40.100 \pm 3.884$ |
| SPA | $57.686 \pm 0.722$ | $24.400 \pm 3.043$ |
| SPARC | $59.714 \pm 0.186$ | $34.700 \pm 3.341$ |
| EUk | $58.160 \pm 0.286$ | $38.300 \pm 4.906$ |
| CFk | $59.750 \pm 0.190$ | $30.700 \pm 2.533$ |
| SCRUB | $56.438 \pm 0.485$ | $31.000 \pm 2.632$ |
| SSD | $50.022 \pm 3.216$ | $37.000 \pm 6.101$ |

Table 57: Interclass Confusion results for AllCNN on CIFAR-100 with 100.0% forget size

| Unlearner | Utility | Unlearning Success |
|---|---|---|
| Original | $85.174 \pm 0.124$ | $10.000 \pm 10.000$ |
| Naive | $85.410 \pm 0.109$ | $28.000 \pm 17.436$ |
| SPA | $52.957 \pm 6.306$ | $52.000 \pm 18.547$ |
| SPARC | $75.842 \pm 0.527$ | $76.000 \pm 14.697$ |
| EUk | $83.235 \pm 0.156$ | $36.000 \pm 22.271$ |
| CFk | $84.738 \pm 0.117$ | $36.000 \pm 19.391$ |
| SCRUB | $74.318 \pm 0.663$ | $28.000 \pm 14.967$ |
| SSD | $85.126 \pm 0.176$ | $24.000 \pm 19.391$ |

Table 58: Gradient Matching results for ResNet-18 on CIFAR-10 with 2.0% forget size

| Unlearner | Utility | Unlearning Success |
|---|---|---|
| Original | $85.174 \pm 0.124$ | $10.000 \pm 10.000$ |
| Naive | $85.252 \pm 0.076$ | $32.000 \pm 20.591$ |
| SPA | $71.440 \pm 1.909$ | $68.000 \pm 18.547$ |
| SPARC | $77.260 \pm 0.354$ | $72.000 \pm 17.436$ |
| EUk | $78.408 \pm 0.985$ | $48.000 \pm 18.547$ |
| CFk | $85.464 \pm 0.110$ | $28.000 \pm 19.596$ |
| SCRUB | $86.623 \pm 0.085$ | $40.000 \pm 18.974$ |
| SSD | $85.054 \pm 0.237$ | $24.000 \pm 19.391$ |

Table 59: Gradient Matching results for ResNet-18 on CIFAR-10 with 10.0% forget size

## I.3 FEATURE-ONLY MANIPULATIONS (GRADIENT MATCHING)

This section presents results for gradient matching attacks, where features are manipulated to match target gradients.

### I.3.1 RESNET-18 ON CIFAR-10

| Unlearner | Utility | Unlearning Success |
|---|---|---|
| Original | $85.174 \pm 0.124$ | $10.000 \pm 10.000$ |
| Naive | $85.505 \pm 0.177$ | $36.000 \pm 19.391$ |
| SPA | $69.746 \pm 1.795$ | $68.000 \pm 18.547$ |
| SPARC | $77.674 \pm 0.490$ | $72.000 \pm 17.436$ |
| EUk | $85.495 \pm 0.112$ | $28.000 \pm 19.596$ |
| CFk | $86.123 \pm 0.101$ | $28.000 \pm 19.596$ |
| SCRUB | $77.710 \pm 0.420$ | $44.000 \pm 19.391$ |
| SSD | $85.084 \pm 0.212$ | $24.000 \pm 19.391$ |

Table 60: Gradient Matching results for ResNet-18 on CIFAR-10 with 50.0% forget size

| Unlearner | Utility | Unlearning Success |
|---|---|---|
| Original | $85.174 \pm 0.124$ | $10.000 \pm 10.000$ |
| Naive | $85.026 \pm 0.174$ | $44.000 \pm 23.152$ |
| SPA | $73.674 \pm 1.199$ | $76.000 \pm 16.000$ |
| SPARC | $77.902 \pm 0.374$ | $76.000 \pm 14.697$ |
| EUk | $78.573 \pm 1.186$ | $60.000 \pm 14.142$ |
| CFk | $85.673 \pm 0.075$ | $32.000 \pm 20.591$ |
| SCRUB | $85.716 \pm 0.103$ | $76.000 \pm 19.391$ |
| SSD | $83.228 \pm 1.997$ | $24.000 \pm 19.391$ |

Table 61: Gradient Matching results for ResNet-18 on CIFAR-10 with 80.0% forget size

| Unlearner | Utility | Unlearning Success |
|---|---|---|
| Original | $85.174 \pm 0.124$ | $10.000 \pm 10.000$ |
| Naive | $85.252 \pm 0.127$ | $100.000 \pm 0.000$ |
| SPA | $69.443 \pm 1.891$ | $68.000 \pm 18.547$ |
| SPARC | $77.135 \pm 0.488$ | $72.000 \pm 19.596$ |
| EUk | $80.661 \pm 0.144$ | $100.000 \pm 0.000$ |
| CFk | $83.608 \pm 0.161$ | $88.000 \pm 8.000$ |
| SCRUB | $85.919 \pm 0.101$ | $76.000 \pm 19.391$ |
| SSD | $85.126 \pm 0.176$ | $24.000 \pm 19.391$ |

Table 62: Gradient Matching results for ResNet-18 on CIFAR-10 with 100.0% forget size

| Unlearner | Utility | Unlearning Success |
|---|---|---|
| Original | $84.912 \pm 0.928$ | $40.000 \pm 40.000$ |
| Naive | $85.142 \pm 0.165$ | $20.000 \pm 12.649$ |
| SPA | $77.456 \pm 0.707$ | $92.000 \pm 8.000$ |
| SPARC | $79.301 \pm 0.575$ | $80.000 \pm 12.649$ |
| EUk | $77.011 \pm 0.573$ | $40.000 \pm 14.142$ |
| CFk | $71.793 \pm 1.076$ | $48.000 \pm 13.565$ |
| SCRUB | $75.698 \pm 0.518$ | $60.000 \pm 14.142$ |
| SSD | $83.837 \pm 1.135$ | $24.000 \pm 19.391$ |

Table 63: Gradient Matching results for AllCNN on CIFAR-10 with 2.0% forget size

| Unlearner | Utility | Unlearning Success |
|---|---|---|
| Original | $84.912 \pm 0.928$ | $40.000 \pm 40.000$ |
| Naive | $83.880 \pm 0.878$ | $24.000 \pm 14.697$ |
| SPA | $77.044 \pm 0.593$ | $88.000 \pm 8.000$ |
| SPARC | $77.761 \pm 0.632$ | $76.000 \pm 14.697$ |
| EUk | $80.197 \pm 0.332$ | $56.000 \pm 17.205$ |
| CFk | $79.944 \pm 0.447$ | $52.000 \pm 17.436$ |
| SCRUB | $83.703 \pm 0.335$ | $32.000 \pm 16.248$ |
| SSD | $82.885 \pm 1.362$ | $28.000 \pm 13.565$ |

Table 64: Gradient Matching results for AllCNN on CIFAR-10 with 10.0% forget size

### I.3.2 ALLCNN ON CIFAR-10

| **Unlearner** | Utility | Unlearning Success |
|---|---|---|
| Original | $84.912 \pm 0.928$ | $40.000 \pm 40.000$ |
| Naive | $85.305 \pm 0.345$ | $36.000 \pm 19.391$ |
| SPA | $72.942 \pm 0.682$ | $88.000 \pm 8.000$ |
| SPARC | $76.618 \pm 0.530$ | $100.000 \pm 0.000$ |
| EUk | $78.357 \pm 0.101$ | $68.000 \pm 16.248$ |
| CFk | $83.900 \pm 0.168$ | $24.000 \pm 16.000$ |
| SCRUB | $85.622 \pm 0.061$ | $56.000 \pm 20.396$ |
| SSD | $81.048 \pm 2.417$ | $28.000 \pm 14.967$ |

Table 65: Gradient Matching results for AllCNN on CIFAR-10 with 50.0% forget size

| **Unlearner** | Utility | Unlearning Success |
|---|---|---|
| Original | $84.912 \pm 0.928$ | $40.000 \pm 40.000$ |
| Naive | $85.050 \pm 0.173$ | $28.000 \pm 19.596$ |
| SPA | $75.798 \pm 0.530$ | $88.000 \pm 8.000$ |
| SPARC | $77.358 \pm 0.492$ | $88.000 \pm 8.000$ |
| EUk | $77.384 \pm 0.427$ | $80.000 \pm 12.649$ |
| CFk | $78.793 \pm 0.349$ | $48.000 \pm 16.248$ |
| SCRUB | $86.196 \pm 0.121$ | $16.000 \pm 16.000$ |
| SSD | $83.844 \pm 0.515$ | $28.000 \pm 18.547$ |

Table 66: Gradient Matching results for AllCNN on CIFAR-10 with 80.0% forget size

| **Unlearner** | Utility | Unlearning Success |
|---|---|---|
| Original | $84.912 \pm 0.928$ | $40.000 \pm 40.000$ |
| Naive | $85.805 \pm 0.122$ | $96.000 \pm 4.000$ |
| SPA | $76.998 \pm 0.511$ | $88.000 \pm 8.000$ |
| SPARC | $80.589 \pm 0.526$ | $84.000 \pm 9.798$ |
| EUk | $81.349 \pm 0.208$ | $92.000 \pm 8.000$ |
| CFk | $81.173 \pm 0.416$ | $88.000 \pm 12.000$ |
| SCRUB | $82.240 \pm 0.284$ | $88.000 \pm 8.000$ |
| SSD | $68.018 \pm 4.985$ | $32.000 \pm 10.198$ |

Table 67: Gradient Matching results for AllCNN on CIFAR-10 with 100.0% forget size

| Unlearner | Utility | Unlearning Success |
|---|---|---|
| Original | $90.180 \pm 0.252$ | $89.578 \pm 0.313$ |
| Naive | $90.200 \pm 0.354$ | $89.689 \pm 0.389$ |
| SPA | $81.420 \pm 0.626$ | $89.489 \pm 0.414$ |
| SPARC | $82.560 \pm 0.634$ | $89.822 \pm 0.147$ |
| EUk | $63.360 \pm 3.730$ | $61.667 \pm 4.263$ |
| CFk | $56.860 \pm 5.405$ | $55.756 \pm 6.463$ |
| SCRUB | $80.040 \pm 1.730$ | $78.800 \pm 1.937$ |
| SSD | $44.800 \pm 17.263$ | $47.667 \pm 17.851$ |

Table 68: Selective Unlearning results for ResNet-18 on Lacuna-10 with 2.0% forget size

| Unlearner | Utility | Unlearning Success |
|---|---|---|
| Original | $90.180 \pm 0.252$ | $89.578 \pm 0.313$ |
| Naive | $90.960 \pm 0.075$ | $90.489 \pm 0.114$ |
| SPA | $80.900 \pm 0.192$ | $89.756 \pm 0.197$ |
| SPARC | $81.380 \pm 0.292$ | $89.956 \pm 0.229$ |
| EUk | $90.460 \pm 0.133$ | $90.022 \pm 0.166$ |
| CFk | $87.200 \pm 0.434$ | $86.556 \pm 0.333$ |
| SCRUB | $87.580 \pm 1.037$ | $86.956 \pm 1.079$ |
| SSD | $54.340 \pm 15.350$ | $60.378 \pm 17.055$ |

Table 69: Selective Unlearning results for ResNet-18 on Lacuna-10 with 10.0% forget size

## I.4 CLASSIFICATION UNLEARNING (SELECTIVE UNLEARNING)

This section presents comprehensive results for classification unlearning experiments with detailed forget and retain set metrics.

### I.4.1 RESNET-18 ON LACUNA-10

| **Unlearner** | Utility | Unlearning Success |
|---|---|---|
| Original | $90.180 \pm 0.252$ | $89.578 \pm 0.313$ |
| Naive | $90.440 \pm 0.392$ | $90.467 \pm 0.340$ |
| SPA | $80.840 \pm 0.209$ | $89.778 \pm 0.246$ |
| SPARC | $81.300 \pm 0.288$ | $89.844 \pm 0.218$ |
| EUk | $75.560 \pm 2.493$ | $75.889 \pm 2.268$ |
| CFk | $85.180 \pm 1.035$ | $85.289 \pm 1.073$ |
| SCRUB | $75.660 \pm 2.156$ | $75.378 \pm 2.400$ |
| SSD | $84.320 \pm 2.357$ | $89.689 \pm 0.254$ |

Table 70: Selective Unlearning results for ResNet-18 on Lacuna-10 with 50.0% forget size

| **Unlearner** | Utility | Unlearning Success |
|---|---|---|
| Original | $90.180 \pm 0.252$ | $89.578 \pm 0.313$ |
| Naive | $89.560 \pm 0.291$ | $91.000 \pm 0.306$ |
| SPA | $80.700 \pm 0.207$ | $89.667 \pm 0.230$ |
| SPARC | $80.840 \pm 0.214$ | $89.756 \pm 0.252$ |
| EUk | $73.560 \pm 7.498$ | $76.022 \pm 7.371$ |
| CFk | $82.240 \pm 0.287$ | $82.933 \pm 0.610$ |
| SCRUB | $82.220 \pm 1.404$ | $82.244 \pm 1.209$ |
| SSD | $20.840 \pm 3.168$ | $23.156 \pm 3.520$ |

Table 71: Selective Unlearning results for ResNet-18 on Lacuna-10 with 80.0% forget size

| **Unlearner** | Utility | Unlearning Success |
|---|---|---|
| Original | $90.180 \pm 0.252$ | $89.578 \pm 0.313$ |
| Naive | $79.360 \pm 1.433$ | $88.178 \pm 1.592$ |
| SPA | $80.520 \pm 0.166$ | $89.467 \pm 0.184$ |
| SPARC | $80.920 \pm 0.287$ | $89.911 \pm 0.319$ |
| EUk | $80.780 \pm 0.301$ | $89.756 \pm 0.334$ |
| CFk | $79.760 \pm 0.178$ | $88.622 \pm 0.198$ |
| SCRUB | $79.600 \pm 0.259$ | $88.444 \pm 0.288$ |
| SSD | $82.560 \pm 2.210$ | $89.244 \pm 0.673$ |

Table 72: Selective Unlearning results for ResNet-18 on Lacuna-10 with 100.0% forget size

| **Unlearner** | Utility | Unlearning Success |
|---|---|---|
| Original | $54.848 \pm 0.606$ | $54.683 \pm 0.586$ |
| Naive | $55.514 \pm 0.387$ | $55.321 \pm 0.380$ |
| SPA | $43.972 \pm 2.846$ | $44.230 \pm 2.791$ |
| SPARC | $53.168 \pm 0.255$ | $53.461 \pm 0.275$ |
| EUk | $55.364 \pm 0.564$ | $55.133 \pm 0.562$ |
| CFk | $51.590 \pm 0.393$ | $51.374 \pm 0.393$ |
| SCRUB | $57.922 \pm 0.405$ | $57.747 \pm 0.408$ |
| SSD | $52.934 \pm 2.369$ | $52.879 \pm 2.246$ |

Table 73: Selective Unlearning results for ResNet-18 on CIFAR-100 with 2.0% forget size

| **Unlearner** | Utility | Unlearning Success |
|---|---|---|
| Original | $54.848 \pm 0.606$ | $54.683 \pm 0.586$ |
| Naive | $55.462 \pm 0.299$ | $55.297 \pm 0.307$ |
| SPA | $47.386 \pm 2.026$ | $47.822 \pm 2.028$ |
| SPARC | $53.222 \pm 0.318$ | $53.695 \pm 0.333$ |
| EUk | $53.512 \pm 0.469$ | $53.370 \pm 0.448$ |
| CFk | $53.818 \pm 0.286$ | $53.653 \pm 0.293$ |
| SCRUB | $57.304 \pm 0.321$ | $57.149 \pm 0.321$ |
| SSD | $44.356 \pm 10.865$ | $44.287 \pm 10.845$ |

Table 74: Selective Unlearning results for ResNet-18 on CIFAR-100 with 10.0% forget size

### I.4.2 RESNET-18 ON CIFAR-100

| Unlearner | Utility | Unlearning Success |
|-----------|---------|--------------------|
| Original | $54.848 \pm 0.606$ | $54.681 \pm 0.584$ |
| Naive | $54.834 \pm 0.260$ | $54.758 \pm 0.261$ |
| SPA | $48.058 \pm 2.757$ | $48.422 \pm 2.742$ |
| SPARC | $53.346 \pm 0.445$ | $53.756 \pm 0.473$ |
| EUk | $28.860 \pm 3.276$ | $28.956 \pm 3.228$ |
| CFk | $27.350 \pm 6.370$ | $27.418 \pm 6.350$ |
| SCRUB | $53.522 \pm 0.334$ | $53.549 \pm 0.327$ |
| SSD | $53.568 \pm 1.741$ | $53.471 \pm 1.652$ |

Table 75: Selective Unlearning results for ResNet-18 on CIFAR-100 with 50.0% forget size

| Unlearner | Utility | Unlearning Success |
|-----------|---------|--------------------|
| Original | $54.848 \pm 0.606$ | $54.683 \pm 0.586$ |
| Naive | $54.848 \pm 0.490$ | $54.925 \pm 0.489$ |
| SPA | $46.064 \pm 3.053$ | $46.424 \pm 3.035$ |
| SPARC | $53.266 \pm 0.488$ | $53.701 \pm 0.493$ |
| EUk | $48.502 \pm 0.265$ | $48.776 \pm 0.251$ |
| CFk | $23.616 \pm 5.016$ | $23.844 \pm 5.063$ |
| SCRUB | $56.990 \pm 0.413$ | $56.925 \pm 0.393$ |
| SSD | $44.158 \pm 10.799$ | $44.202 \pm 10.807$ |

Table 76: Selective Unlearning results for ResNet-18 on CIFAR-100 with 80.0% forget size

| Unlearner | Utility | Unlearning Success |
|-----------|---------|--------------------|
| Original | $54.848 \pm 0.606$ | $54.683 \pm 0.586$ |
| Naive | $54.568 \pm 0.394$ | $55.119 \pm 0.398$ |
| SPA | $49.618 \pm 1.591$ | $49.988 \pm 1.553$ |
| SPARC | $55.082 \pm 0.298$ | $55.501 \pm 0.349$ |
| EUk | $53.288 \pm 0.357$ | $53.826 \pm 0.360$ |
| CFk | $50.646 \pm 0.328$ | $51.158 \pm 0.331$ |
| SCRUB | $58.218 \pm 0.568$ | $58.347 \pm 0.538$ |
| SSD | $44.144 \pm 10.792$ | $44.194 \pm 10.800$ |

Table 77: Selective Unlearning results for ResNet-18 on CIFAR-100 with 100.0% forget size

| Unlearner | Utility | Unlearning Success |
|-----------|---------|--------------------|
| Original | $81.926 \pm 0.331$ | $81.100 \pm 0.295$ |
| Naive | $86.164 \pm 0.250$ | $85.736 \pm 0.310$ |
| SPA | $46.832 \pm 10.056$ | $52.036 \pm 11.173$ |
| SPARC | $62.530 \pm 5.101$ | $69.478 \pm 5.667$ |
| EUk | $84.008 \pm 0.139$ | $83.269 \pm 0.165$ |
| CFk | $77.596 \pm 0.869$ | $76.560 \pm 0.892$ |
| SCRUB | $71.924 \pm 1.866$ | $71.753 \pm 1.854$ |
| SSD | $68.172 \pm 6.443$ | $72.556 \pm 5.657$ |

Table 78: Selective Unlearning results for AllCNN on CIFAR-10 with 2.0% forget size

| Unlearner | Utility | Unlearning Success |
|-----------|---------|--------------------|
| Original | $81.926 \pm 0.331$ | $81.100 \pm 0.295$ |
| Naive | $85.794 \pm 0.190$ | $85.329 \pm 0.140$ |
| SPA | $50.300 \pm 7.942$ | $55.889 \pm 8.825$ |
| SPARC | $65.156 \pm 3.183$ | $72.396 \pm 3.537$ |
| EUk | $83.860 \pm 0.103$ | $83.120 \pm 0.134$ |
| CFk | $79.300 \pm 0.798$ | $77.973 \pm 1.039$ |
| SCRUB | $85.582 \pm 0.141$ | $84.896 \pm 0.117$ |
| SSD | $59.700 \pm 8.654$ | $66.333 \pm 9.616$ |

Table 79: Selective Unlearning results for AllCNN on CIFAR-10 with 10.0% forget size

### I.4.3 ALLCNN ON CIFAR-10

| Unlearner | Utility | Unlearning Success |
|---|---|---|
| Original | $81.926 \pm 0.331$ | $81.100 \pm 0.295$ |
| Naive | $86.118 \pm 0.165$ | $86.042 \pm 0.081$ |
| SPA | $45.956 \pm 7.994$ | $51.062 \pm 8.882$ |
| SPARC | $64.148 \pm 3.426$ | $71.276 \pm 3.807$ |
| EUk | $81.158 \pm 0.822$ | $81.813 \pm 0.645$ |
| CFk | $73.700 \pm 1.551$ | $73.104 \pm 1.573$ |
| SCRUB | $71.228 \pm 0.979$ | $75.704 \pm 0.730$ |
| SSD | $71.784 \pm 1.170$ | $79.727 \pm 1.278$ |

Table 80: Selective Unlearning results for AllCNN on CIFAR-10 with 50.0% forget size

| Unlearner | Utility | Unlearning Success |
|---|---|---|
| Original | $81.926 \pm 0.331$ | $81.100 \pm 0.295$ |
| Naive | $84.172 \pm 0.909$ | $84.880 \pm 1.019$ |
| SPA | $48.950 \pm 7.843$ | $54.389 \pm 8.714$ |
| SPARC | $64.846 \pm 3.230$ | $72.051 \pm 3.588$ |
| EUk | $68.534 \pm 0.372$ | $73.464 \pm 0.363$ |
| CFk | $72.708 \pm 2.632$ | $73.809 \pm 2.462$ |
| SCRUB | $67.518 \pm 1.364$ | $71.076 \pm 1.181$ |
| SSD | $55.942 \pm 10.488$ | $62.158 \pm 11.654$ |

Table 81: Selective Unlearning results for AllCNN on CIFAR-10 with 80.0% forget size

| Unlearner | Utility | Unlearning Success |
|---|---|---|
| Original | $81.926 \pm 0.331$ | $81.100 \pm 0.295$ |
| Naive | $77.566 \pm 0.273$ | $86.184 \pm 0.304$ |
| SPA | $47.698 \pm 8.223$ | $52.998 \pm 9.137$ |
| SPARC | $62.336 \pm 2.415$ | $69.262 \pm 2.684$ |
| EUk | $75.972 \pm 0.101$ | $84.413 \pm 0.112$ |
| CFk | $74.512 \pm 0.167$ | $82.791 \pm 0.186$ |
| SCRUB | $76.562 \pm 0.083$ | $85.069 \pm 0.092$ |
| SSD | $58.864 \pm 9.682$ | $65.404 \pm 10.758$ |

Table 82: Selective Unlearning results for AllCNN on CIFAR-10 with 100.0% forget size

| **Unlearner** | Utility | Unlearning Success |
|---|---|---|
| Original | $93.420 \pm 0.356$ | $92.822 \pm 0.415$ |
| Naive | $92.460 \pm 0.269$ | $92.000 \pm 0.312$ |
| SPA | $69.940 \pm 6.432$ | $75.511 \pm 6.243$ |
| SPARC | $85.740 \pm 1.181$ | $91.422 \pm 0.796$ |
| EUk | $89.100 \pm 0.897$ | $88.378 \pm 0.959$ |
| CFk | $87.500 \pm 2.342$ | $86.911 \pm 2.210$ |
| SCRUB | $93.020 \pm 0.385$ | $92.467 \pm 0.452$ |
| SSD | $81.720 \pm 6.354$ | $86.400 \pm 5.480$ |

Table 83: Selective Unlearning results for AllCNN on Lacuna-10 with 2.0% forget size

| **Unlearner** | Utility | Unlearning Success |
|---|---|---|
| Original | $93.420 \pm 0.356$ | $92.822 \pm 0.415$ |
| Naive | $92.900 \pm 0.335$ | $92.422 \pm 0.309$ |
| SPA | $32.360 \pm 8.347$ | $35.956 \pm 9.274$ |
| SPARC | $64.900 \pm 9.457$ | $72.000 \pm 10.463$ |
| EUk | $92.440 \pm 0.268$ | $91.867 \pm 0.288$ |
| CFk | $70.500 \pm 2.296$ | $69.711 \pm 2.401$ |
| SCRUB | $86.320 \pm 0.852$ | $85.178 \pm 0.968$ |
| SSD | $79.880 \pm 5.819$ | $86.533 \pm 5.615$ |

Table 84: Selective Unlearning results for AllCNN on Lacuna-10 with 10.0% forget size

### I.4.4 AllCNN on Lacuna-10

| Unlearner | Utility | Unlearning Success |
|---|---|---|
| Original | $93.420 \pm 0.356$ | $92.822 \pm 0.415$ |
| Naive | $92.580 \pm 0.218$ | $92.311 \pm 0.191$ |
| SPA | $26.920 \pm 11.956$ | $29.911 \pm 13.284$ |
| SPARC | $55.640 \pm 8.047$ | $61.822 \pm 8.942$ |
| EUk | $64.880 \pm 0.424$ | $70.222 \pm 0.330$ |
| CFk | $66.400 \pm 3.900$ | $67.356 \pm 3.431$ |
| SCRUB | $93.880 \pm 0.414$ | $93.333 \pm 0.447$ |
| SSD | $83.080 \pm 4.417$ | $87.978 \pm 2.830$ |

Table 85: Selective Unlearning results for AllCNN on Lacuna-10 with 50.0% forget size

| Unlearner | Utility | Unlearning Success |
|---|---|---|
| Original | $93.420 \pm 0.356$ | $92.822 \pm 0.415$ |
| Naive | $91.940 \pm 0.320$ | $92.556 \pm 0.333$ |
| SPA | $45.060 \pm 10.732$ | $50.067 \pm 11.924$ |
| SPARC | $77.700 \pm 2.327$ | $86.067 \pm 2.495$ |
| EUk | $75.220 \pm 1.142$ | $83.578 \pm 1.269$ |
| CFk | $68.640 \pm 3.427$ | $71.489 \pm 3.459$ |
| SCRUB | $82.580 \pm 1.479$ | $83.556 \pm 1.849$ |
| SSD | $79.460 \pm 5.290$ | $86.067 \pm 4.874$ |

Table 86: Selective Unlearning results for AllCNN on Lacuna-10 with 80.0% forget size

| Unlearner | Utility | Unlearning Success |
|---|---|---|
| Original | $93.420 \pm 0.356$ | $92.822 \pm 0.415$ |
| Naive | $81.240 \pm 0.982$ | $90.267 \pm 1.091$ |
| SPA | $73.420 \pm 4.169$ | $81.400 \pm 4.676$ |
| SPARC | $84.940 \pm 0.645$ | $92.400 \pm 0.342$ |
| EUk | $83.720 \pm 0.278$ | $93.022 \pm 0.309$ |
| CFk | $83.460 \pm 0.223$ | $92.733 \pm 0.247$ |
| SCRUB | $79.280 \pm 1.057$ | $88.089 \pm 1.175$ |
| SSD | $78.640 \pm 6.071$ | $85.156 \pm 5.806$ |

Table 87: Selective Unlearning results for AllCNN on Lacuna-10 with 100.0% forget size

| **Unlearner** | Utility | Unlearning Success |
|---|---|---|
| Original | $57.112 \pm 0.508$ | $56.978 \pm 0.519$ |
| Naive | $57.168 \pm 0.658$ | $56.980 \pm 0.667$ |
| SPA | $50.438 \pm 2.337$ | $50.632 \pm 2.322$ |
| SPARC | $55.162 \pm 0.961$ | $55.388 \pm 0.925$ |
| EUk | $55.392 \pm 0.305$ | $55.222 \pm 0.327$ |
| CFk | $54.568 \pm 0.515$ | $54.408 \pm 0.509$ |
| SCRUB | $59.306 \pm 0.269$ | $59.127 \pm 0.273$ |
| SSD | $57.112 \pm 0.508$ | $56.980 \pm 0.519$ |

Table 88: Selective Unlearning results for AllCNN on CIFAR-100 with 2.0% forget size

| **Unlearner** | Utility | Unlearning Success |
|---|---|---|
| Original | $57.112 \pm 0.508$ | $56.980 \pm 0.519$ |
| Naive | $56.188 \pm 0.478$ | $56.069 \pm 0.478$ |
| SPA | $53.698 \pm 1.437$ | $54.141 \pm 1.393$ |
| SPARC | $55.560 \pm 0.909$ | $56.014 \pm 0.854$ |
| EUk | $55.158 \pm 0.373$ | $55.004 \pm 0.387$ |
| CFk | $54.038 \pm 0.315$ | $53.913 \pm 0.307$ |
| SCRUB | $58.932 \pm 0.247$ | $58.784 \pm 0.226$ |
| SSD | $49.962 \pm 3.042$ | $49.970 \pm 3.100$ |

Table 89: Selective Unlearning results for AllCNN on CIFAR-100 with 10.0% forget size

### I.4.5 ALLCNN ON CIFAR-100

| Unlearner | Utility | Unlearning Success |
|---|---|---|
| Original | $57.112 \pm 0.508$ | $56.980 \pm 0.519$ |
| Naive | $58.188 \pm 1.178$ | $58.160 \pm 1.185$ |
| SPA | $55.752 \pm 0.745$ | $56.194 \pm 0.724$ |
| SPARC | $56.786 \pm 0.570$ | $57.222 \pm 0.540$ |
| EUk | $53.858 \pm 0.189$ | $53.881 \pm 0.182$ |
| CFk | $51.822 \pm 0.753$ | $51.743 \pm 0.783$ |
| SCRUB | $60.460 \pm 0.123$ | $60.354 \pm 0.122$ |
| SSD | $52.306 \pm 3.247$ | $52.214 \pm 3.250$ |

Table 90: Selective Unlearning results for AllCNN on CIFAR-100 with 50.0% forget size

| Unlearner | Utility | Unlearning Success |
|---|---|---|
| Original | $57.112 \pm 0.508$ | $56.980 \pm 0.519$ |
| Naive | $56.460 \pm 0.369$ | $56.677 \pm 0.370$ |
| SPA | $55.842 \pm 0.679$ | $56.279 \pm 0.658$ |
| SPARC | $56.704 \pm 0.529$ | $57.139 \pm 0.506$ |
| EUk | $53.800 \pm 0.132$ | $54.204 \pm 0.152$ |
| CFk | $42.860 \pm 1.554$ | $43.044 \pm 1.538$ |
| SCRUB | $60.300 \pm 0.173$ | $60.230 \pm 0.174$ |
| SSD | $56.580 \pm 0.543$ | $56.703 \pm 0.512$ |

Table 91: Selective Unlearning results for AllCNN on CIFAR-100 with 80.0% forget size

| Unlearner | Utility | Unlearning Success |
|---|---|---|
| Original | $57.112 \pm 0.508$ | $56.980 \pm 0.519$ |
| Naive | $57.104 \pm 0.691$ | $57.681 \pm 0.698$ |
| SPA | $55.066 \pm 0.828$ | $55.564 \pm 0.824$ |
| SPARC | $57.488 \pm 0.293$ | $57.990 \pm 0.280$ |
| EUk | $57.826 \pm 0.240$ | $58.410 \pm 0.242$ |
| CFk | $56.122 \pm 0.242$ | $56.648 \pm 0.259$ |
| SCRUB | $58.458 \pm 0.153$ | $58.903 \pm 0.139$ |
| SSD | $52.026 \pm 2.606$ | $51.994 \pm 2.648$ |

Table 92: Selective Unlearning results for AllCNN on CIFAR-100 with 100.0% forget size

| Unlearner | Utility | Unlearning Success |
|---|---|---|
| Original | $80.046 \pm 0.449$ | $79.298 \pm 0.461$ |
| SPA | $32.500 \pm 4.777$ | $34.482 \pm 4.740$ |
| SPARC | $32.452 \pm 3.285$ | $35.904 \pm 3.589$ |
| SCRUB | $60.174 \pm 0.853$ | $58.958 \pm 0.954$ |
| SSD | $29.138 \pm 8.143$ | $32.364 \pm 9.045$ |

Table 93: Selective Unlearning results for ViT on CIFAR-10 with 2.0% forget size

| Unlearner | Utility | Unlearning Success |
|---|---|---|
| Original | $80.046 \pm 0.449$ | $79.298 \pm 0.461$ |
| SPA | $44.244 \pm 5.673$ | $45.653 \pm 5.437$ |
| SPARC | $35.010 \pm 2.084$ | $38.764 \pm 2.199$ |
| SCRUB | $43.538 \pm 2.382$ | $43.333 \pm 2.300$ |
| SSD | $43.948 \pm 3.851$ | $48.711 \pm 4.215$ |

Table 94: Selective Unlearning results for ViT on CIFAR-10 with 10.0% forget size

I.4.6  VIT ON CIFAR-10

| Unlearner | Utility | Unlearning Success |
|---|---|---|
| Original | $80.046 \pm 0.449$ | $79.298 \pm 0.461$ |
| SPA | $32.530 \pm 6.119$ | $33.953 \pm 5.962$ |
| SPARC | $36.550 \pm 6.080$ | $39.171 \pm 6.175$ |
| SCRUB | $41.962 \pm 1.479$ | $43.620 \pm 1.448$ |
| SSD | $35.404 \pm 4.277$ | $39.331 \pm 4.748$ |

Table 95: Selective Unlearning results for ViT on CIFAR-10 with 50.0% forget size

| Unlearner | Utility | Unlearning Success |
|---|---|---|
| Original | $80.046 \pm 0.449$ | $79.298 \pm 0.461$ |
| SPA | $33.112 \pm 6.009$ | $34.467 \pm 5.802$ |
| SPARC | $34.290 \pm 4.556$ | $37.718 \pm 4.803$ |
| SCRUB | $73.002 \pm 0.540$ | $74.147 \pm 0.510$ |
| SSD | $42.554 \pm 3.261$ | $47.249 \pm 3.617$ |

Table 96: Selective Unlearning results for ViT on CIFAR-10 with 80.0% forget size

| Unlearner | Utility | Unlearning Success |
|---|---|---|
| Original | $80.046 \pm 0.449$ | $79.298 \pm 0.461$ |
| SPA | $32.362 \pm 5.982$ | $33.784 \pm 5.808$ |
| SPARC | $36.060 \pm 6.161$ | $38.729 \pm 6.245$ |
| SCRUB | $24.828 \pm 1.655$ | $27.587 \pm 1.839$ |
| SSD | $31.148 \pm 6.308$ | $34.607 \pm 7.007$ |

Table 97: Selective Unlearning results for ViT on CIFAR-10 with 100.0% forget size

| Unlearner | Utility | Unlearning Success |
|---|---|---|
| Original | $10.239 \pm 0.077$ | $63.676 \pm 1.576$ |
| Naive | $9.194 \pm 0.039$ | $73.838 \pm 0.238$ |
| SPA | $10.733 \pm 0.193$ | $73.225 \pm 4.007$ |
| SPARC | $10.572 \pm 0.138$ | $68.973 \pm 1.601$ |
| EUk | $10.215 \pm 0.341$ | $64.505 \pm 6.340$ |
| CFk | $11.243 \pm 0.153$ | $67.928 \pm 2.542$ |
| GaussianAmnesiac | $11.227 \pm 0.113$ | $61.730 \pm 2.277$ |
| Blindspot | $10.264 \pm 0.059$ | $63.964 \pm 0.839$ |

Table 98: Ages 0-30 results for ResNet-18 on AgeDB with 2.0% forget size

| Unlearner | Utility | Unlearning Success |
|---|---|---|
| Original | $10.239 \pm 0.077$ | $63.676 \pm 1.576$ |
| Naive | $9.161 \pm 0.059$ | $74.342 \pm 2.188$ |
| SPA | $11.026 \pm 0.356$ | $78.162 \pm 4.912$ |
| SPARC | $10.441 \pm 0.114$ | $67.099 \pm 1.489$ |
| EUk | $10.942 \pm 0.181$ | $69.441 \pm 6.128$ |
| CFk | $11.398 \pm 0.289$ | $74.486 \pm 5.599$ |
| GaussianAmnesiac | $10.292 \pm 0.058$ | $65.910 \pm 1.445$ |
| Blindspot | $10.279 \pm 0.074$ | $65.874 \pm 1.257$ |

Table 99: Ages 0-30 results for ResNet-18 on AgeDB with 10.0% forget size

## I.5 REGRESSION UNLEARNING (AGES 0-30)

This section presents results for regression unlearning on the AgeDB dataset for age range 0-30.

### I.5.1 RESNET-18 ON AGEDB

| Unlearner | Utility | Unlearning Success |
|---|---|---|
| Original | $10.239 \pm 0.077$ | $63.676 \pm 1.576$ |
| Naive | $9.555 \pm 0.062$ | $88.541 \pm 1.095$ |
| SPA | $11.894 \pm 0.428$ | $86.342 \pm 5.001$ |
| SPARC | $10.458 \pm 0.140$ | $65.189 \pm 1.344$ |
| EUk | $10.291 \pm 0.098$ | $73.622 \pm 2.994$ |
| CFk | $10.574 \pm 0.070$ | $69.910 \pm 4.610$ |
| GaussianAmnesiac | $10.224 \pm 0.061$ | $63.712 \pm 1.369$ |
| Blindspot | $10.278 \pm 0.075$ | $68.468 \pm 1.941$ |

Table 100: Ages 0-30 results for ResNet-18 on AgeDB with 50.0% forget size

| Unlearner | Utility | Unlearning Success |
|---|---|---|
| Original | $10.239 \pm 0.077$ | $63.676 \pm 1.576$ |
| Naive | $9.857 \pm 0.087$ | $95.820 \pm 0.978$ |
| SPA | $10.843 \pm 0.190$ | $77.946 \pm 3.756$ |
| SPARC | $10.448 \pm 0.114$ | $64.613 \pm 3.006$ |
| EUk | $10.497 \pm 0.210$ | $93.874 \pm 1.783$ |
| CFk | $11.311 \pm 0.315$ | $84.108 \pm 2.681$ |
| GaussianAmnesiac | $10.302 \pm 0.093$ | $72.036 \pm 3.119$ |
| Blindspot | $10.453 \pm 0.101$ | $80.649 \pm 3.680$ |

Table 101: Ages 0-30 results for ResNet-18 on AgeDB with 80.0% forget size

| Unlearner | Utility | Unlearning Success |
|---|---|---|
| Original | $10.239 \pm 0.077$ | $63.676 \pm 1.576$ |
| Naive | $10.374 \pm 0.036$ | $99.964 \pm 0.036$ |
| SPA | $11.115 \pm 0.233$ | $79.820 \pm 4.357$ |
| SPARC | $10.513 \pm 0.109$ | $63.604 \pm 3.858$ |
| EUk | $10.357 \pm 0.099$ | $71.279 \pm 3.137$ |
| CFk | $11.918 \pm 0.277$ | $99.928 \pm 0.072$ |
| GaussianAmnesiac | $10.365 \pm 0.052$ | $72.685 \pm 2.928$ |
| Blindspot | $10.357 \pm 0.101$ | $70.667 \pm 2.900$ |

Table 102: Ages 0-30 results for ResNet-18 on AgeDB with 100.0% forget size

| Unlearner | Utility | Unlearning Success |
|---|---|---|
| Original | $9.796 \pm 0.078$ | $66.667 \pm 1.011$ |
| Naive | $9.907 \pm 0.079$ | $71.027 \pm 3.434$ |
| SPA | $20.178 \pm 2.202$ | $99.928 \pm 0.072$ |
| SPARC | $13.547 \pm 1.202$ | $56.685 \pm 14.373$ |
| EUk | $9.776 \pm 0.065$ | $66.306 \pm 0.975$ |
| CFk | $10.242 \pm 0.445$ | $62.018 \pm 6.435$ |
| GaussianAmnesiac | $9.827 \pm 0.063$ | $66.450 \pm 1.088$ |
| Blindspot | $10.714 \pm 0.176$ | $73.405 \pm 8.980$ |

Table 103: Ages 0-30 results for AllCNN on AgeDB with 2.0% forget size

| Unlearner | Utility | Unlearning Success |
|---|---|---|
| Original | $9.796 \pm 0.078$ | $66.667 \pm 1.011$ |
| Naive | $9.917 \pm 0.093$ | $72.901 \pm 1.622$ |
| SPA | $26.341 \pm 4.298$ | $100.000 \pm 0.000$ |
| SPARC | $15.526 \pm 1.846$ | $28.468 \pm 11.776$ |
| EUk | $9.772 \pm 0.053$ | $69.333 \pm 0.545$ |
| CFk | $9.824 \pm 0.078$ | $67.604 \pm 1.014$ |
| GaussianAmnesiac | $9.840 \pm 0.096$ | $74.162 \pm 1.442$ |
| Blindspot | $9.790 \pm 0.052$ | $69.441 \pm 0.626$ |

Table 104: Ages 0-30 results for AllCNN on AgeDB with 10.0% forget size

### I.5.2 ALLCNN ON AGEDB

| Unlearner | Utility | Unlearning Success |
|---|---|---|
| Original | 9.796 ± 0.078 | 66.667 ± 1.011 |
| Naive | 9.999 ± 0.043 | 78.991 ± 1.534 |
| SPA | 9.829 ± 0.070 | 66.090 ± 0.991 |
| SPARC | 12.719 ± 0.528 | 87.604 ± 6.329 |
| EUk | 10.543 ± 0.232 | 91.604 ± 4.165 |
| CFk | 9.902 ± 0.070 | 77.009 ± 1.675 |
| GaussianAmnesiac | 10.225 ± 0.077 | 94.450 ± 0.542 |
| Blindspot | 9.801 ± 0.076 | 67.279 ± 0.801 |

Table 105: Ages 0-30 results for AllCNN on AgeDB with 50.0% forget size

| Unlearner | Utility | Unlearning Success |
|---|---|---|
| Original | 9.796 ± 0.078 | 66.667 ± 1.011 |
| Naive | 10.450 ± 0.109 | 93.009 ± 1.509 |
| SPA | 25.103 ± 3.138 | 100.000 ± 0.000 |
| SPARC | 12.899 ± 0.530 | 54.991 ± 8.461 |
| EUk | 10.925 ± 0.214 | 84.577 ± 4.275 |
| CFk | 11.422 ± 0.541 | 85.261 ± 5.328 |
| GaussianAmnesiac | 10.410 ± 0.054 | 97.910 ± 0.409 |
| Blindspot | 9.771 ± 0.060 | 65.658 ± 0.695 |

Table 106: Ages 0-30 results for AllCNN on AgeDB with 80.0% forget size

| Unlearner | Utility | Unlearning Success |
|---|---|---|
| Original | 9.796 ± 0.078 | 66.667 ± 1.011 |
| Naive | 10.700 ± 0.126 | 98.955 ± 0.527 |
| SPA | 19.891 ± 3.404 | 99.027 ± 0.973 |
| SPARC | 16.493 ± 1.980 | 96.180 ± 1.065 |
| EUk | 11.219 ± 0.219 | 99.964 ± 0.036 |
| CFk | 12.118 ± 0.656 | 97.622 ± 2.115 |
| GaussianAmnesiac | 11.509 ± 0.034 | 99.495 ± 0.105 |
| Blindspot | 9.873 ± 0.067 | 62.631 ± 0.803 |

Table 107: Ages 0-30 results for AllCNN on AgeDB with 100.0% forget size

| Unlearner | Utility | Unlearning Success |
|---|---|---|
| Original | $10.239 \pm 0.077$ | $48.918 \pm 1.120$ |
| Naive | $9.222 \pm 0.066$ | $47.514 \pm 2.327$ |
| SPA | $12.551 \pm 0.967$ | $85.123 \pm 6.168$ |
| SPARC | $12.072 \pm 0.895$ | $70.702 \pm 9.945$ |
| EUk | $10.724 \pm 0.217$ | $56.926 \pm 4.093$ |
| CFk | $11.521 \pm 0.891$ | $46.603 \pm 8.837$ |
| GaussianAmnesiac | $10.715 \pm 0.163$ | $48.235 \pm 1.842$ |
| Blindspot | $10.311 \pm 0.070$ | $48.235 \pm 0.514$ |

Table 108: Ages 60-101 results for ResNet-18 on AgeDB with 2.0% forget size

| Unlearner | Utility | Unlearning Success |
|---|---|---|
| Original | $10.239 \pm 0.077$ | $48.918 \pm 1.120$ |
| Naive | $9.278 \pm 0.061$ | $46.717 \pm 2.765$ |
| SPA | $17.622 \pm 2.296$ | $99.734 \pm 0.186$ |
| SPARC | $16.996 \pm 3.095$ | $94.004 \pm 3.700$ |
| EUk | $10.241 \pm 0.129$ | $44.820 \pm 2.434$ |
| CFk | $10.260 \pm 0.103$ | $47.742 \pm 0.852$ |
| GaussianAmnesiac | $11.184 \pm 0.350$ | $51.082 \pm 5.949$ |
| Blindspot | $10.319 \pm 0.053$ | $49.184 \pm 1.383$ |

Table 109: Ages 60-101 results for ResNet-18 on AgeDB with 10.0% forget size

## I.6 REGRESSION UNLEARNING (AGES 60-101)

This section presents results for regression unlearning on the AgeDB dataset for age range 60-101.

### I.6.1 RESNET-18 ON AGEDB

| Unlearner | Utility | Unlearning Success |
|---|---|---|
| Original | $10.239 \pm 0.077$ | $48.918 \pm 1.120$ |
| Naive | $9.530 \pm 0.072$ | $58.710 \pm 1.572$ |
| SPA | $14.258 \pm 1.454$ | $89.905 \pm 6.091$ |
| SPARC | $12.904 \pm 1.175$ | $83.795 \pm 8.163$ |
| EUk | $10.383 \pm 0.116$ | $48.880 \pm 4.523$ |
| CFk | $10.403 \pm 0.110$ | $60.380 \pm 4.122$ |
| GaussianAmnesiac | $10.290 \pm 0.078$ | $52.258 \pm 1.533$ |
| Blindspot | $10.480 \pm 0.102$ | $60.797 \pm 2.940$ |

Table 110: Ages 60-101 results for ResNet-18 on AgeDB with 50.0% forget size

| Unlearner | Utility | Unlearning Success |
|---|---|---|
| Original | $10.239 \pm 0.077$ | $48.918 \pm 1.120$ |
| Naive | $10.149 \pm 0.113$ | $74.345 \pm 2.756$ |
| SPA | $13.113 \pm 1.268$ | $82.239 \pm 8.696$ |
| SPARC | $12.637 \pm 0.891$ | $74.801 \pm 9.554$ |
| EUk | $15.636 \pm 1.478$ | $95.484 \pm 2.689$ |
| CFk | $11.056 \pm 0.484$ | $73.890 \pm 7.363$ |
| GaussianAmnesiac | $11.918 \pm 0.149$ | $82.581 \pm 3.036$ |
| Blindspot | $11.459 \pm 0.165$ | $96.698 \pm 1.580$ |

Table 111: Ages 60-101 results for ResNet-18 on AgeDB with 80.0% forget size

| Unlearner | Utility | Unlearning Success |
|---|---|---|
| Original | $10.239 \pm 0.077$ | $48.918 \pm 1.120$ |
| Naive | $11.183 \pm 0.064$ | $98.748 \pm 0.266$ |
| SPA | $12.810 \pm 1.014$ | $82.163 \pm 8.656$ |
| SPARC | $12.735 \pm 0.808$ | $84.782 \pm 5.728$ |
| EUk | $11.023 \pm 0.093$ | $95.712 \pm 1.172$ |
| CFk | $11.639 \pm 0.241$ | $98.975 \pm 0.348$ |
| GaussianAmnesiac | $11.351 \pm 0.306$ | $86.755 \pm 6.437$ |
| Blindspot | $12.093 \pm 0.191$ | $99.924 \pm 0.076$ |

Table 112: Ages 60-101 results for ResNet-18 on AgeDB with 100.0% forget size

| Unlearner | Utility | Unlearning Success |
|---|---|---|
| Original | 9.796 ± 0.078 | 48.387 ± 1.483 |
| Naive | 9.799 ± 0.079 | 49.715 ± 1.745 |
| SPA | 19.228 ± 1.783 | 99.924 ± 0.076 |
| SPARC | 16.946 ± 1.735 | 99.431 ± 0.569 |
| EUk | 10.499 ± 0.204 | 51.537 ± 6.930 |
| CFk | 10.468 ± 0.116 | 52.638 ± 4.272 |
| GaussianAmnesiac | 10.160 ± 0.087 | 49.791 ± 4.860 |
| Blindspot | 9.849 ± 0.058 | 50.664 ± 0.933 |

Table 113: Ages 60-101 results for AllCNN on AgeDB with 2.0% forget size

| Unlearner | Utility | Unlearning Success |
|---|---|---|
| Original | 9.796 ± 0.078 | 48.387 ± 1.483 |
| Naive | 9.937 ± 0.079 | 52.296 ± 0.942 |
| SPA | 20.993 ± 2.203 | 100.000 ± 0.000 |
| SPARC | 19.805 ± 2.075 | 99.924 ± 0.076 |
| EUk | 11.085 ± 0.461 | 42.049 ± 10.448 |
| CFk | 10.029 ± 0.127 | 47.287 ± 4.322 |
| GaussianAmnesiac | 10.164 ± 0.115 | 54.687 ± 1.457 |
| Blindspot | 9.809 ± 0.049 | 51.195 ± 0.843 |

Table 114: Ages 60-101 results for AllCNN on AgeDB with 10.0% forget size

### I.6.2 ALLCNN ON AGEDB

| Unlearner | Utility | Unlearning Success |
|---|---|---|
| Original | 9.796 ± 0.078 | 48.387 ± 1.483 |
| Naive | 10.261 ± 0.068 | 62.087 ± 1.769 |
| SPA | 19.836 ± 2.165 | 99.772 ± 0.228 |
| SPARC | 18.182 ± 2.172 | 99.279 ± 0.721 |
| EUk | 10.384 ± 0.061 | 52.979 ± 2.267 |
| CFk | 9.770 ± 0.062 | 49.336 ± 0.411 |
| GaussianAmnesiac | 10.317 ± 0.085 | 78.899 ± 0.944 |
| Blindspot | 12.956 ± 0.280 | 99.848 ± 0.152 |

Table 115: Ages 60-101 results for AllCNN on AgeDB with 50.0% forget size

| Unlearner | Utility | Unlearning Success |
|---|---|---|
| Original | 9.796 ± 0.078 | 48.387 ± 1.483 |
| Naive | 10.804 ± 0.140 | 75.142 ± 1.843 |
| SPA | 20.113 ± 2.205 | 99.924 ± 0.076 |
| SPARC | 18.234 ± 2.315 | 99.317 ± 0.683 |
| EUk | 10.627 ± 0.131 | 51.992 ± 3.150 |
| CFk | 9.831 ± 0.080 | 49.715 ± 0.312 |
| GaussianAmnesiac | 11.357 ± 0.100 | 97.495 ± 0.639 |
| Blindspot | 13.573 ± 0.316 | 99.772 ± 0.111 |

Table 116: Ages 60-101 results for AllCNN on AgeDB with 80.0% forget size

| Unlearner | Utility | Unlearning Success |
|---|---|---|
| Original | 9.796 ± 0.078 | 48.387 ± 1.483 |
| Naive | 11.757 ± 0.130 | 95.294 ± 0.302 |
| SPA | 20.839 ± 2.237 | 100.000 ± 0.000 |
| SPARC | 18.488 ± 2.431 | 99.355 ± 0.645 |
| EUk | 11.060 ± 0.138 | 92.220 ± 3.675 |
| CFk | 11.449 ± 0.191 | 94.991 ± 1.461 |
| GaussianAmnesiac | 11.907 ± 0.027 | 99.772 ± 0.111 |
| Blindspot | 15.522 ± 0.255 | 99.886 ± 0.114 |

Table 117: Ages 60-101 results for AllCNN on AgeDB with 100.0% forget size

