# OpenReview forum: "Sample Efficient Corrective Deep Unlearning"
_ICLR.cc/2026/Conference — Submitted to ICLR 2026_

### Official Review · Reviewer_S1KG · 2025-10-31

**Soundness:** 3
**Presentation:** 2
**Contribution:** 2
**Rating:** 4
**Confidence:** 4

**Summary:**

This paper focuses on corrective machine unlearning, where the objective is to restore a model’s performance by removing corrupted or poisoned training data, rather than pursuing privacy-oriented unlearning. The authors propose Selective Parameter Adjustment and ReCalibration (SPARC), a two-stage unlearning framework. In the first stage, SPARC selectively adjusts model parameters based on an activation-difference influence measure; in the second stage, it recalibrates low-importance parameters using orthogonal gradient descent inspired by PCGrad. Experiments cover five unlearning scenarios—backdoor attacks, label confusion, data poisoning, classification errors, and regression tasks—across multiple datasets (CIFAR-10/100, Lacuna-10, AgeDB) and architectures (ResNet-18, AllCNN, ViT-Tiny).

**Strengths:**

1. The method works across CNNs, ResNets, and Transformers without architecture-specific modifications, demonstrating good generalizability.

2. The paper covers five distinct corrective unlearning scenarios with consistent evaluation across multiple datasets and architectures, providing strong empirical evidence.

3. The proposed method demonstrates robust performance with forget set sizes as low as 2%.

**Weaknesses:**

1. The paper claims to be “sample efficient,” yet both stages require access to the entire retain set $D_r$. For instance, Eq. (1) explicitly computes activation differences between $D_f$ and $D_r$, and Eq. (5) requires gradients from both. The main concern lies in the use of $D_r$, which, according to the paper’s definition, corresponds to $D_{\text{train}} / D_u$. This potentially dominates the computational cost, contradicting the claim of sample efficiency. This issue is not sufficiently discussed in the paper.

2. The paper lacks a systematic analysis of individual components—such as $|\theta|$, incoming activations, outgoing activations—and the designed hyperparameter $\tau$. More detailed ablations would help clarify the contribution of each element to the overall performance.

3. Missing discussion and comparison with related works on corrective unlearning, particularly [1,2]. If my understanding is correct, the “SAP” mentioned in the experiments does not refer to the existing method [2], which may cause confusion. A more explicit comparison and discussion would strengthen the paper.

[1] Wei S, Zhang M, Zha H, et al. Shared adversarial unlearning: Backdoor mitigation by unlearning shared adversarial examples. NeurIPS 2023.

[2] Kodge S, Ravikumar D, Saha G, et al. SAP: Corrective Machine Unlearning with Scaled Activation Projection for Label Noise Robustness. AAAI 2025.

**Questions:**

1. What is the actual wall-clock time breakdown between $D_f$ and $D_r$ processing?

2. Can the method work with a random subset of $D_r$ (e.g., 10%)? How does $|D_r|$ affect unlearning performance?

---

> ### Author Response · Authors · 2025-11-21
> **Rebuttal (Part-1)**
>
> > W1: The paper claims to be “sample efficient,” yet both stages require access to the entire retain set $D_r$. For instance, Eq. (1) explicitly computes activation differences between $D_f$ and $D_r$, and Eq. (5) requires gradients from both. The main concern lies in the use of $D_r$, which, according to the paper’s definition, corresponds to $D_{\text{train}} / D_u$. This potentially dominates the computational cost, contradicting the claim of sample efficiency. This issue is not sufficiently discussed in the paper.
>
> Our intention with “sample-efficient” is not that SPARC never touches the retain set, but that it (i) achieves strong corrective unlearning from a small forget set and (ii) uses a small unlearning budget (in epochs and FLOPs) compared to retraining and prior methods. In all experiments, SPA uses a single epoch of forward passes on D_f and D_r plus a closed-form parameter update, and SPARC adds one extra epoch of recalibration. All competing methods are run for 10 unlearning epochs under the same training setup. To make this precise, we have added a FLOPs-based cost analysis in the appendix of the rebuttal revision.
>
> > W2: The paper lacks a systematic analysis of individual components—such as $|\theta|$, incoming activations, outgoing activations—and the designed hyperparameter $\tau$. More detailed ablations would help clarify the contribution of each element to the overall performance.
>
> We agree that more fine-grained ablations would be informative. In the current version, our main robustness study is the 20×20 grid over the two key hyperparameters \gamma (forgetting factor) and \tau (influence threshold) on CIFAR-10/ResNet-18 (Sec. 5.1, Fig. 7). This shows broad regions where SPARC simultaneously achieves high unlearning success and utility, and smooth trade-offs rather than brittle behavior, indicating that the method is not overly sensitive to these choices.
>
> Regarding the internal components (incoming vs outgoing activations, etc.), we did explore straightforward variants (e.g., using only incoming or only outgoing activations, or alternative simple choices) and found that our final design, using both incoming and outgoing activations with the chosen contrast, consistently gave the most stable performance across architectures and settings. Due to computational and space constraints, we prioritized breadth of settings (datasets, architectures, and corruption types) and the 2D (\gamma, \tau) sensitivity over an exhaustive micro-ablation grid. We view a more detailed component-wise study as a natural future work.
>
> > W3: Missing discussion and comparison with related works on corrective unlearning, particularly [1,2]. If my understanding is correct, the “SAP” mentioned in the experiments does not refer to the existing method [2], which may cause confusion. A more explicit comparison and discussion would strengthen the paper.
>
> **Relation to [1]:** [1] is tailored specifically to backdoor mitigation by unlearning shared adversarial examples. Unlike [1] our work follows the broader corrective unlearning framework: we handle backdoors, gradient-matching poisons, label and feature removal, class removal, and regression-range removal across CNNs, ResNets, and ViTs. SPARC operates at the parameter level via influence-based damping plus orthogonal recalibration and is applied unchanged across all these scenarios.
>
> **Relation to [2]:** In our paper, SAP [2] is only mentioned in Related Work, exactly as the SAP method you cite, and we position it as a method for mitigating label noise via projection techniques. Similar to [1], SAP also does not focus on complete corrective unlearning and cannot be considered a related work for corrective unlearning.
>
> > Q1: What is the actual wall-clock time breakdown between $D_f$ and $D_r$ processing?
>
> Although we do not have actual wall-clock times, implementation-wise, we operate on mini-batches drawn from D_f and D_r, and perform the same operations on the samples from each set. The only difference in wall-clock times would depend on the size of each set.

---

> > ### Author Response · Authors · 2025-11-21
> > **Rebuttal (Part-2)**
> >
> > > Q2: Can the method work with a random subset of $D_r$ (e.g., 10%)? How does $|D_r|$ affect unlearning performance?
> >
> > Yes, SPARC can, in principle, be applied with a random subset of the retain set:
> >  - In SPA, the influence in Eq. (1) depends on empirical means of activations over D_f and D_r; these means can be estimated using any representative subset \tilde{D}_r \subseteq D_r
> > - In ReCalibration, we already rely on mini-batch SGD, so each update step only sees a small subset of D_r at a time.
> >
> > In our reported experiments, we use the full retain set in the standard mini-batch way rather than an explicit fixed subsampling ratio (e.g., exactly 10%), because our main goal was to compare algorithms under a common protocol and unlearning budget. Exploring the trade-off between retain-set subsampling and performance is an interesting extension, but we view it as orthogonal to the main contribution and therefore outside the scope of the current work.

---

### Official Review · Reviewer_fV2S · 2025-10-31

**Soundness:** 2
**Presentation:** 3
**Contribution:** 2
**Rating:** 2
**Confidence:** 4

**Summary:**

This paper tackles “corrective unlearning”: a setting where we discover that part of the training data is "bad" (corrupted, malicious, mislabeled, etc), and we want to remove its influence from the model without a full retrain. In this setting the unlearning algorithm is given only a fraction of 'bad' examples and the algorithm aims to remove the influence of all without damaging utility.

The proposed approach, SPARC, works in two stages: (i) it estimates which internal parameters are disproportionately important by the bad examples compared to clean ones (looking at activations along forward paths through the network). and (ii) SPARC then fine-tunes on clean examples trying to avoid reintroducing the bad behavior.

**Strengths:**

1. the problem studied is practical/realistic and interesting and differs compared to solutions for 'traditional' unlearning methods.

2. Their two-phase approach makes sense (removing influence and recalibrating to ensure utility).

3. The paper aims to address and evaluates against multiple tasks (classification, regression-like settings, sequential unlearning), and reports both unlearning and utility performance.

**Weaknesses:**

1. A major weakness is that the work makes an **implicit assumption that the bad data/behavior (and their pathways) pathways are clearly/markedly separable from acceptable data/behaviour.** It is only then that their method will not hurt model utility, right?.
This assumption (if correct) should be:

(i) clearly **spelled-out** in their paper;

(ii) **analyzed**, w.r.t. to which target applications this assumption makes more or less sense, eg:

- for backdoor attacks, this probably makes sense;

- for targeted poisoning attacks, this may make less sense;

- for inter-class confusion cases, where the “bad” signal can overlap (correct) decision boundaries, one could argue that this assumption does not make sense;

(iii) **test explicitly** the above (and maybe other) cases, quantifying possible utility loss for each.

2. There are serious misgivings about the paper as it does not discuss, analyze, and quantify performance **based on how representative is and on the coverage of the given forget set**. One would have expected that this is studied in detail, showing that things like:

(i) the **representativeness**: the cardinality of the forget set is important, ie may be too small for being representative of the *exact* set of examples to be unlearned. Their results on varying the size of the forget set are valuable. But, is it not preferable to experiment, say with forget sets activating pathways which are used by 'normal' data?

(ii) the **coverage** of the original forget set versus that of the full (forget) set to be unlearned.

Basically, if the starting point (forget set) is not "good enough" (wrt coverage or representativeness), the method may end up either hurting 'forget performance' or model utility.

**1+2 above negate the selling of the new method** that says "Give me a forget set and I will deliver true forgetting and utility for all bad exampels, and for all applications of bad data/behavior (such as backdoor attacks, data poisoning, class confusion etc.).

3. Another key concern centers around the set of chosen **baselines**. These are rather weak and do not represent SOTA methods.

- I would strongly urge the authors to consider using the Salun method (https://arxiv.org/abs/2310.12508), especially since it is also based on defining **saliency** (albeit differently) of examples, which drives unlearning.

- The authors appear to be unaware of a new SOTA method, called RUM (https://arxiv.org/abs/2406.01257) and its companion
(https://arxiv.org/abs/2410.16516) which actually can improve performance substantially. **RUM is especially relevant here as one may intuit that the separability of learning/activation pathways (see 1 and 2 above) depends on the degree of memorization of examples**. And this is what RUM leverages in order to improve both forgetting performance and utility. For example, for low-memorized examples in the forget set it does nothing; for medium ones and for highly-memorized ones, it uses different unlearning algorithms...

4. One key issue that is also overlooked is that the work seems to not address the possibility that, after their recalibration step (to maintain utility), their **method may be inadvertently reintroducing "bad" behaviour.** **Your metric meant to appease us of this possibility is an average**, missing the possibility that **some bad behavior (associations) may have been reintroduced**. So, for safety and compliance, knowing whether this is a failure mode and to what extent is necessary.

5. it is not clear if baselines were afforded the same "budget" for hyper-parameter tuning as the proposed method (for which it appears that it is very large).

**Questions:**

Please address all above weaknesses.

The paper is lacking and should provide:
1. A study where the full set of corrupted samples, S_c is fixed. Then they could subsample from S_c forming the provided forget set, S. Then they can quantify how well unlearning "generalizes" to the rest of S_c (not just to S).

2. Measurements of accuracy/utility specifically on S_rel (set of retain samples that are closest (in their activation pathways) to the provided forget examples, before vs after unlearning.

3. The paper should study better baselines (eg SalUn and RUM) and whether (i) memorization affects their separabiloty (assumption) and (ii) how the performance of their method compares against SalUn and RUM. In my view both SalUn and RUM are especially relevant.

4. Study/analyze the possibility for "recontamination" - especially important as the work is about corrective learning.

---

> ### Author Response · Authors · 2025-11-21
> **Rebuttal (Part-1)**
>
> > W1: “Implicit assumption that bad vs acceptable pathways are clearly separable”
>
> Our method does not assume that “bad behaviour (and their pathways) are clearly / markedly separable from acceptable behaviour,” nor do we claim that SPARC guarantees no utility loss whenever this fails.
>
> What SPARC uses is a relative notion of influence: for each parameter, we compare its activations on the forget set and the retain set, and rank parameters by how disproportionately they support the forget set. This is a contrastive measure (forget vs retain), not a hard assumption that the underlying pathways are disjoint.
>
> We also explicitly study settings where the distinction between “bad” and “normal” behaviour is not cleanly separable:
> - In the interclass confusion / label-only manipulations, the corrupted and clean examples share highly overlapping decision boundaries and activation patterns, and SPARC is still competitive with naïve retraining in terms of both unlearning success and utility.
> - In targeted poisoning scenarios, we similarly operate in a regime where the corrupted examples are deliberately crafted to live close to the clean manifold.
>
> In these more entangled cases, we do not claim perfect forgetting with zero utility loss; rather, our empirical claim is that SPARC improves the unlearning–utility trade-off compared to existing unlearning methods given the same budget.
>
> > W2: Representativeness and coverage of the forget set
>
> We agree that representativeness and coverage of the forget set are central issues in corrective unlearning. This is exactly why we:
> - Vary the forget-set size over a wide range (down to very small fractions of the corrupted pool) and track both unlearning success and utility, and
> - Evaluate across multiple corruption types where the forget set is inherently “entangled” with the retain distribution (backdoors, gradient-matching poisons, interclass confusion, regression range removal).
>
> These experiments are designed to probe the regime where the forget set is small and potentially non-representative of all corrupted behaviour.
>
> Our intended “selling point” is therefore not that “Give me a forget set and I will deliver true forgetting and utility for all bad exampels, and for all applications of bad data/behavior (such as backdoor attacks, data poisoning, class confusion etc.)” but rather that Given a limited and potentially incomplete forget set, SPARC achieves strong corrective unlearning with a small unlearning budget, and with a better utility trade-off than existing methods.
>
> The more detailed coverage analysis you suggest, by using “forget sets activating pathways which are used by 'normal' data”, is indeed interesting. However, finding a forget set like that is another challenge that falls outside the scope of this work.
>
> > W3: Baselines and missing SOTA (SalUn, RUM)
>
> Our baseline selection was guided by two criteria:
> - Methods that are commonly used in the unlearning literature
> - Methods that have been previously considered (and sometimes shown to struggle) in corrective-style settings, such as SCRUB, SSD, Gaussian-Amnesiac, Blindspot, and CF_k/EU_k.
>
> SalUn and RUM are both relevant and complementary, but they target somewhat different design axes:
> - SalUn is a saliency-based unlearning method that has mainly been evaluated on more standard classification-unlearning benchmarks, rather than on the broader corrective tasks we consider (backdoors, gradient-matching poisons, interclass confusion, regression).
> - RUM and its extension focus on memorization-aware unlearning policies, using memorization scores to decide how aggressively to unlearn a given example. This is orthogonal to our parameter-level influence routing: SPARC operates on parameter influence via activation contrasts, not on per-example memorization scores.
>
> Due to compute and page constraints, we could not include every recent method on every dataset. Our goal in this work is to isolate the effect of activation-path-based parameter routing for corrective unlearning, rather than to provide a comprehensive empirical ranking of all unlearning algorithms.

---

> > ### Author Response · Authors · 2025-11-21
> > **Rebuttal (Part-2)**
> >
> > > W4: Risk of “recontamination” after recalibration
> >
> > > Q4: Study/analyze the possibility for "recontamination" - especially important as the work is about corrective learning.
> >
> > We agree that, in principle, any post-forgetting fine-tuning step must be checked for the risk of reintroducing undesired behaviour.
> >
> > Two aspects of SPARC are designed specifically to mitigate this:
> > - Structure of the recalibration step: The recalibration updates are restricted to low-influence parameters and are projected orthogonally to the forget-set gradient directions. This means that parameters most responsible for the forget set are damped and then not used for utility recovery, and the directions used to fit the forget set are explicitly treated as “do-not-revisit” directions.
> > - Evaluation protocol: For the corrective scenarios where “bad behaviour” is sharp and measurable:
> >   - In backdoor and targeted poisoning experiments, we evaluate attack success on appropriately triggered inputs; any recontamination would directly increase backdoor success.
> >   - For classification / regression removal, our unlearning metrics are computed specifically on designated unlearn test sets D_{\text{test},u}, where reintroducing the behaviour would immediately degrade the reported unlearning success.
> >
> > So our current metrics are not arbitrary averages and do target “recontamination”. They are targeted at the exact behaviours we want to remove, and are quite sensitive to their re-emergence.
> >
> > We agree that more fine-grained, per-example analyses (e.g., histograms of prediction changes on D_{\text{test},u}) could further illuminate rare failure modes, but we view this as an interesting extension rather than a core requirement for the main claims of the paper.
> >
> > > Q1: Study with fixed S_c, subsampling to form S, and generalization of forgetting
> >
> > Our current protocol already accounts for this. For a given corruption/manipulation mechanism (fixed way of constructing the corrupted set), we vary the forget-set size and measure unlearning performance and utility across multiple fractions, but the corrupted samples are fixed. This effectively probes how well corrections obtained from a small forget set extend to the broader corrupted distribution under that mechanism.
> >
> > > Q2: Measurements of accuracy/utility specifically on S_{\text{rel}}
> >
> > Conceptually, S_{\text{rel}} (retain samples whose activation pathways are closest to the forget examples) is exactly where incorrect unlearning would most likely hurt utility. Our experiments already stress challenging regimes where such overlap is substantial:
> > - Interclass confusion and label/feature manipulations, where the forget and retain distributions overlap in both input space and internal activations.
> > - Gradient-matching poisoning, where poisoned examples are deliberately placed near the clean data manifold.
> >
> > Thus, while we do not explicitly define and log a separate S_{\text{rel}} metric, the scenarios **we evaluate, for both accuracy/utility and unlearning success, are chosen so that many retain examples are “close” to the forget examples in practice**. A dedicated analysis based on activation-path proximity (defining S_{\text{rel}} explicitly and reporting utility restricted to it) is an interesting refinement; however, for computational and space reasons, we confined the current paper to the broader unlearning/utility metrics reported in the main text and appendix.
> >
> > > Q3: SalUn and RUM, memorization, and separability
> >
> > We agree that SalUn and RUM are relevant to the general unlearning landscape and appreciate the pointers.
> >
> > Our perspective is:
> > - SalUn and RUM operate primarily at the example level, using saliency or memorization scores to guide what to unlearn and how.
> > - SPARC operates at the parameter level, using activation-based contrasts to decide which parameters to damp (SPA) and which to use for utility recovery under orthogonal constraints (ReCalibration).
> >
> > Memorization levels and separability of pathways are indeed related: strongly memorized corrupted examples may carve out more distinct pathways, which could make parameter-level routing easier; weakly memorized ones may overlap more with clean behaviour. However, SPARC does not rely on a hard separability assumption and is designed to work in both backdoor-style (more separable) and interclass-confusion-style (less separable) regimes, as reflected in our experiments.
> >
> > A systematic empirical comparison against SalUn and RUM across all of our corrective settings, and an in-depth study of how memorization scores interact with activation-path influence, would require substantial additional engineering and compute. We therefore leave such a study, and potential hybrid methods combining memorization-aware policies with parameter influence routing, to future work.

---

> > > ### Comment · Reviewer_fV2S · 2025-11-26
> > >
> > > Thank you for your explaination regarding Q1.
> > >
> > > For the other comments, I remain unconvinced, unfortunately. Even if we accept that looking into activation paths is the way to go, IMHO you need to show that scenarios where there exist good and bad examples sharing such activation paths are not catastrophic for your approach. Unless you can convincgly argue against the existence of such scenarios. Are these scenarios not relevant in the real world?
> > >
> > > My earlier proposal (ie identifying nearest neighbours) is likely to reveal such cases.
> > > You claim that with your mechanisms these cases are likely to emerge. But do you ever show this is the case?
> > >
> > > Why am I making a big deal of this: Corrective learning approaches hinge on exactly maintaining utility. The latter depends on exactly what happens to other *neighbouring* data - for clearly separated data, this is not an issue (no problem to solve). So corrective learning papers **must** be very comprehensively studying this issue.

---

> > > > ### Author Response · Authors · 2025-12-03
> > > >
> > > > **On the “nearest-neighbour” diagnostic and activation-path overlap**
> > > >
> > > > We agree that retain samples that lie close to the forget set are the most challenging regime for corrective unlearning. Our evaluations already target precisely these regimes, though not under the explicit terminology of “nearest neighbours.”
> > > >
> > > > In the inter-class confusion setting, the retain and forget samples have substantial overlap in input space. In the gradient-matching poisoning case, poisoned examples are explicitly constructed to align gradients with clean targets, leading to substantial overlap in the internal activations. These are exactly the scenarios where activation paths between retain and forget data coincide, and where a method that implicitly requires separability would fail.
> > > >
> > > > If SPARC assumed such separability, it would manifest immediately as a collapse in utility on these entangled regimes. Instead, SPARC matches or outperforms naive retraining in utility while achieving strong unlearning across these settings. Because utility on these tasks is dominated by behaviour on overlapping regions, the reported metrics already reflect SPARC’s performance on the mentioned "trickiest cases."
> > > >
> > > > We agree that an explicit S_rel diagnostic, e.g., defining a neighbourhood via embedding distance or activation-path proximity and isolating utility on this subset, would be a useful additional analysis. However, this requires several non-trivial design choices to define and calculate these subsets. Our focus in this work is on demonstrating SPARC’s behaviour across settings where activation overlap is inherent, which provides a broad and realistic stress-test of the concern raised.
> > > >
> > > > **On baselines and inclusion of SalUn / RUM**
> > > >
> > > > We agree that SalUn and RUM are interesting recent methods. However, they are not directly designed for the corrective unlearning setting we study, and they do not drop into our tasks in a trivial way.
> > > > - SalUn is formulated and evaluated primarily in standard classification unlearning settings, not in scenarios where (i) only a subset of corrupted data is known and (ii) the goal is to correct backdoors, targeted poisons, or regression-range removal.
> > > > - RUM is explicitly memorization-aware and operates at the level of per-example policies; its design and evaluation similarly focus on classical unlearning regimes rather than the corrective setting with partial knowledge of D_u and multiple manipulation types.
> > > >
> > > > In contrast, our experiments are built around settings that are specifically introduced and studied as corrective unlearning benchmarks. For these, we already include the methods that prior work identifies as the most relevant and competitive in this regime, i.e., algorithms that either have established behaviour under corrective-style evaluations or have been explicitly used as baselines in this context.
> > > >
> > > > Adapting SalUn and RUM to all of these corrective settings (including backdoor and regression) would require non-trivial redesign and tuning. A thorough study of such adaptations and of potential hybrids combining memorization-aware policies with SPARC’s parameter-level activation routing is an interesting direction for follow-up work, but is beyond the scope of this paper.

---

> > ### Comment · Reviewer_fV2S · 2025-11-26
> >
> > On W1 amd W2:
> >
> > Have I missed tests/experiments where you show that the nearest examples (e.g., in embedding space) are unaffected? Or studying how said nearest examples (the trickiest cases?) are affected by your proposal? Is the request to study this in depth relevant/significant? Is that question not central in your stated selling point of "Given a limited and potentially incomplete forget set, SPARC achieves strong corrective unlearning with a small unlearning budget, and with a better utility trade-off than existing methods"?
> >
> > Finally, researchers want to know much more than simply saying that in this task "we are better than others...". The **question is how good is the approach, in terms of maintaining utility, **(again for the tricky cases, looking into the impact into nearest neighours).
> >
> > It was the lack of answers to these/such questions that led to think that implicitly you assume clear separations between good and bad examples.
> >
> > On W3:
> > "baseline selection was guided by two criteria:...". Your baseline selection methods should have been guided by using the most relevant and best-performing methods, IMHO. The key question is: does your contribution help when using the best/relevant algorithms? While I (as we all) do appreciate constraints on paper length and compute etc. the key question I pose above should be considered by a paper for a top-venue, no?

---

### Official Review · Reviewer_Jrti · 2025-10-31

**Soundness:** 3
**Presentation:** 3
**Contribution:** 3
**Rating:** 6
**Confidence:** 4

**Summary:**

This paper introduces SPARC, a novel, architecture-agnostic algorithm for corrective machine unlearning. Unlike prior methods designed primarily for privacy-oriented unlearning, SPARC targets correcting the influence of corrupted or poisoned data. The method has two stages: a) Selective Parameter Adjustment (SPA) which identifies and attenuates parameters with high influence on the forget set relative to the retain set via an activation-based influence score, b) ReCalibration, which is an orthogonal gradient descent on low-influence parameters to restore utility while avoiding relearning of forget samples.Experiments across multiple datasets and model architectures show that SPARC achieves competitive or superior performance in both unlearning success and retained utility under various settings (classification, regression, feature/label/label+feature manipulation), with a small computational budget.

**Strengths:**

**S1**.  The influence computation is forward-pass only, making the approach computationally efficient. Demonstrating strong results with as little as 2% forget-set data is compelling for real-world applications.

**S2**. The two-stage forget then recalibrate process is logical (and has appeared in prior unlearning works too), and the core contributions are strong.

**S3**. The experiments are comprehensive covering different baselines, datasets, and model architectures.

**Weaknesses:**

**W1**. Although the combination is novel, both components i.e., parameter influence estimation and orthogonal gradient descent, draw heavily from existing ideas. The contribution lies in their integration for unlearning rather than a new learning principle.

**W2**. The authors report that for regression unlearning, the SPA component (forgetting-only) performs best, and the full SPARC algorithm (with recalibration) underperforms it. This is a significant finding that directly contradicts the general applicability of the full SPARC method. The paper's justification, that it is "less suited for continuous targets" is vague and unsatisfying.

**W3**. Although efficiency is discussed qualitatively, the paper lacks explicit wall-clock or FLOPs comparisons vs. baselines like SCRUB or SSD to substantiate “sample-efficient” claims quantitatively.

**W4**. The paper can benefit from more ablations. For example, what happens if the influence measuring method is replaced by gradient-based influences?  What happens if the activation path is longer than just one layer before and after the parameter? What is the effect of the choice of the activation function?

**W5**. I suggest moving section 2.1 to after section 3, as it’s more similar to the experimental setup.

**Questions:**

**Q1**. How sensitive is the influence measure to activation function choice or network depth?

**Q2**. In the regression setting, did you measure how far predictions are pushed outside the forget range (i.e., margin of forgetting)?

**Q3**. How would SPARC perform if the recalibration step were replaced by ordinary fine-tuning without orthogonal projection?

---

> ### Author Response · Authors · 2025-11-21
> **Rebuttal (Part-1)**
>
> > W1: Novelty vs. prior ideas:
>
> We agree that both components of SPARC build on existing ideas, but we see our contribution as more than a simple integration:
> - **Unlearning-specific influence measure:** Our influence measure is explicitly defined as a forget-vs-retain contrast along the activation path of each parameter, using only forward activations. This differs from classic importance or influence estimates and allows the same parameter to be high-importance or low-importance, depending on the relative activation discrepancy between the forget and retain sets, which is a corrective-unlearning-specific design.
> - **Two-stage, parameter-routed corrective unlearning:** Prior work like PCGrad is designed for multi-task learning, where the goal is to satisfy all tasks simultaneously. Our contribution is to adapt this geometric idea to corrective unlearning via a two-stage design:
>     - SPA: explicitly damp high-influence parameters (forgetting step).
>     - ReCalibration: update only low-influence parameters along directions orthogonal to the forget gradient (utility recovery without relearning).
>
>     This routing of forgetting on high-influence parameters and recalibrating on low-influence parameters under an orthogonality constraint is specific to the unlearning problem and, to our knowledge, novel.
> - **Architecture-agnostic corrective unlearning with strong sample efficiency:** Because the influence measure is activation-based and layer-local, the same algorithm applies unchanged to CNNs and ViTs, and to classification, regression, and multiple corruption types (feature/label/feature+label). The fact that we can achieve strong unlearning with as little as 2% forget-set data and 1–2 epochs of unlearning (vs. 10% of training for many baselines) is precisely a result of this design.
>
> > W2: Regression Unlearning:
>
> We agree that the current phrasing (“less suited for continuous targets”) is too vague and can be misread as a contradiction. Our intended message is:
> - **Empirical observation (already in the paper):** In the regression setting, SPA (forgetting-only) gives the best trade-off between unlearning and retain-set performance. The full SPARC (with recalibration) is still competitive but slightly weaker because the recalibration step partially softens the aggressive forgetting produced by SPA.
> - **Reason for this behavior:** In regression, forget and retain targets lie on a continuous range. Consequently, the retain-set gradients often point in similar directions to those that were used to fit the forget range. Orthogonal projection in this case tends to shrink effective updates for low-influence parameters, which is beneficial in discrete-label settings (where we want to avoid re-learning a specific class or trigger) but can be conservative when we want to avoid just a sub-interval of outputs and still fit nearby continuous values.
>
> We have clarified this in the revision.
>
> > W3: FLOPs comparison vs. baselines:
>
> We have added the comparison in the appendix of the rebuttal revision.
>
> > W4: More ablations:
>
> We agree that the suggested ablations are interesting directions, but due to space/compute constraints, we deliberately focused on breadth across datasets, architectures, and corruption types rather than a large grid of micro-ablations. We respond to each suggestion:
> - **Gradient-based influences:** In a corrective-unlearning setting we need a relative notion of influence: parameters that matter “more for forget than retain.” With gradients, this requires somehow combining two gradient signals (one from the forget set, one from the retain set). In practice, this leads to an additional hyperparameter that trades off the two (as done in SSD), and this trade-off is quite task- and scale-dependent. Our influence score is defined directly as a contrast between forget and retain activations along the upstream–downstream path of each parameter. This makes the relative nature of influence explicit and eliminates the need to introduce a new hyperparameter to balance the forget and retain sets.
> - **Longer activation paths:** Our influence score uses direct upstream/downstream activations (the immediate “path around” a parameter). Extending to longer paths is straightforward in principle, but it will dilute the local signal that is critical for deciding which parameters to damp.
> - **Choice of activation function in the influence score:** During initial research, we conducted preliminary tests with different activation function choices (i.e., ReLU, LeakyReLU, GELU, tanh,  sigmoid, ELU, and Mish) and observed that the overall trends in unlearning vs utility remain stable. We did not include these small-scale variants to keep the main text focused, and this was due to the nature of the experiments. We suggest that a more robust comparison of the performance of unlearning algorithms, particularly in relation to the choice of activation functions, could be an avenue for future research.

---

> > ### Author Response · Authors · 2025-11-21
> > **Rebuttal (Part-2)**
> >
> > > W5: Section ordering:
> >
> > We appreciate the suggestion. We experimented with both orderings and, based on feedback from colleagues and early reviewers, decided to place Section 2.1 before Section 3. Their main comment was that seeing the unlearning settings and problem variants early makes it easier to follow the algorithmic design choices that come right after.
> >
> > > Q1: Sensitivity to activation function and network depth:
> >
> > **Activation Function:** As noted in the W4, we conducted preliminary experiments where we replaced the activation function with alternatives and found that the ranking of highly influential parameters and the overall unlearning/utility trade-off curves remained stable. This suggests that SPARC is not highly sensitive to the precise choice of the activation function.
> >
> > **Network Depth:** Our influence measure is inherently local: we use immediate pre/post activations of each parameter, not a path that grows with depth. Hence, while deeper models have more parameters that can potentially be influenced, the definition of influence and the per-parameter cost remain unchanged. The fact that SPARC works consistently on AllCNN, ResNet, and ViT in our experiments supports that the method scales naturally with depth.
> >
> > > Q2: Margin of forgetting in regression:
> >
> > We agree that the “margin of forgetting” is a meaningful additional metric. In our current experiments, we focus on two criteria:
> > - Unlearning Success: predictions on forget-range points move outside the target interval
> > - Utility: RMSE on retain data remains competitive.
> >
> > This ensures that predictions do not merely stay within the forget range; however, we do not explicitly log the distance to the interval boundaries as a separate statistic.
> >
> > We agree that a more detailed analysis of the full margin distribution is a natural extension of this work, especially for safety-critical regression tasks.
> >
> > > Q3: Replacing orthogonal recalibration with ordinary fine-tuning:
> >
> > The recalibration step in SPARC updates low-influence parameters along directions orthogonal to the gradient signal associated with the forget set. This is precisely to prevent the model from drifting back toward the pre-unlearning solution. Ordinary fine-tuning on retain data (without orthogonal projection) effectively removes this constraint. In settings where the forget and retain gradients are partially aligned, such fine-tuning will naturally push parameters back in directions that re-learn the forgotten behavior.
> >
> > Our experiments show that naive retraining under a comparable budget does not achieve the unlearning performance of SPARC, and the combination of SPA and orthogonal recalibration improves the unlearning/utility trade-off over both NR and purely forgetting-only baselines.

---

> > > ### Comment · Reviewer_Jrti · 2025-11-27
> > >
> > > Thank you for the clarifications. Regarding 'regression unlearning', I still find the explanation speculative, as it's not supported by any proper tests. Regarding the ablations, I think understanding why and how a method works is as important as the results. Through some of these ablations, one may find that certain design choices are suboptimal or certain components are unnecessary.

---

> > > > ### Author Response · Authors · 2025-12-03
> > > >
> > > > Thank you again for the thoughtful follow-up.
> > > >
> > > > **On regression unlearning**
> > > >
> > > > We agree that our current explanation of why SPA outperforms SPA+ReCalibration in our regression setting is a hypothesis rather than a tested claim. In the revision, we will explicitly mention this.
> > > >
> > > > We also want to stress that, despite this nuance, an important part of the contribution is that the same SPARC operates across both classification and regression settings without architectural modifications, whereas prior non-trivial unlearning methods (to our knowledge) are typically tailored to one or the other and require non-trivial modifications to be applied for both settings. Because of this Regression Unlearning is an important setting for our experiments even without concrete evidence of the reason behind the difference in recalibration performance.
> > > >
> > > > **On ablations**
> > > >
> > > > We completely share your view that understanding why and how a method works is as important as the results, and that ablations can reveal suboptimal or unnecessary choices.
> > > >
> > > > A point that may not have been sufficiently emphasized in the text is that SPARC is not defined around fixed components. For example, in the influence measure we explicitly introduce:
> > > > - a generic function \phi : \mathbb{R} \to \mathbb{R} (e.g., absolute value, positive part), and
> > > > - a generic normalization operator N(\cdot) (global, layer-wise, or hybrid).
> > > >
> > > > Appendix B.1 clarifies that, in the current experiments, we instantiate these as the positive-part function and global max-scaling, but other choices are allowed and discussed as alternatives. Thus, SPARC is intended as a template with explicit design slots, and we do not claim that the specific instantiations we used are uniquely optimal.
> > > >
> > > > In the camera-ready paper, we can add a compact set of ablations on a small subset of settings.
> > > >
> > > > We appreciate your push toward a sharper evidence vs hypothesis separation and toward deeper analysis of the design choices, and we will incorporate this perspective and the above clarifications into the camera-ready version.

---

### Official Review · Reviewer_HTkw · 2025-11-01

**Soundness:** 2
**Presentation:** 2
**Contribution:** 3
**Rating:** 4
**Confidence:** 2

**Summary:**

This paper proposes a sample-efficient correction framework for machine unlearning. The author aims to reduce the reliance on the knowledge of data points to unlearn without degrading model performance. The proposed approach contains two steps: 1. a targeted parameter adjustment step to selectively forget undesired data 2. a recalibration phase that updates low-importance parameters
to recover overall model utility. The approach is evaluated on several benchmark datasets, showing strong performance while using fewer correction samples and lower computational cost.

**Strengths:**

1. The focus on sample-efficient correction directly addresses a key problem in existing unlearning methods.
2. The paper provides a clear problem formulation and the motivation is well-grounded.
3. The paper provides extensive experiments across multiple datasets, architecture and scenarios.
4. The paper presents experiments on multiple datasets, model architectures, and unlearning scenarios. The results are solid, demonstrating clear improvements in computational efficiency and comparable accuracy recovery relative to full retraining.

**Weaknesses:**

1. Experimental results are presented as plots without numerical tables. Some plots are dense and difficult to read precisely. Providing corresponding tables with numerical values would make the comparisons clearer and improve interpretability/readablity of the results.
2. The overall experimental scope remains moderate. It does not include large-scale datasets that could demonstrate scalability.
3. The design choices are described at a high level, but their rationale and potential alternatives are not thoroughly discussed.
4. While the paper clearly explains the overall framework and includes a basic ablation study on module contributions, the rationale behind key design choices is not well justified/discussed. Moreover, no detailed sensitivity analysis is conducted to examine robustness to important hyperparameters.

**Questions:**

1. How does the proposed perform on large-scale datasets?
2. Would the authors consider providing numerical tables corresponding to the plots? Many figures are dense and difficult to read precisely, and tables could make the quantitative comparisons clearer and easier to interpret.
3. Could the authors elaborate on the motivation behind specific design choices in the proposed method? Were alternative designs considered, and how did they perform?
4. Beyond the current ablation, have the authors examined the effect of important hyperparameters? A more detailed sensitivity analysis could help assess the robustness and general applicability of the proposed under different conditions.

---

> ### Author Response · Authors · 2025-11-21
>
> > Q1:How does the proposed perform on large-scale datasets?
>
> Our primary design choice was to emphasize breadth across settings rather than a single large-scale dataset. We evaluate SPARC on multiple datasets (CIFAR-10/100, Lacuna-10, AgeDB), architectures (AllCNN, ResNet, ViT), and unlearning scenarios (backdoor, label/feature removal, class removal, and regression). This enables us to demonstrate that the exact algorithm is effective across various architectures, tasks, and corruption types.
>
> Regarding scalability: SPARC is explicitly designed to be efficient, and its cost decomposes into:
> - Influence estimation via forward passes on the forget and retain sets (no backward pass), which scales linearly with data size
> - Recalibration via a small number of epochs (1–2 in most experiments), restricted to low-influence parameters
>
> Thus, SPARC remains a strictly cheaper unlearning algorithm by construction, and our comparisons already report cost as a fraction of naive retraining. In the revision, we have added a FLOPs comparison of SPARC and baseline algorithms in the appendix to strengthen the efficiency and scalability claims.
>
> >Q2: Would the authors consider providing numerical tables corresponding to the plots? Many figures are dense and difficult to read precisely, and tables could make the quantitative comparisons clearer and easier to interpret.
>
> We have added the tables in the appendix for your reference.
>
> > Q3: Could the authors elaborate on the motivation behind specific design choices in the proposed method? Were alternative designs considered, and how did they perform?
>
> Our main design choices were guided by the specific structure of the corrective unlearning problem:
> - Influence measure: We chose a forward-activation-based, forget-vs-retain contrast because it directly captures how much a parameter’s activation pathway disproportionately supports the forget set relative to the retain set. This makes it naturally suited for corrective unlearning, where the goal is not global importance but relative importance to the data to be forgotten. We will clarify this motivation more explicitly in the method section by emphasizing that the same parameter can have high or low influence depending on this contrast, which is what we exploit in the SPA step.
> - Two-stage structure (SPA + ReCalibration): We use a two-stage pipeline to separate forgetting from utility recovery: SPA explicitly dampens parameters that are highly biased toward the forget set, while ReCalibration uses only low-influence parameters to restore overall performance. This separation enables us to use a very small number of unlearning epochs while preserving utility. We will make this design rationale more explicit in the text, rather than presenting the steps as merely procedural.
> - Orthogonal updates in ReCalibration: We adopt orthogonal updates so that, when there is a conflict between forget and retain gradients, the recalibration step does not move parameters back in directions that relearn the forgotten behavior. Conceptually, this is a natural adaptation of geometric gradient ideas to the unlearning setting: instead of balancing multiple tasks, we treat the forget gradient as a “do-not-revisit” direction and restrict utility recovery to directions that do not reintroduce it. We will add a short explanation to highlight this interpretation.
>
> We did consider straightforward alternatives such as magnitude-only importance scores and standard fine-tuning on retain data without orthogonalization. These behaved similarly or worse than the baselines we already report (e.g., naive retraining under a comparable budget or forgetting-only variants).
>
> > Q4: Beyond the current ablation, have the authors examined the effect of important hyperparameters? A more detailed sensitivity analysis could help assess the robustness and general applicability of the proposed under different conditions.
>
> Section 5.1 and Figure 7 already present a 20x20 grid sensitivity analysis over the two key hyperparameters (forgetting factor \gamma and threshold \tau) for the CIFAR-10 / ResNet-18 setting with 100% forget-set size, showing:
> - Broad regions where both unlearning success and utility remain high
> - Smooth trade-offs rather than brittle behavior
> - That \gamma and \tau control different aspects of the trade-off (aggressiveness vs selectivity)
>
> We also suggest reviewing the response to reviewer Jrti for W4 for more context regarding this.

---

### Author Response · Authors · 2025-12-03
**Overview of Reviews and Responses**

To help the Area Chairs and readers quickly see how we addressed the main concerns across reviews and follow-up discussions, we summarize below:

**Sample efficiency and use of the retain set (S1KG, HTkw):**

By “sample-efficient” we refer to the aspect that SPARC achieves strong corrective unlearning from a small forget set (down to 2% of the corrupted pool in our experiments). Also, it uses a small unlearning compute budget compared to retraining and prior methods. SPA requires a single epoch of forward passes on the forget and retain sets to compute activation-based influence scores, followed by a closed-form update; SPARC adds only one additional epoch of orthogonal recalibration. All competing methods are run with a 10-epoch unlearning budget under the same training setup. We have added a FLOPs-based comparison in the appendix to explicitly quantify cost relative to naïve retraining and baselines. While we do touch the retain set (as any corrective unlearning method must), the total unlearning cost remains a small fraction of training, and our comparisons are made under a common budget.

**Computational efficiency: FLOPs / wall-clock time (S1KG, Jrti):**

Implementation-wise, SPA’s dominant cost is a forward-only pass over D_f and D_r to compute activation differences, followed by a cheap parameter update, while ReCalibration adds a single epoch of standard mini-batch updates with orthogonal projections restricted to low-influence parameters. We now provide FLOPs-based comparisons in the appendix, showing that SPARC’s total unlearning cost is strictly below naïve retraining and competitive with prior methods under the same budget. Exact wall-clock times will depend on hardware and implementation, but the FLOPs analysis is meant to clarify the relative cost profile of SPARC vs. baselines.

**Regression unlearning and SPA vs SPARC behaviour (Jrti):**

In regression settings, SPA (forgetting-only) empirically yields the best unlearning/utility trade-off, whereas SPARC with orthogonal recalibration is competitive but slightly weaker. Our earlier explanation that continuous targets lead to greater gradient alignment between forget and retain ranges, making orthogonal projection conservative, was indeed a hypothesis rather than supported by dedicated tests. In the paper, we explicitly label this as a hypothesis and avoid presenting it as a definitive explanation. We also stress that one key contribution is that the same SPARC template operates across both classification and regression without architectural modifications. In contrast, prior non-trivial unlearning methods are typically tailored to one or the other. The regression setting therefore, remains important even though the exact mechanism behind SPA vs SPARC performance differences deserves deeper future analysis.

**Separation / overlap of “bad” vs “good” activation paths and nearest-neighbour behaviour (fV2S):**

Our method does not assume that “bad” and “good” behaviours are cleanly separable in activation space. SPARC uses a relative influence measure, contrasting activations on the forget vs retain sets, and is explicitly evaluated in regimes where overlap is substantial: inter-class confusion, label/feature manipulations, and gradient-matching poisons, where corrupted examples are constructed to lie close to the clean manifold and share activation pathways. In these regimes, if SPARC implicitly required separability, we would expect a sharp collapse in utility, which we do not observe. We agree that an explicit “nearest-neighbour” diagnostic (e.g., defining an S_rel subset via embedding or activation-path proximity and reporting utility restricted to it) would be valuable. However, this requires non-trivial experimentation design and additional computation, and we view it as an extension rather than a prerequisite for the main contribution. Our current experiments are deliberately chosen to already stress the “trickiest cases” where retain and forget behaviours overlap.

**Risk of “recontamination” after recalibration (fV2S):**

We agree that any corrective unlearning method must guard against reintroducing unwanted behaviour. SPARC’s design addresses this in two ways. First, the recalibration step is restricted to low-influence parameters and uses orthogonal updates with respect to the forget-set gradient directions, treating those directions explicitly as “do-not-revisit.” Second, our evaluation metrics are targeted at the undesired behaviour: backdoor/poisoning experiments measure attack success on triggered inputs; classification/regression removal experiments measure predictions on designated unlearn test sets D_{\text{test},u}. Any substantial recontamination would directly show up as degraded unlearning success on these focused test sets. We agree that more fine-grained, per-example analyses would be informative and are a promising direction for follow-up work.

---

> ### Author Response · Authors · 2025-12-03
> **Overview of Reviews and Responses (Part-2)**
>
> **Ability to use only a subset of the retain set and impact on performance (S1KG):**
>
> Conceptually, SPARC can operate with a subset of the retain set. In SPA, the influence measure depends on empirical means of activations over D_f and D_r; these can be estimated using any representative subset \tilde{D}_r \subseteq D_r. In ReCalibration, we already rely on mini-batch SGD, so each update step uses only a small subset of retain examples. In our experiments, we used the full retain set with the same data-loader protocol as baselines, to keep the comparison fair and controlled. A systematic study of performance vs. subsampling ratio (e.g., using only 10% of D_r) is an interesting orthogonal axis; we explicitly leave this as future work.
>
> **Ablations, sensitivity to design choices, and SPARC as a template (HTkw, Jrti, S1KG):**
>
> We fully agree that ablations are crucial for understanding “why” a method works. Our current paper includes a 20×20 grid over the two key hyperparameters \gamma (forgetting factor) and \tau (influence threshold) on CIFAR-10 / ResNet-18, showing broad regions with high unlearning and utility and smooth trade-offs, indicating that SPARC is not brittle in these choices. For the internal components of the influence measure (incoming vs outgoing activations, function \phi, normalization N(\cdot)), SPARC is explicitly defined as a template: we introduce slots for a generic \phi : \mathbb{R} \to \mathbb{R} (e.g., absolute value, positive part) and a generic normalization N(\cdot) (global, layer-wise, or hybrid). Appendix B.1 specifies how we instantiate these in current experiments, but other choices are allowed. During development, we tested straightforward variants (e.g., using only incoming or only outgoing activations, alternative simple activations) and observed similar qualitative trends, with the final variant giving the most stable performance overall. Due to computational and space constraints, we prioritized breadth (datasets, architectures, corruption types) over an exhaustive micro-ablation grid. In a camera-ready version, we can include a compact set of additional ablations on a subset of settings and make the “template vs instantiation” distinction more explicit.
>
> **Baselines and missing recent methods (fV2S, S1KG):**
>
> Our baseline choice followed two criteria: (i) methods that are widely used or discussed in the unlearning literature, and (ii) methods that have been evaluated in corrective-style settings (backdoors, poisons, etc.) and are known to struggle or compete there, such as SCRUB, SSD, Gaussian-Amnesiac, Blindspot, and CF_k/EU_k. We agree that SalUn, RUM, and closely related works are relevant to the broader unlearning landscape. However, they are not designed for exactly the corrective setting we study, and do not trivially apply to our tasks with partial knowledge of corrupted data and multiple manipulation types (backdoor, gradient-matching poisoning, regression-range removal, etc.). SalUn and RUM operate primarily at the example level via saliency/memorization scores, whereas SPARC operates at the parameter level via activation-path influence contrasts. Adapting these methods to all our corrective scenarios (including regression and backdoor) in a fair and well-tuned way would require substantial additional engineering and compute. We position this as a valuable follow-up direction, including possible hybrids that combine memorization-aware policies with SPARC’s parameter-routing, rather than something we can realistically include within the current paper’s scope.
>
> **Representativeness and coverage of the forget set, and generalization of forgetting (fV2S):**
>
> We agree that the quality and coverage of the forget set are central in corrective unlearning. Our experiments already vary the forget-set size over a wide range (down to small fractions of the corrupted pool) under a fixed corruption mechanism, and measure how unlearning and utility behave as the forget set becomes more or less representative. This effectively probes generalization from a small forget set to the larger corrupted distribution. Moreover, the scenarios we choose (backdoors, gradient-matching poisons, inter-class confusion, regression-range removal) are precisely those where corrupted and clean behaviour are entangled, so that the forget set is not trivially representative. We do not claim that any arbitrary, poorly chosen forget set yields perfect forgetting with no utility loss; instead, our claim is that given a limited and potentially incomplete forget set, SPARC yields a better unlearning–utility trade-off than existing methods under the same budget. More fine-grained analyses of coverage would be valuable but are beyond the current scope.

---

> > ### Author Response · Authors · 2025-12-03
> > **Overview of Reviews and Responses (Part-3)**
> >
> > **Large-scale datasets and scalability (HTkw):**
> >
> > We chose to emphasize breadth across settings (multiple datasets, architectures, and corruption types) rather than a single very large-scale dataset. Scalability is built into SPARC’s design: influence estimation uses forward passes only and scales linearly with data size, and recalibration runs for a small, fixed number of epochs on low-influence parameters. We now make this clearer in the text and supplement it with FLOPs comparisons. We view applying SPARC to truly large-scale (e.g., ImageNet-scale) corrective unlearning as an important but orthogonal engineering step and a promising direction for future empirical work.
> >
> > **Presentation issues: dense plots, missing tables, and section ordering (HTkw, S1KG, Jrti):**
> >
> > We have added numerical tables corresponding to dense plots in the appendix, so that quantitative comparisons are easier to read. For efficiency and scaling, we added FLOPs-based tables as well. Regarding section ordering, we experimented with different placements of the unlearning settings section; based on feedback from colleagues, early readers, and reviewers, we kept the current ordering because presenting the unlearning problem variants early made it easier for readers to contextualize the subsequent algorithmic design. If accepted, we are happy to further polish figure layout, cross-references, and section transitions to improve readability.
> >
> > We thank all reviewers and the Area Chairs again for the detailed feedback. We believe the revisions, added analyses, and clarifications address the core concerns while keeping the focus on SPARC’s main contribution: an activation-path-based, architecture-agnostic corrective unlearning framework that achieves strong unlearning-utility trade-offs with small forget sets and modest unlearning budgets across a diverse set of corruption types.

---

### Meta-Review · Area_Chair_KsFR · 2026-01-07

**Summary:**

This paper proposes a new approach for corrective machine unlearning and evaluates it across multiple datasets, architectures, and unlearning scenarios. Reviewers generally agreed that the problem is practically motivated and that the method has broad empirical coverage. However, several reviewers raised substantive concerns regarding the clarity of the method’s implicit assumptions, the alignment between key claims and supporting evidence, and the maturity of the contribution. Specifically, concerns focused on (i) insufficient analysis of challenges (e.g., overlapping representations or nearest-neighbour effects), (ii) novelty, the extent to which the contribution represents integration of existing ideas rather than a new learning principle, (iii) ambiguity and limited empirical support for core claims such as sample efficiency and broad applicability, and (iv) the lack of ablations to assess key design choices and potential failure modes.

**Reviewer Concerns:**

Concerns addressed:

1 The rebuttal provided clearer intuition for the design, the influence-based parameter selection, and the role of orthogonal updates, which helped clarify the implementation details and rationale.

2 Some presentation and related-work issues were addressed.

3 The authors discussed existing hyperparameter sensitivity results, suggesting a degree of robustness in specific settings.

Concerns that remain:

1 Key concerns regarding implicit assumptions were not. The rebuttal relied primarily on mechanism-level intuition and indirect arguments rather than direct evidence in the most challenging regimes.

2 In regression settings, the full method underperformed a partial variant, and the authors acknowledged that. This weakens claims of general applicability.

3 The definition and evidence for “sample efficiency” remain ambiguous. Although rebuttal redefine the “sample efficiency” in a different way compared to existing works, the contribution is then reduced compare the method needs access to full training data. Practical scalability and wall-clock implications were not conclusively demonstrated.

4The necessity of key design choices (e.g., influence definition, component interactions, and alternatives) was not established through systematic ablations.

**Reviewer Scores:**

Reviewer HTkw: While some clarifications were provided, the core concerns were not fully resolved; the score would most likely remain unchanged.

Reviewer Jrti: Raised concerns about novelty and internal consistency regarding regression results. Subsequent discussion did not alleviate these concerns; the score would not increase.

Reviewer fV2S (the reviewer giving the lowest score): Reviewers raised questions about the basic assumption of the paper. Despite multiple clarifications, the requested evidence was not provided; the reviewer’s position will remain unchanged.

Reviewer S1KG: Emphasized consistency between definitions and implementation, especially around sample efficiency and computational cost. The rebuttal clarified some but did not fully resolve the concern; the score would probably remain unchanged or lower.

---

### Decision · Program_Chairs · 2026-01-26

Reject